# Pre-movement sensorimotor oscillations shape the sense of agency by gating cortical connectivity

Tommaso Bertoni [1,2] ✉, Jean-Paul Noel [3], Marcia Bockbrader[4], Carolina Foglia[1], Sam Colachis[5], Bastien Orset[6], Nathan Evans[6], Bruno Herbelin[6], Ali Rezai [7], Stefano Panzeri [8], Cristina Becchio [2,9], Olaf Blanke [6,10] & Andrea Serino [1,10]

Our sense of agency, the subjective experience of controlling our actions, is a crucial component of self-awareness and motor control. It is thought to originate from the comparison between intentions and actions across broad cortical networks. However, the underlying neural mechanisms are still not fully understood. We hypothesized that oscillations in the theta-alpha range, thought to orchestrate long-range neural connectivity, may mediate sensorimotor comparisons. To test this, we manipulated the relation between intentions and actions in a tetraplegic user of a brain machine interface (BMI), decoding primary motor cortex (M1) activity to restore hand functionality. We found that the pre-movement phase of low-alpha oscillations in M1 predicted the participant's agency judgements. Further, using EEG-BMI in healthy participants, we found that pre-movement alpha oscillations in M1 and supplementary motor area (SMA) correlated with agency ratings, and with changes in their functional connectivity with parietal, temporal and prefrontal areas. These findings argue for phase-driven gating as a key mechanism for sensorimotor integration and sense of agency.

The sense of agency refers to the subjective feeling of causing and controlling our actions[1], which enables us to perceive ourselves as autonomous, self-governing agents. This feeling is fundamental to self-awareness[1–3]. Disturbances in the sense of agency are associated with psychiatric disorders, such as schizophrenia and autism[4,5]. Different explanations have been offered for the sense of agency[6–8]. Among those, one influential view states that the sense of agency arises from the comparison between predicted and observed sensory outcomes of intended actions[8,9]. If they match, a sense of agency over the intended action is experienced. Due to its link with basic sensorimotor mechanisms, it is likely that the sense of agency does not merely accompany volitional motor control, but also directly contributes to it[10,11]. This makes sense of agency important not only for natural closed-loop control but also for artificial control loops such as in prosthetics and brain-machine interfaces[12,13].

[1]MySpace Lab, Department of Clinical Neuroscience, University Hospital Lausanne (CHUV), Lausanne, Switzerland. [2]C'MoN, Cognition, Motion and Neuroscience Unit, Fondazione Istituto Italiano di Tecnologia, Genova, Italy. [3]Department of Neuroscience, University of Minnesota, Minneapolis, Minnesota, USA. [4]Department of Physical Medicine and Rehabilitation, The Ohio State University, Columbus, Ohio, USA. [5]Medical Devices and Neuromodulation, Battelle Memorial Institute, Columbus, Ohio, USA. [6]Neuro-X Institute, Faculty of Life Sciences, Swiss Federal Institute of Technology (EPFL), Lausanne, Switzerland. [7]Rockefeller Neuroscience Institute, West Virginia University, Morgantown, West Virginia, USA. [8]Institute for Neural Information Processing, Center for Molecular Neurobiology (ZMNH), University Medical Center Hamburg-Eppendorf (UKE), Hamburg, Germany. [9]Department of Neurology, University Medical Center Hamburg-Eppendorf (UKE), Hamburg, Germany. [10]These authors contributed equally: Olaf Blanke, Andrea Serino. ✉ e-mail: tommaso.bertoni90@gmail.com

To perform the sensorimotor comparisons that underlie the sense of agency, the brain must integrate pre- and post-movement signals on a large scale, orchestrating the information flow among functionally specialised but widely distributed brain regions. Consistent with this notion, a large fronto-parietal network, including the premotor cortex, the supplementary motor area (SMA), the angular gyrus and the dorsal parietal cortex, has been implicated in the sense of agency[14–20]. Specifically, it has been suggested that the neural substrate for the sense of agency may lie in the connectivity between the frontal motor areas involved in action initiation, and parietal areas that monitor their sensory consequences[21]. However, empirical studies on neural connectivity are scarce and not conclusive. One functional neuroimaging (fMRI) study highlighted connectivity between the angular gyrus and dorsolateral prefrontal cortices[17], while one magnetoencephalography (MEG) study emphasised connectivity between the primary motor cortex (M1), middle temporal gyrus and insular cortex[22]. Furthermore, the brain-wide neural dynamics regulating neural connectivity as a function of sensory feedback and sense of agency are not well understood. Most previous studies focused on post-movement processing related to agency (e.g., ref. 23). A separate line of research investigated pre-movement intentions (e.g., refs. 24,25). Indeed, previous neuroimaging studies lacked the temporal precision needed to resolve the fast dynamics of this process[17,26]. Other studies[14,16], mainly utilising invasive recordings, could gather temporally resolved information but lacked the spatial coverage to link it to large-scale neural processes. In sum, we still lack specific knowledge of the neural mechanisms regulating the integration of endogenous pre-movement signals with post-movement reafferent information.

Mounting evidence indicates that theta (4–8 Hz) and alpha band (8–13 Hz) oscillations play a crucial role in coordinating information exchange[27,28]. Their pre-stimulus phase and/or power have been shown to correlate with changes in perceptual abilities[29,30], neural connectivity[31] and information integration across sensory modalities[32]. In the motor domain, the phase of M1 alpha oscillations has been shown to modulate corticospinal excitability and cortical responses induced by transcranial magnetic stimulation (TMS)[33,34]. Furthermore, paradigms studying visual

anticipation have shown that pre-stimulus alpha oscillations modulate visual responses to stimuli reflecting task-related expectations, and thus likely carry top-down predictions[35–37]. Top-down predictions responsible for the sense of agency may be similarly conveyed by pre-movement neural oscillations, but their mechanistic role remains untested. Here, we tested the hypothesis that pre-movement theta-alpha oscillations influence the sense of agency by conveying top-down predictions used for sensorimotor comparisons.

We started by leveraging data from two experiments (Experiments 1 and 2) previously conducted by our group[13,38], applying further analyses to investigate this specific hypothesis. Experiments 1 and 2 involved the participation of a tetraplegic individual who is a proficient user of an intracranial brain-machine interface (BMI). This BMI system translates motor commands decoded from the M1 into functional hand movements through a neuromuscular electrical stimulation system (NMES, see Fig. 1). In Experiment 1, the participant was asked to explicitly rate his sense of agency for cued hand movements[13]. These previous results focused on the role of M1 in post-movement processing, highlighting that LFP amplitude and multiunit activity in M1 encode exogenous sensory feedback congruency and that these signals covary with agency judgements for BMI actions. Here, we instead focused on the contribution of endogenous pre-movement oscillations to the sense of agency. In Experiment 2, we developed an implicit measure of the sense of agency based on the subjective perception of the timing of self-initiated movements[38]. To extend our investigation to whole-brain dynamics, crucial to test our overarching hypothesis, we collected further data from a cohort of healthy participants in a conceptual replication of Experiment 1 using a BMI based on scalp electroencephalography (EEG). This allowed us to study the interplay between pre-movement neural oscillations, sense of agency and post-movement neural connectivity.

## Results
### Experiment 1 – The phase of pre-movement low alpha oscillations in M1 predicts explicit agency judgements
We hypothesised that, if sensorimotor predictions are conveyed by theta-alpha oscillations, a correlation between agency judgements and

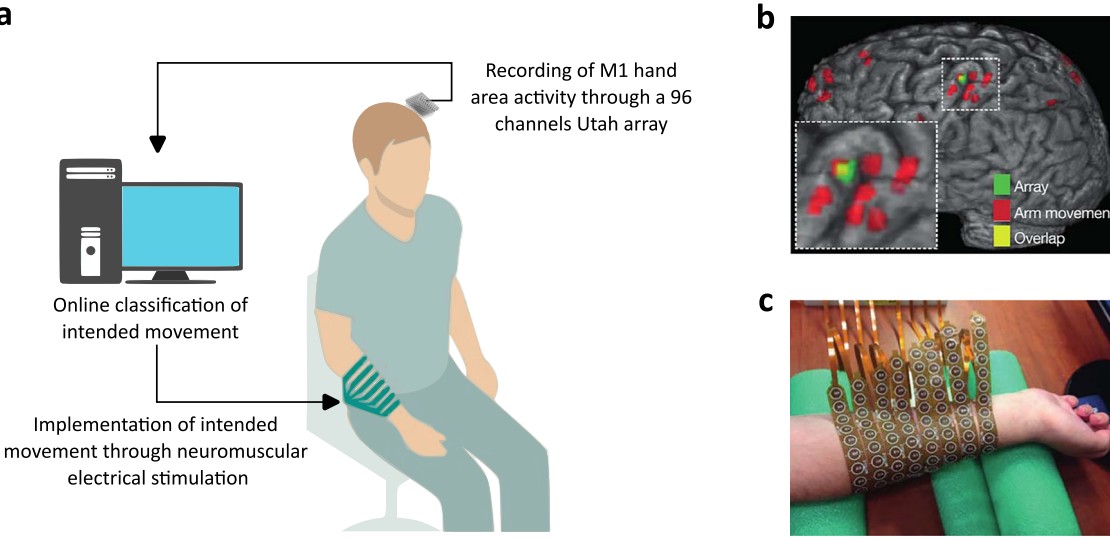

**Fig. 1 | BMI setup for Experiment 1 and 2. a** Neural activity generated as the participant attempted to move his right hand was recorded from the region controlling hand movements in the participant's left motor cortex through a 96 channels Utah array. Motor attempts were decoded by a nonlinear support vector machine based on oscillatory power in the multiunit range (234–3750 Hz) for each channel. The decoded movement was executed through a custom NMES sleeve (panel **c**). Figure 1a was adapted from Serino, A., Bockbrader, M., Bertoni, T. et al. Sense of agency for intracortical brain-machine interfaces. Nat Hum Behav 6,

565–578 (2022). https://doi.org/10.1038/s41562-021-01233-2[13]. **b** fMRI scan showing areas coding for hand movement (red), array position (green) and their overlap (yellow). **c** Custom NMES system fitted on the participant's hand. Figures 1b, c were adapted with the permission of Springer Nature (License Number 5991990373709, https://s100.copyright.com/order/709aa318-3008-4709-aff3-5a7d49b394f5) from Bouton, C., Shaikhouni, A., Annetta, N. et al. Restoring cortical control of functional movement in a human with quadriplegia **533**, 247–250 (2016). https://doi.org/10.1038/nature17435[74].

their power or phase would emerge. We first tested this hypothesis in data previously collected in an expert intracranial BMI user[13].

The participant was instructed to plan and execute one of four possible hand movements (hand closing/opening, thumb flexion/extension) using the BMI prosthesis. Somatosensory feedback was manipulated by producing either the decoded hand movement (congruent feedback, S +) or the opposite hand movement (incongruent feedback, S-; e.g., hand closing instead of opening, and vice versa) through NMES. By the same logic, visual feedback was concurrently manipulated by displaying either the decoded hand movement (V +) or the opposite hand movement (V −) using a virtual hand, superposed to the participant's (hidden) real hand (Fig. 2a). All possible combinations of congruent and incongruent visual and somatosensory feedback (V +/S +, V −/S −, V +/S −, V −/S +) were presented in a randomised order. Following each trial, the participant was asked to provide an agency judgement for the executed movement (Q1: "Was it you who generated the movement? Yes - No").

Congruent (V +/S +) or incongruent (V −/S −) trials consistently elicited positive or negative agency judgements, respectively. In these trials, the congruency of exogenous sensory feedback thus accounted for almost all the variability observed in the data (see Fig. 2b and ref. 13). To emphasise the role of endogenous neural oscillations, we focused our analysis on conflicting feedback conditions (V +/S − and V −/S +), exhibiting a much weaker correlation between agency judgements and sensory feedback (McFadden's $R^2$ in a logistic regression Q1- feedback = 0.02).

To study the hypothesised relationship between agency judgements and the phase of oscillations in the 4–13 Hz range in M1 LFP, we time-locked the LFP to the onset of the hand movement and contrasted the instantaneous phase between trials with positive and negative (Q1 = Yes / No) agency judgements by using the phase opposition product[39]. This measure indexes the amount of clustering of phase angles for high and low agency trials around opposite phases. We found a significant cluster of phase opposition with $p = 0.0004$, corrected for multiple comparisons across time-frequency points (unless otherwise specified, all $p$-values reported are corrected for multiple comparisons as described in the methods). The cluster spanned the 6–9 Hz range and peaked at about 8 Hz, from 500 to 50 ms before movement onset (Fig. 2e). Plotting single-trial phases as in ref. 40, it is apparent that phases within this specific frequency range and pre-movement period were repeatable across trials within the same condition and different across conditions (Fig. 2d). Phase angles at the maximal phase opposition time-frequency point (8 Hz, -256 ms) were clustered between π and π/2 for high agency trials, and between 0 and 3π/2 for low agency trials (Fig. 2f). This phase opposition pattern is visible also in the time course of the trial-averaged LFP, although this measure is not suited for illustrating single-trial phases (Fig. 2c). The relationship between the 8 Hz phase and agency held also at fixed sensory feedback congruency, and was similar in V +/S − and V −/S + trials, with positive agency judgements becoming increasingly frequent as phase angles approached the optimal phase in both conditions (Fig. 2g). In contrast, power in the 4–13 Hz range (up to 40 Hz) showed no significant difference between high and low agency trials (Supplementary Fig. S1a). When analysing higher frequencies (up to 40 Hz), we found no significant phase opposition (Supplementary Fig. S2a).

The observed effect peaked at 8 Hz, at the boundary between conventional theta (4–8 Hz) and alpha (8–13 Hz) frequency bands. However, as shown in Supplementary Fig. S3a, in our implanted participant, the peak of the power spectrum, typically observed in the alpha band[41], was rather low in frequency (6.2 Hz). Movement-related desynchronization, expected in the alpha band (8–13 Hz) around movement onset[42], was also observed in a lower frequency range, peaking at 6 Hz (Supplementary Fig. S3b, c). This suggests that the observed phase opposition occurred within the participant's individual range of the sensorimotor mu rhythm, which is typically

associated with the alpha (8–13 Hz) frequency band[43]. This deviation from the typical frequency range is not surprising, as the mu rhythm has been observed to be lower in frequency in patients with chronic paralysis[44]. Therefore, we hereinafter refer to the frequency range of this effect as "low alpha".

As a complementary analysis, we also investigated the relationship between agency and readiness potentials, the negative deflection of the LFP thought to be a correlate of pre-movement neural activity[45]. We observed a trend of higher agency trials associated with a stronger negative deflection of the readiness potential. However, this effect did not survive correction for multiple comparisons (see Supplementary Fig. S4).

### Experiment 2 – The phase of pre-movement low alpha oscillations correlates with perceived action timing anticipation

The previous analyses establish a relationship between pre-movement low-alpha oscillations and explicit agency judgements. We next tested whether the same phase opposition can distinguish between high vs. low agency actions as defined from an implicit marker of agency based on the subjective perception of the timing of self-initiated movement[32]. Leveraging our BMI setup, in a previous study, we showed that temporal judgements of voluntary actions triggered by the participant's intention to move are anticipated compared with involuntary actions triggered by NMES, resulting in a temporal compression between the intention to move and the action[38].

The experimental paradigm and temporal compression results are extensively reported in our previous work[38]. Below we provide a brief summary of the methods and findings in the subset of conditions relevant to this study. A rotating clock was displayed on a screen, and the participant was asked to report the position of the clock at the onset of a hand movement triggered by the NMES system (Fig. 3a). In the voluntary session, the action was triggered by the participant's intention to move as decoded by the BMI system. In the involuntary session, the movement was randomly generated via the NMES system without motor intention. The participant perceived voluntary BMI-generated movements as occurring earlier relative to their actual timing than involuntary movements (median voluntary = − 497.8 ± 299 ms interquartile range, median involuntary = − 384 ± 185 ms, Wilcoxon $p = 0.033$, Fig. 3b). This effect was specific for actions, as it did not apply to the perceived timing of a sound following the movement, ruling out a generic effect due to the surprise induced by the external induction of the movement (Supplementary Fig. S5). We thus hypothesised that trials showing stronger intention-action temporal compression may be associated with a higher sense of agency, and with the specific oscillatory phase observed in high agency trials in Experiment 1. To test this, we first computed the phase opposition product between trials in which the movement was perceived earlier (high agency) and trials in which the movement was perceived later (low agency).

We found a significant cluster of phase opposition in the 6–10 Hz frequency range ($p = 0.0017$), with the peak occurring at 8 Hz and at − 342 ms before movement onset (Fig. 3e). As depicted in Fig. 3d, f, the 8 Hz phase 256 ms before movement in trials with movement perceived early (late) was qualitatively similar to trials with high (low) explicit agency in Experiment 1. Note that, to facilitate comparisons with Experiment 1 (Fig. 2f) without artificially rotating phase angles, we show phase angles at the time of maximal phase opposition for Experiment 1 (− 256 ms) rather than at -342 ms. As for Experiment 1, no significant difference in 4–13 Hz power was found (Supplementary Fig. S1b). However, analyses at higher frequencies (15–40 Hz) revealed a significant phase opposition cluster at around 30 Hz (Supplementary Fig. S2b), not found in Experiment 1 (Supplementary Fig. S2a). Importantly, the early vs late differences in the pre-movement LFP phase were far stronger and more significant in the pre-movement LFP phase than in either pre- or post-movement LFP amplitude. LFP amplitude discriminated maximally between the two conditions at 544 ms post-movement (minimal $p$-value,

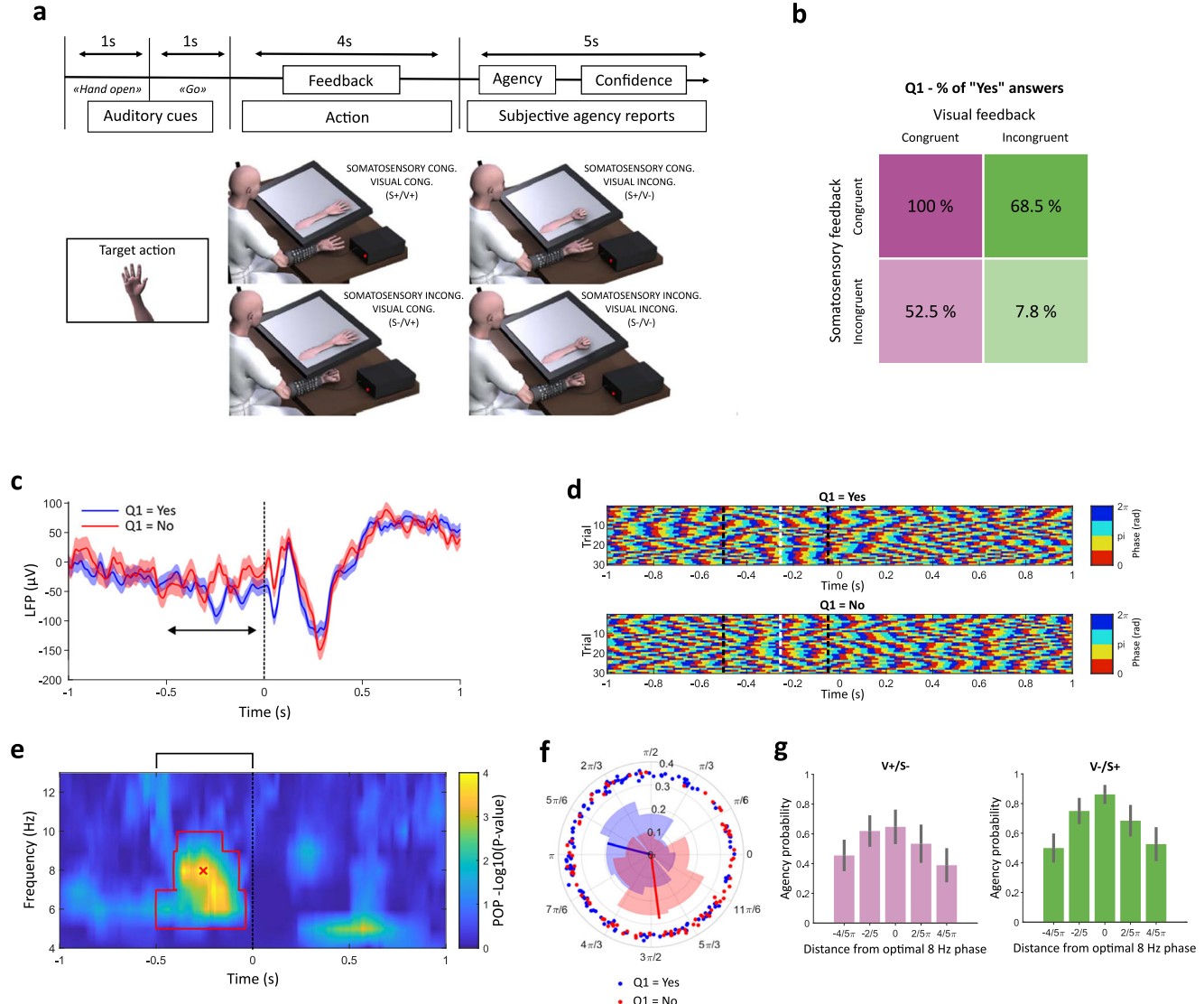

**Fig. 2 | Experiment 1 design and results. a** Timeline and experimental conditions. The participant was cued to perform one of 4 possible hand movements. Once the movement was decoded by the BMI, visual and somatosensory feedback was provided, being either congruent or incongruent with the decoded movement. Figure 2a was adapted from Serino, A., Bockbrader, M., Bertoni, T. et al. Sense of agency for intracortical brain-machine interfaces. Nat Hum Behav 6, 565–578 (2022). https://doi.org/10.1038/s41562-021-01233-2[13]. **b** Percentage of high agency trials as a function of visual and somatosensory feedback. **c** Trial-averaged LFP for high (blue) and low (red) agency trials, time-locked to the onset of the hand movement, showing the pre-movement phase opposition. The black arrow indicates the period of significant phase opposition; shades denote standard errors, and the vertical dashed line the onset of the movement. **d** Instantaneous 8 Hz phase for 30 individual trials with Q1 = Yes (top) and Q1 = No. Black dashed lines indicate the time limits of the significance cluster shown in panel (**e**), and the white dashed lines indicate the timepoint of strongest phase opposition. **e** Uncorrected p-values

(-Log10(p-value)) for the phase opposition product based on a one-sided comparison with 10000 permutations. The red contour delimits the significant cluster after a cluster-based permutation test (corrected p = 0.0004). The black bracket above the plot indicates the time window of interest for cluster-based correction (− 0.5/0 s), and the red cross the time-frequency point of maximal phase opposition (− 256 ms, 8 Hz). **f** Phase angles for individual trials at the time-frequency point of maximal phase opposition. Individual high-agency trials are displayed in blue, and low-agency trials in red. The blue/red vectors indicate the preferred angles for high/low agency, respectively, and their length is proportional to the inter-trial coherence (ITC, see methods). **g** Dependency of agency judgements on the pre-movement 8 Hz phase, separately for V + /S- (left, N = 117) and V-/S + (right, N = 93) trials. The bars indicate the probability ($N_{Yes}/N_{TOT}$) of reporting high agency depending on the distance from the optimal phase of 8 Hz oscillations at − 256 ms. Error bars indicate 66% confidence intervals for the mean of a binomial distribution based on the agency probability and number of trials.

t test, Fig. 3g). Such difference was not significant after cluster correction for multiple comparisons across time points (p > 0.08). Accordingly, phase distributions within the pre-movement phase opposition cluster (− 256 ms, Fig. 3f) had much less overlap than LFP amplitude distributions at the post-movement timepoint of maximal amplitude-based discriminability across early and late movement perception (Fig. 3i).

To rule out that our low alpha phase effect was merely a result of attentional or perceptual processes related to the timing judgement required by the task, we ran the same analysis on the involuntary control

session. Here, the perceptual task is identical, but no voluntary action is required, and thus no agency is expected. We found no association between pre-movement oscillations and perceived action timing in the involuntary control session (Supplementary Fig. S6). We also verified that both the behavioural and the phase effect held for the canonical operant condition of the intentional binding paradigm, when the hand movement is followed by a sound (Supplementary Fig. S7).

As shown in Fig. 4a, trial-averaged LFPs filtered at 8 Hz shared a qualitatively similar phase when comparing high (or low) agency

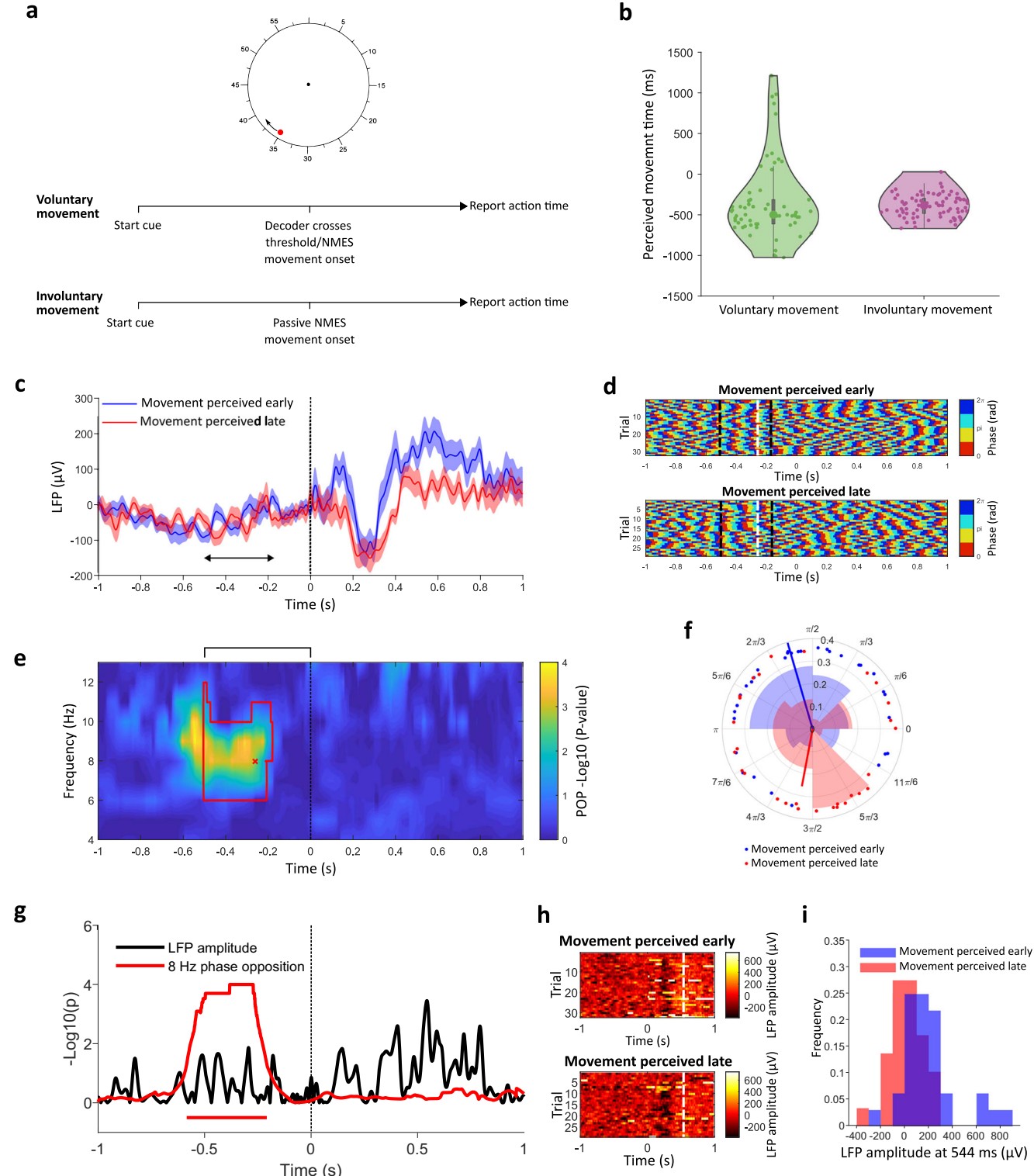

judgements from Experiment 1 (explicit) with early and late action perception trials from Experiment 2 (implicit). This was statistically confirmed by bootstrapping (Fig. 4b, c), and by computing the phase opposition between all trials with high agency ("Yes" answers in Experiment 1 and movements perceived early in Experiment 2) and low agency ("No" answers in Experiment 1 and movements perceived late in Experiment 2) (Fig. 4d, e). Therefore, the same pre-movement 8 Hz oscillatory phase was associated with a higher explicit judgement of agency, and with anticipated action timing perception, which was also observed in voluntary vs. involuntary movements.

## Low alpha LFP oscillations capture modulations of M1 firing

Rhythms recorded with LFPs capture a multitude of neural phenomena, which may not be straightforward to interpret[46]. To better characterise the neural bases of the 8 Hz rhythm reflected in the LFP and modulating the sense of agency, we quantified the relationship between the phase of this rhythm and M1 spiking activity. We focused on data from Experiment 1 due to the larger number of trials per experimental session. We quantified the strength of the relation between the level of firing and the LFP oscillatory phase by computing the LFP-spike phase locking value (PLV, e.g., as in ref. 47) for different

**Fig. 3 | Experiment 2 design and results. a** Timeline. In the voluntary movement session, the participant spontaneously initiated hand movements, realised by NMES upon the decoder crossing threshold. In the involuntary movement session, movements were generated by activating the NMES system at a random time, with no intention by the participant. After each trial, the participant reported the position of a dot rotating on a clock displayed on a screen to indicate his perceived movement timing. **b** Perceived movement timing in voluntary (left, $N = 61$) and involuntary (right, $N = 80$) sessions. Grey boxes contain the central 75% of trials, black dots indicate medians, and whiskers indicate 1.5 interquartile intervals. **c** Trial-averaged LFP for trials with early (blue) and late (red) perception of movement (median split). Shades denote standard errors. **d** Instantaneous pre-movement phase (as in Fig. 2d) for individual early (top) and late (bottom) movement perception trials. Black dashed lines indicate the time limits of the cluster shown in panel (**e**), the white dashed line indicates the timepoint shown in (**f**). **e** Uncorrected log p-values for phase opposition between trials with early and late

movement perception, based on a one-sided comparison with 10000 permutations. The red contour delimits the significant cluster (corrected $p = 0.017$) after the permutation test. The black bracket above the plot indicates the window for cluster-based correction ($-0.5/0$ s). **f** Histograms of individual phase angles for at the time-frequency point of maximal phase opposition in Experiment 1 (-256 ms, 8 Hz) to allow comparison with Fig. 2e. Blue/red lines indicate the preferred angles for high/low implicit agency, respectively, and their length is proportional to the ITC. **g** Comparison of statistical significance (uncorrected log p-values) of phase differences (red, 10000 permutations) and LFP amplitude differences (black, $t$ test). Solid horizontal lines indicate time windows significant after cluster correction for multiple comparisons across timepoints. **h** LFP amplitude (raw, unfiltered signal) for individual early (top) and late (bottom) movement perception trials. The white dashed line indicates the timepoint of maximal dissociation. **i** Histogram of LFP amplitudes at the timepoint of maximal dissociation between early (blue) and late (red) trials.

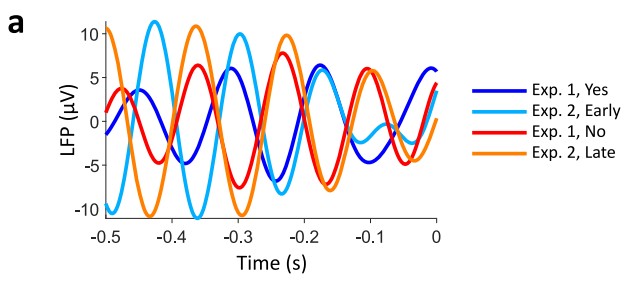

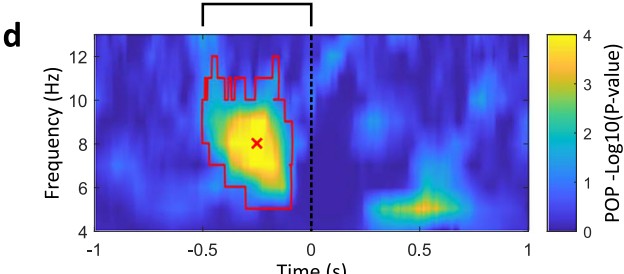

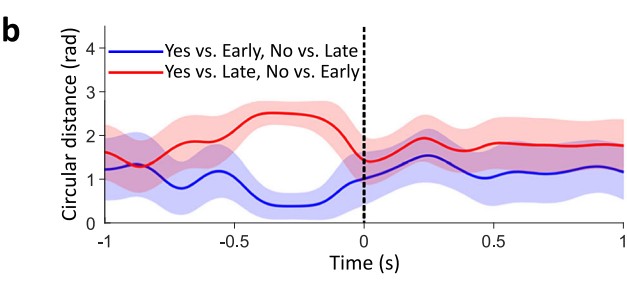

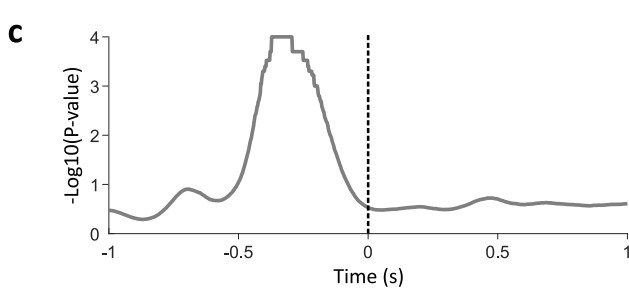

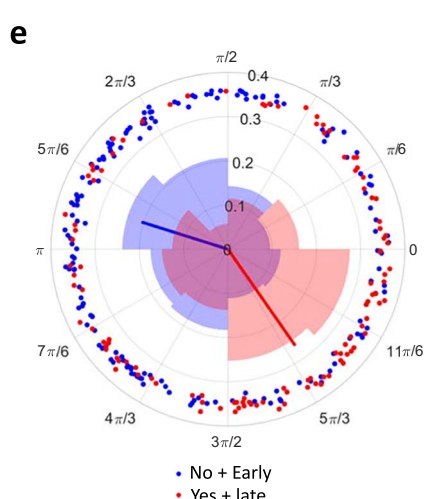

**Fig. 4 | Pre-movement phase and agency across Experiments 1 and 2. a** Trial-averaged LFP, filtered at 8 Hz, for high and low explicit (blue and red curves, respectively) and implicit (cyan and orange curves) agency. **b** The blue curve shows the average circular distance between 8 Hz LFP phases of explicit high and implicit high agency, and between explicit low and implicit low agency trials. The red curve, as a control, shows the average circular distance between explicit high and implicit low agency, and between explicit low and implicit high agency trials. Shaded areas indicate standard errors estimated through bootstrapping. **c** Two-sided statistical comparison, obtained by bootstrapping, of the circular distances (red and blue lines) shown in panel (**b**). The plot shows the probability (uncorrected log p-value), over 10000 resamples, that the blue curve in panel (**b**) is larger than the red curve.

**d** Uncorrected log p-values for the phase opposition product between trials with high and low agency (grouping implicit and explicit assessment), based on a one-sided comparison with 10000 permutations. The red contour indicates a significant cluster (corrected $p = 0.0002$), confirming that trials with high explicit agency can be grouped with trials with early perception of movement, and vice versa. The black bracket above the plot indicates the time window of interest for cluster-based correction ($-0.5/0$ s). **e** Phase histograms at 8 Hz, $-256$ ms (red cross in panel **d**) for explicit and implicit high (blue) and low (red) agency trials. Blue/red vectors indicate the preferred angles for high/low agency, respectively, and their length is proportional to the ITC.

frequencies in the 4–13 Hz range. We found that the LFP-spike PLV peaked at the LFP frequency of 8 Hz (Fig. 5a), indicating that firing activity was most strongly modulated by the phase of 8 Hz oscillations. The firing was about 6% higher when in the most favourable oscillatory

phase of 8 Hz LFP oscillations than when in the least favourable (Fig. 5b). Moreover, the preferred LFP phase angles of individual units exhibited a clear clustering around $4\pi/3$ (Fig. 5c). These results suggest that 8 Hz LFP oscillations recorded in our experiments capture

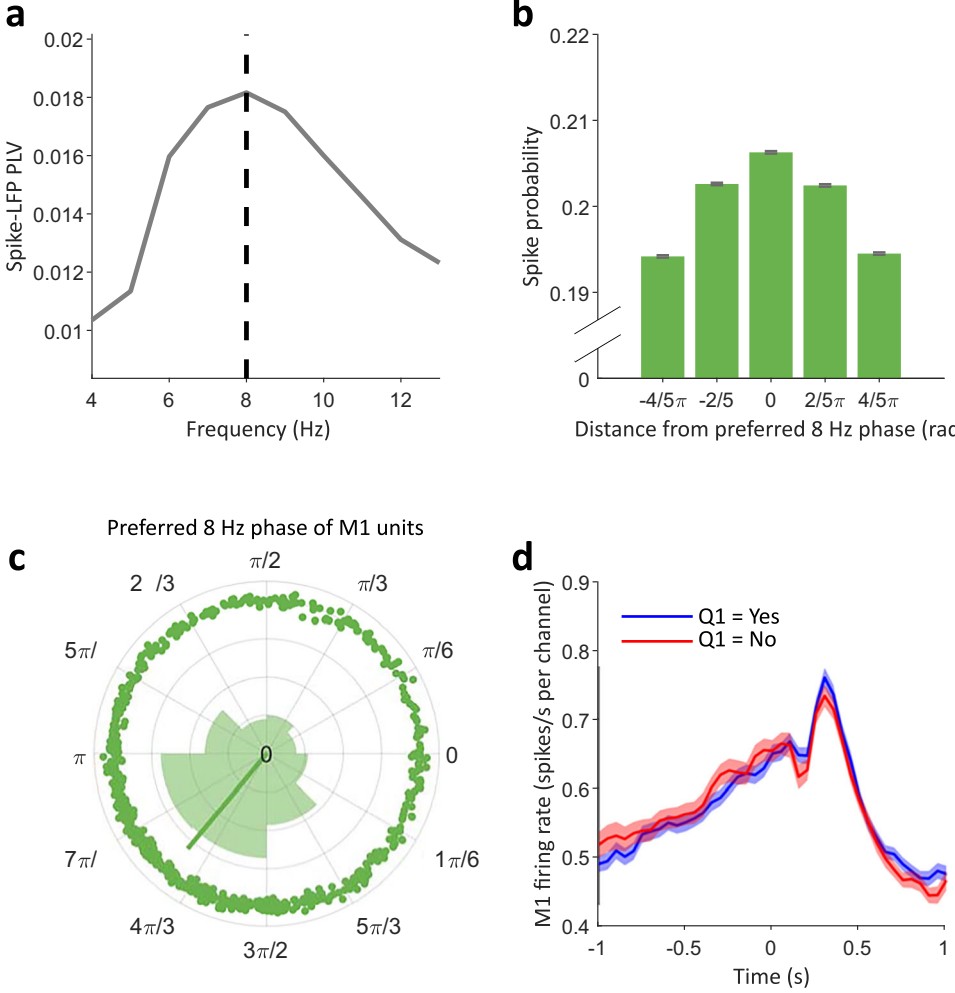

**Fig. 5 | LFP oscillatory phase and M1 spiking activity. a** Phase-locking value between LFP oscillations and pooled spiking activity in M1, as a function of frequency. The phase-locking value is defined as the ITC of phase angles of oscillations at a given frequency, taken at each spike. **b** Histogram of the relative probability that a spike occurs in either of 5 bins of the 8 Hz phase of LFP oscillations ($N = 1.954 \times 106$). In case of no relation between spikes and 8 Hz oscillations, spikes should be distributed equally across the five bins, at $1/5 = 0.2$. The 5 phase bins were defined relatively to the global preferred phase of all units. Error bars indicate 66% confidence intervals for the mean of a binomial distribution based on the agency probability and number of trials. **c** Distribution of the preferred 8 Hz phase angle for spiking across 665 units, defined as the sum of phase vectors of 8 Hz oscillations taken at each spike. Error bars denote a 66% confidence interval on the spike count, assuming a binomial distribution. **d** Trial-averaged firing rate across all units (V + / S − and V −/S + trials), split by agency judgement. Shades denote standard errors.

periodic fluctuations in M1 firing activity. To rule out that the overall level of firing per se, rather than its periodic component, influenced the sense of agency, we assessed whether the pre-movement global M1 firing rate was associated to agency judgements. As shown in Fig. 5d, this analysis revealed no significant difference in firing rates between high and low agency trials. These results suggest that the sense of agency is predicted by the specific 8 Hz periodic component of firing fluctuations, rather than the average firing rate itself. Moreover, LFP-spike PLV values did not differ between trials of high vs. low agency (Supplementary Fig. S8), suggesting that only the oscillatory 8 Hz phase, but not the amount of coupling between such phase and spiking activity, covaries with agency ratings.

**Experiments 3 & 4 – pre-movement SMA and M1 alpha oscillations predict agency ratings in healthy participants**
The above results establish a relationship between pre-movement 8 Hz oscillations in M1 - the only recording site in our implanted participant - and sense of agency. To investigate the potential contribution of areas beyond M1, in Experiment 3, we devised an EEG-based version of Experiment 1, which we believe to be the closest conceptual extension of its paradigm achievable in healthy participants. Thirty healthy

participants were trained to use an EEG-BMI based on kinaesthetic motor imagery to trigger the movement (hand closing) of an anatomically congruent virtual hand on a screen. After each movement, they rated their sense of agency for the movement on a scale from 1 to 9 (Fig. 6a, b). After verifying the validity of our setup through preliminary behavioural analyses (see Methods and Supplementary Fig. S9), we examined whether the phase of pre-movement sensorimotor oscillations was associated with agency ratings. To localise the source of neural oscillations, we projected EEG activity to 114 cortical regions of interest (ROIs) through eLORETA[48]. For each ROI, we then contrasted the highest and lowest 33% of agency ratings via the phase opposition product in the alpha (8–13 Hz) range in the 0.5 sec preceding the movement. The alpha range was chosen to match the overall higher spectral peak in healthy participants compared to our implanted participant (see Experiment 1 and Supplementary Fig. S3). The cortical map of p-values for the alpha range phase opposition product based on agency ratings is shown in Fig. 6c. A cluster of two regions survived multiple comparison corrections across all 114 ROIs ($p = 0.04$). These regions correspond to the posterior part of the left supplementary motor area (SMA), showing the strongest effect with uncorrected $p = 0.0002$, and the left M1 (uncorrected $p = 0.003$), reproducing our

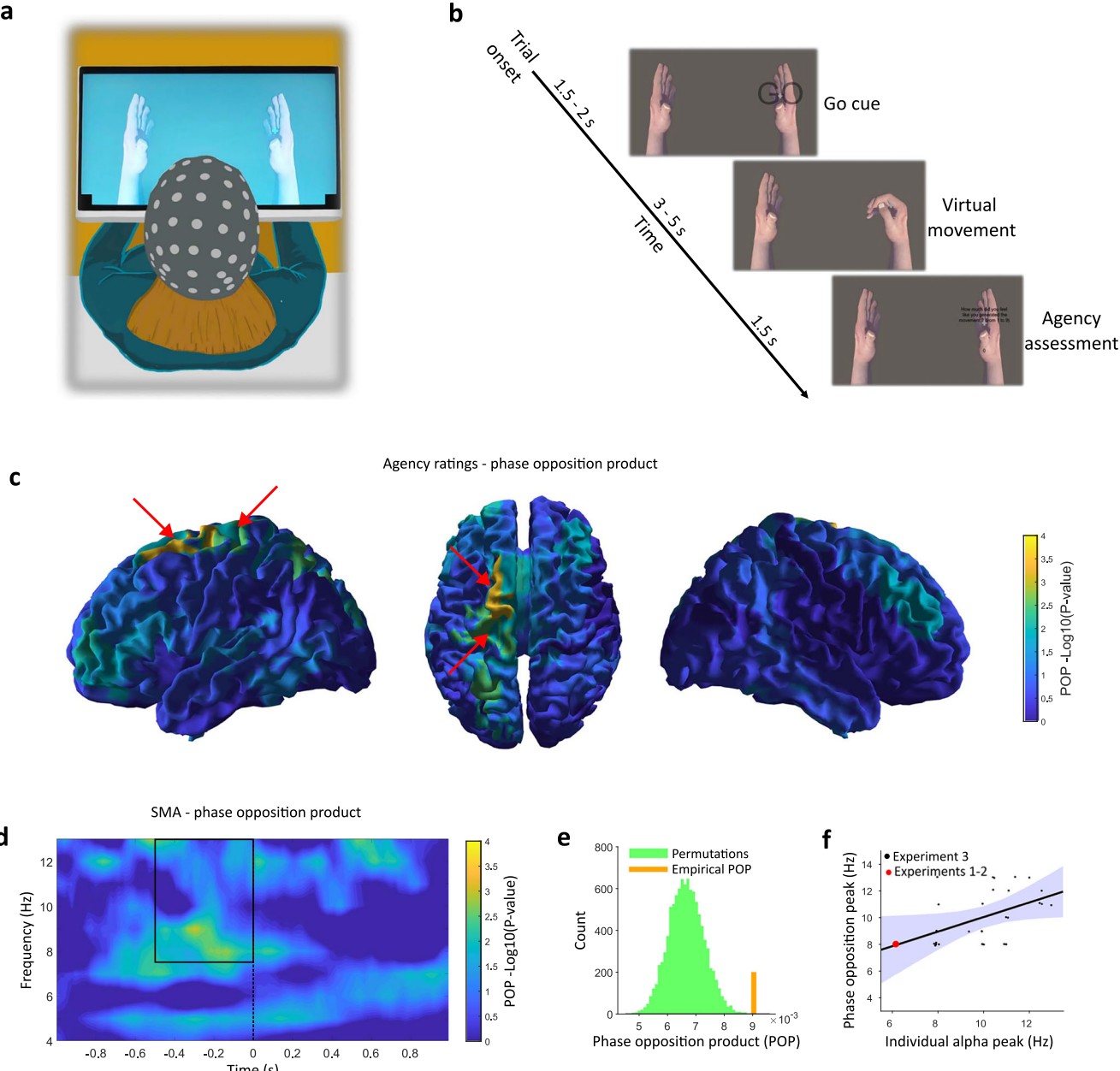

**Fig. 6 | Experiment 3 design and results. a** Experimental setup of the non-invasive BMI paradigm, whereby healthy participants used motor imagery to control a virtual hand and provide agency ratings. **b** Timeline of a cued BMI trial, constituting the core of the experiment. These trials were two-thirds of the total and used sham BMI without participants being aware of it, to minimise variability related to decoder performance. One-third of the trials (not shown) were self-paced and used actual BMI. They were only used to keep participants convinced that they were in control of the virtual hand. In total, 5 blocks of 60 trials were collected. **c** Uncorrected log p-values for the phase opposition product in the − 0.5/0 s, 8–13 Hz range, computed over 114 ROIs after eLORETA source reconstruction via one-sided comparison with 10000 permutations. Red arrows point at the location of the cluster of two ROIs surviving whole-brain cluster correction, in the left contralateral SMA and M1. **d** Time-frequency plot of the phase opposition product (negative log p-value from one-sided comparison with 10000 permutations) for the most significant ROI, corresponding to the posterior part of the left-contralateral SMA. The black rectangle denotes the selected time and frequency range, the dashed line the onset of sensory feedback. **e** Empirically measured Phase Opposition Product (8–13 Hz, − 0.5/0 s) for the left SMA compared to 10000 permutations with shuffled agency ratings. **f** Correlation between the individual alpha peak and the frequency of strongest phase opposition in the left SMA. The shaded area indicates the 95% confidence interval for the regression line. Data of the tetraplegic participant from Experiments 1-2 (not included in the regression) is shown for comparison (red dot).

results in the implanted participant. SMA effects were robust to the specific choice of the frequency range (Supplementary Fig. S10a, b). In addition, no significant difference in power (Supplementary Fig. S1c, d) phase at higher frequencies (Supplementary Fig. S2c, d) or readiness potentials (Supplementary Fig. S4) was found.

The SMA effect peaked at 9 Hz, close to what was observed in our implanted participant, but was relatively spread across the whole alpha

band (Fig. 6d). Since alpha-band peak frequencies vary across individuals[49] and correlations of alpha-band activity with behaviour are stronger at frequencies closer to the individual alpha peak[50], we predicted that individual variations in agency-related phase opposition might reflect individual variations in alpha peak frequency. Confirming this prediction, the frequency at which maximal SMA phase opposition was found for each subject correlated with their individual SMA alpha

band frequency of maximal power ($R = 0.46$, $p = 0.011$, Fig. 6f). This suggests that individual variations in the frequency at which the phase better predicts agency depends on individuals' idiosyncratic alpha band peak.

To further confirm that pre-movement alpha oscillations discriminate high-agency and low-agency actions, we analysed data from an independent cohort of 10 participants, who performed a classic agency judgement paradigm. Briefly, participants were asked to freely lift their index finger while receiving congruent visual feedback from a virtual hand, superimposed on their own. Visual feedback was delivered at various temporal delays from their actual movement. At the end of each trial, participants were asked to report whether they felt agency or not for the virtual hand. Comparing trials with "yes" vs. "no" agency reports, we identified the same phase opposition in alpha oscillations in M1 and SMA, thus confirming and further generalising our results to a different experimental paradigm (Experiment 4, see methods and Supplementary Fig. S11).

### The optimal phase for agency is associated with increased alpha-band functional connectivity

Results from our four experiments showed that the pre-movement oscillatory state of motor areas is associated with the subjective sense of agency for a subsequent movement. Since agency is reported (and most likely experienced) post-movement, we searched for a trace of the pre-movement oscillatory phase in post-movement signals, possibly affecting the sense of agency. In line with theories about brain rhythms and communication[28], previous studies[31] have highlighted correlations between the pre-stimulus phase in a given brain area and post-stimulus connectivity originating from that area. Thus, we searched for an association between the pre-movement SMA oscillatory phase (the region showing the strongest phase effect) and post (and during) movement functional connectivity between SMA and the rest of the brain, as a putative source of modulation of subjective agency.

For each participant from Experiment 3, we extracted the left, contralateral SMA phase at the time-frequency point in which the modulation of the sense of agency was strongest. We then selected trials in which the pre-movement oscillatory phase was close to the optimal phase for agency, and trials in which it was far from it (see Methods for details). We contrasted between these subsets of trials functional connectivity in the 4-45 Hz range, measured through the debiased weighted phase-lag index, WPLI[51]. Specifically, we studied the link between pre-movement SMA oscillatory phase and post-movement (0.2–1.2 s; see "Methods") connectivity between SMA and the rest of the brain. When computing a global average of functional connections between SMA and all other cortical regions, we found that the optimal oscillatory phase was associated with an increase in connectivity in a cluster spanning the 9–12 Hz range ($p = 0.005$, Fig. 7a, b). To localise the source of this effect while reducing the degrees of freedom of our analysis, we analysed the 9–12 Hz connectivity from SMA after grouping brain regions by lobe and hemisphere, obtaining eight macro-regions (see "Methods"). Left (contralateral to the movement) frontal, temporal, and parietal areas survived Bonferroni correction ($t(24) = 3.47$, $3.41$, and $2.99$; uncorrected $p = 0.002$, $0.0023$, and $0.0063$, respectively), confirming that the optimal oscillatory phase for agency was associated with a widespread increase in connectivity from the left SMA (Fig. 7c). The same analysis using the left M1 as a seed revealed a similar pattern of connectivity changes (Supplementary Fig. S12). When performing the analysis at a finer spatial scale with 114 ROIs, three clusters survived multiple comparisons correction. The first cluster (eight regions, $p = 0.0046$) spanned the middle prefrontal cortex and anterior cingulate, the second (three regions, $p = 0.046$) was located in the posterior parietal cortex, the third (six regions, $p = 0.01$) was located in the temporal cortex (Fig. 7d).

Finally, we tested whether the observed connectivity changes were associated with a change in the directionality of functional connectivity. To do this, we studied the 9–12 Hz phase coherence of left SMA signals with time-shifted signals obtained from each of the three significant target regions (using the methodology set in refs. 52,53). The time shift at which the coherence is stronger indicates the direction in which functional connectivity is stronger. Stronger coherence for positive time shifts would support a stronger correlation of left SMA oscillatory activity with future oscillatory activity of the target region (i.e., from SMA). Conversely, the stronger correlation for negative time shift would support a stronger correlation with past oscillatory activity of the target region (i.e., from the target region). We found that values of phase coherence were higher at positive time shifts in trials with optimal phase, whereas coherence was stronger at negative time shifts in trials with non-optimal phase, (Fig. 7e). Time shifts of peak coherence were significantly different between optimal and non-optimal phase trials ($t(24) = 3.02$, $p = 0.0059$, Fig. 7f). This suggests that functional connectivity in trials with the optimal phase for agency was enhanced more in the direction from SMA to temporal areas, compared to trials with non-optimal phase. To better localise this effect, we performed the same directionality analysis at a finer spatial resolution only within the temporal lobe (the only region showing a significant effect at the coarser spatial resolution), at the scale of the original 114 ROIs used for source reconstruction. We found the region with the most significant ($t(24) = 3.14$, $p = 0.0022$, one-tailed) directionality shift, again from SMA to the target area, to be localised in the posterior part of the temporal lobe (Fig. 7g).

## Discussion

We investigated the relationship between neural oscillations and the sense of agency for hand movements through invasive (Experiments 1 and 2, one tetraplegic participant) and non-invasive BMI (Experiment 3, thirty healthy participants, Experiment 4, ten healthy participants) experiments. BMIs offer an exceptional setting for studying the sense of agency, as they allow perturbing the coherence between intentions and actions. Thus, BMIs permit the introduction of nuances in the sense of agency for self-generated movements, which is typically very high and hard to manipulate under normal circumstances. In Experiment 1, we decoded motor intentions from an expert user of an implanted BMI and manipulated the visual and somatosensory congruency between intended actions and sensory feedback via virtual reality and NMES. The participant's agency judgements were predicted by the phase of low alpha (8 Hz) LFP oscillations, ~ 250 ms prior to the movement (Fig. 2c–e).

Although most studies concur that both pre- and post-movement signals contribute to the sense of agency[21,54], the exact processes underlying the integration of sensorimotor predictions and sensory feedback are not yet understood. Seminal studies showed that intentional movements are preceded by a slow negative deflection in scalp potentials above motor areas[45] and that such potential precedes the reported timing of the conscious intention to move[24] (also see ref. 25 for a study focusing on stochastic fluctuations in these potentials). During this period, as motor intention builds up, sensory predictions are likely formulated. However, these studies focused solely on the efferent aspect of the "intentional chain", from intention to action, without exploring its relationship with sensory reafference. Here, we showed that endogenous oscillatory activity before sensory feedback about the movement becomes available predicts the sense of agency for the movement. In our previous work[13] (including data from Experiment 1), we showed how post-movement LFPs and multiunit activity in M1 encode congruency between motor commands and sensory feedback, a key aspect in the sensorimotor comparisons underlying the sense of agency. Our findings in Experiment 1 extend these previous results, suggesting that M1 and SMA also play pivotal roles in the sense of agency at an earlier stage when sensorimotor predictions are computed during motor preparation and before execution. Our results do not rule out the contribution of post-

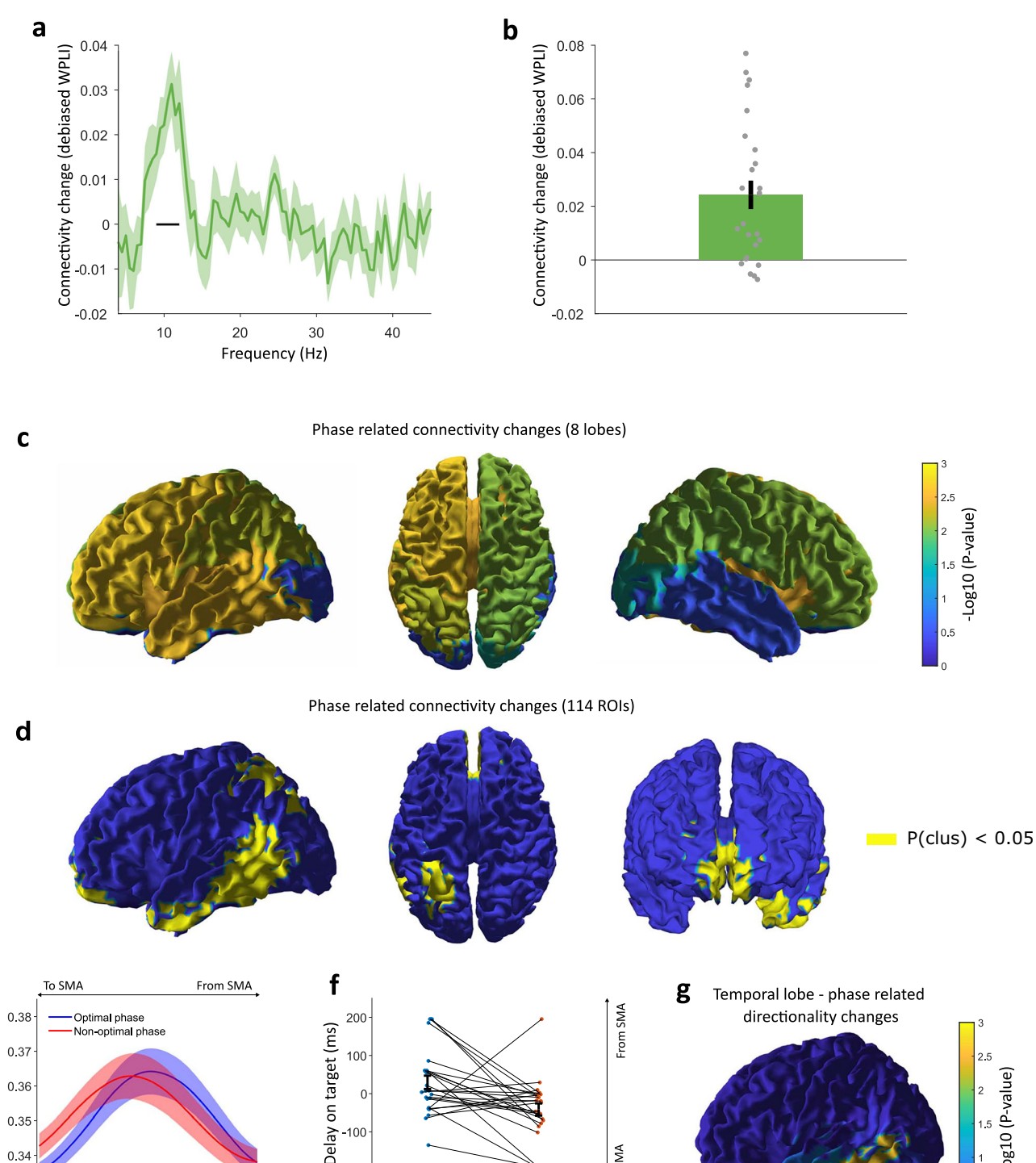

movement signals to the sense of agency. These may be relevant for postdictive inference of causality (apparent mental causation theory[7]), or for sensorimotor comparisons, whereby sensorimotor oscillations may be integrated with higher level cognitive cues to determine the final experience of agency (cue integration theory[6]). This contribution may be reflected in the dissociation of post-movement LFP between high and low agency trials, which was here observed in Experiment 2 and extensively investigated in ref. 13. Pre-movement sensorimotor oscillations may also serve as a trigger for apparent mental causation, and gauge the integration between low- and high-level cues in determining the sense of agency.

In Experiment 2, we corroborated the role of the pre-movement alpha phase using an implicit measure of agency based on the perceived timing of BMI movements in a Libet-like paradigm. The 8 Hz phase preceding movement onset predicted the anticipation of perceived movement timing (binding) that characterised voluntary BMI actions, as compared with involuntary movement, (Fig. 3c–e), thus used here as an implicit proxy of agency. Importantly, the optimal phase for explicitly (Experiment 1) and implicitly (Experiment 2) assessed sense of agency was the same (Fig. 4). Furthermore, the implicit nature of our measure suggests that the effect relates to the genuine pre-reflexive experience of agency, rather than to

**Fig. 7 | Pre-movement phase and connectivity. a** Average functional connectivity changes from the left SMA to all other cortical regions, computed in the 0.2–1.2 sec window from movement onset, when contrasting trials in which the movement started in the optimal vs. non-optimal alpha phase. Shades indicate standard errors. The black line indicates the cluster of frequencies (9–12 Hz) in which significant changes occur after multiple comparisons correction. **b** Connectivity changes in the significant cluster's frequency range (9–12 Hz) for individual subjects ($N = 25$). The vertical bar indicates the standard error. **c** Cortical maps of uncorrected log $p$-values ($t$ test, two-sided) for the connectivity changes in the 9–12 Hz range induced by the pre-movement alpha phase on 8 cerebral macro-regions, using the left SMA as a seed. **d** Cortical map of regions surviving cluster-based correction across ROIs when the same connectivity analysis is performed at the level of 114 ROIs, based on a two-sided $t$ test. **e** Directionality changes in connectivity from the left SMA to left temporal regions depending on the pre-movement alpha phase. The $y$-axis

represents the 9–12 Hz phase coherence between signals from the seed region and delayed signals from the left temporal regions. The result is plotted as a function of the delay on the target region so that more coherence at positive delays suggests connectivity from the left SMA to left temporal regions, and vice versa. The blue curve is obtained from trials starting in the optimal phase, and the red curve from trials starting in the non-optimal phase. Shades indicate standard errors. **f** Results of the directionality analysis at single subject level ($N = 25$). The dots indicate the peak of the individual delay-coherence curves (averages are shown in panel (**e**)) Again, positive values indicate connectivity mainly from the left SMA to temporal regions, and vice versa. Black error bars indicate standard errors. **g** Uncorrected log $p$-values for the directionality analysis from a two-sided $t$ test, as in panel (**f**), performed only within the temporal lobe at the finer spatial scale of 114 ROIs. Dark blue areas indicate ROIs outside the temporal lobe, not analysed at this parcellation level.

metacognitive aspects of explicit judgements. Importantly, the pre-movement phase biased the implicit agency measure for voluntary movements, but not for involuntary ones. This suggests that the activation of the intentional chain is necessary for sensorimotor oscillations to modulate the temporal binding between intention and action. Incidentally, this also rules out any explanation in terms of attentional or perceptual effects, as these would have equally affected time perception for involuntary movements.

When investigating the link between LFP oscillations and spiking activity, we found an 8 Hz peak in coupling between LFP and M1 spiking activity (Fig. 5a), in line with studies describing alpha-band LFP-spike coupling in the non-human primate sensorimotor system[55]. This result offers an intriguing interpretation for the pre-movement phase effect. Studies in humans show that motor-evoked potentials induced by a TMS pulse are modulated by the phase of sensorimotor oscillations[34]. It is possible that also bursts of M1 activity that trigger the onset of spontaneous movements are more likely to occur during the most excitable phase of M1 LFPs. Then, BMI-generated movements occurring during the excitatory LFP phase may lead to higher agency because natural self-generated movements are more likely to happen at that time. This may underlie the association between the pre-movement phase and sense of agency, which we consistently observed in Experiments 1, 2.

In Experiment 3, we extended our investigation to healthy participants, using EEG to study the interplay between the pre-movement phase, sense of agency and whole-brain neural dynamics. Once again, the phase of alpha-band pre-movement oscillations predicted agency ratings. We traced the most significant correlation between the pre-movement phase and agency ratings to the left (contralateral to the movement) SMA and M1 (Fig. 6c). Interestingly, in healthy participants, the phase opposition effect was stronger in SMA than in M1. Since the invasive implant was limited to M1, we cannot exclude that a stronger phase opposition in SMA was also present in our tetraplegic participant but not directly observed. Several studies highlighted SMA as a key region for agency[56] and as one of the nodes showing the earliest activations in the generation of intentional movements[16]. Such early pre-movement activations suggest that SMA may specifically contribute to the predictive component of the sense of agency, an idea supported by neuroimaging[57] and TMS[58] studies. Here, we add to these findings by linking SMA contribution to the sense of agency to a specific alpha-band oscillatory mechanism. Experiment 3 employed motor imagery rather than genuine motor attempts, as in Experiment 1. Given that motor imagery is known to induce comparably higher activations in SMA than in M1[59], a possibility to be addressed in future work, is that the relative contributions of SMA and M1 vary depending on the modality of BMI control (e.g., imagery versus execution).

We ran functional connectivity analyses to study the interplay between pre-movement oscillations in motor areas (i.e., the source of the phase-related agency modulation) and whole brain dynamics. These analyses revealed that BMI movements starting in the optimal

phase for agency were associated with a higher alpha-band connectivity from SMA to the rest of the brain after movement onset (Fig. 7a, b). Specifically, changes in functional connectivity occurred between SMA and part of the posterior parietal cortex, a region which has been classically associated with visual guidance of movements[60,61] (Fig. 7d). Further connectivity changes were observed between SMA and the medial prefrontal cortex and anterior cingulate, two regions implicated in action selection and initiation[21]. These results point at SMA as a key hub in a network of regions classically associated with the sense of agency and are in line with the hypothesis[21] that a sense of agency may emerge from neural connectivity within such a network. Additionally, the optimal phase for the agency was also associated with a change in the directionality of functional connectivity between SMA and the posterior part of the temporal lobe (Fig. 7g), which has been implicated in the visual processing of hand movements[62,63]. It is tempting to speculate that the pre-movement SMA phase may gate the information exchange involved in sensorimotor comparisons between motor intentions (encoded in SMA and pre-frontal areas) and visual information about their outcome, which is used for motor control (encoded in the parietal and temporal lobe). This may, in turn, modulate the amount of binding between intentions and actions and the associated measures of agency. Previous studies have cast the parietal cortex[64,65] as a "sensorimotor comparator", while other work has pointed at the cerebellum[66] and the premotor cortex[67]. Our results possibly encompass these findings by suggesting a distributed architecture, wherein sensorimotor comparisons are performed through alpha-band communication between SMA and temporal-parietal regions.

Previous research has implicated alpha-band communication in the top-down, anticipatory modulation of sensory areas[36,37,68]. This is believed to contribute to sensory processing, e.g., by enhancing relevant stimuli[68] or optimising detection performance[69]. Transposing this idea to the intentional chain, we propose that during the preparatory phase preceding a movement, premotor and motor areas, such as SMA and M1, send predictive signals through alpha-band connectivity to temporal and parietal sensory areas. These predictive signals likely carry information about the expected sensory consequences of the impending action, contributing not only to anticipating sensorimotor comparisons but also to pre-select relevant sensory features for closed-loop motor control (see for example[11,70]). The sense of agency could then be seen as the subjective correlate of selective communication between motor and sensory areas encoding features that are self-generated and should thus be integrated with efferent commands to improve motor performance[11]. Under this perspective, the functional relevance of the neural process leading to the sense of agency becomes apparent. Using invasive BMIs, we highlighted the key role of alpha-band sensorimotor oscillations in this process and generalised this finding to more broadly implementable non-invasive BMIs or other forms of human-computer interactions. These findings may become even more relevant as decoding and actuation technologies

advance to incorporate cognitive aspects of motor control for efficient and intuitive use.

Nonetheless, without direct experimental intervention, the current evidence remains correlational. Further studies are needed to causally assess the contribution of the pre-movement phase to the sense of agency, both when measured explicitly and implicitly. Our unique BMI set-up enabled us to relate pre-movement phase opposition to the temporal compression between intention and action observed for voluntary actions. More studies are needed to validate this intention-action compression as an implicit marker of agency. Furthermore, differences in techniques and experimental paradigms applied in one implanted participant and healthy controls make it difficult to infer whether comparable phase opposition effects observed at different frequencies across different experiments are related to the same or different neural mechanisms. Because of this, and of the general difficulties in imputing neural phenomena to frequency bands[46], here we refrained from making such inference and unbiasedly reported the precise frequency of the observed effects in each subject and experimental paradigm. We could, however, gain some intuition about the individual variability of effects within Experiment 3. Namely, individual differences in the peak phase opposition frequency could be accounted for by individual variations in the individual alpha peak (Fig. 6f). Compatibly with other reports in chronic paralysis[44], the individual alpha peak in the implanted participant was lower than in healthy participants, possibly explaining the lower frequency of the phase opposition effect observed in Experiment 1 and 2.

By detecting sensorimotor contingencies, humans can differentiate between self and externally generated events, and the sense of agency is the experiential counterpart of such a process. It has been argued that such mechanism underlies personal responsibility[71] and even self-awareness, as the self would emerge as the agent of internally generated mental states[1,72]. Here, we provide evidence that alpha oscillations are related to the sense of agency, bolstering functional theories on the role of the alpha rhythm for brain connectivity, binding, and prediction[28,31,36,73]. These findings pave the way to further mechanistic models of key cognitive processes, by applying oscillatory theories of brain communication [28] to the causal binding between internal states and their consequences.

## Methods

### Experiment 1 and 2 - participant
The participant was a 27-year-old (at the time of recordings) male with quadriplegia at the C5/C6 level originating from a cervical spinal cord injury (SCI) dating to 8 years prior to data collection. He had a full range of motion in both shoulders and elbow flexion and could perform twitches of wrist extension (1/5 and 2/5 strength on the left and right wrists, respectively). He had no motor function below C6. His proprioception was intact in the right upper limb/shoulder for internal through external rotation, forearm pronation through supination, and wrist flexion through extension. Proprioception at the level of metacarpal-phalangeal joints for all right-hand digits was impaired. The BMI system required the implantation of a Utah microelectrode array (96 channels, 4.4 × 4.2 mm, 1.5 mm depth) in the hand region of the left primary motor cortex. Reference wires were placed subdurally. The target region was identified via pre-operative functional Magnetic Resonance Imaging as the patient was asked to attempt performing right-hand movements. See the first description of the BMI system[74] for further details about the participant and surgical process.

The participant was enrolled in a pilot clinical trial (NCT01997125, Date: November 22, 2013) of a custom BMI system (Battelle Memorial Institute) to restore motor functionality of the upper limb following SCI. Approval for this study was obtained from the US Food and Drug Administration (Investigational Device Exemption) and the Ohio State University Medical Centre Institutional Review Board (Columbus, Ohio). The participant completed an informed consent process before taking part in the study. He also provided written permission for photographs and video.

### Experiment 1 and 2 - BMI system
The BMI system consisted of a 96-channels Utah array (Blackrock Microsystems) acquiring M1 signals, a standard desktop computer decoding the intended movement from M1 activity, and an electrode patch stimulating right forearm muscles to translate decoded movements into functional hand movements. To account for natural changes in the signal from the Utah array, the decoder was re-trained before each experimental session. Training data was generated by asking the participant to attempt performing one of four hand movements (hand open, HO, hand close, HC, thumb extension, TE, thumb flexion, TF). Clearly, due to chronic paralysis, these motor attempts did not lead to actual hand movements as long as electrical stimulation was off but modulated M1 firing rates in movement-specific patterns, which could be detected by the BMI algorithm. In each training session, the subject performed 7 blocks consisting of 3 repetitions per movement type each.

Neural data from the Utah array was sampled at 30 kHz and band-pass filtered between 0.3 Hz and 7.5 kHz at the hardware level (3rd order Butterworth). The data were digitised in 100 ms bins and analysed through custom MATLAB code. Before decoding, artefacts due to NMES were removed by blanking the signal over 3.5 ms around the artefact, defined as a signal amplitude exceeding 500 μV in at least 4 out of 12 randomly selected channels. Neural decoding was based on a non-linear Support Vector Machine[75] (SVM) recognising patterns of M1 firing activity corresponding to each of the four possible hand movements. The SVM used 96 input features consisting of the mean wavelet power (MWP) for each channel and 100 ms bin. To obtain the MWPs, neural activity was decomposed into 11 wavelet scales (Daubechies wavelet, MATLAB), and the coefficients of wavelets 3-6, corresponding to the multi-unit frequency band spanning from 235 to 3.75 kHz, were averaged for each channel. Thus, the decoder's input features were closely related to high-frequency power at each channel, a robust and computationally non-intensive proxy of multi-unit activity. The decoding system achieved more than 90% accuracy for all four movements (see ref. 74).

At the end of each 100 ms acquisition bin, the decoder analysed neural signals and provided four numbers in the −1/1 range, indicating the decoded relative probability for each of the four movements.

Again every 100 ms, the output of the decoder was further smoothed over a 500 ms time window to determine whether and which movement to implement. A movement was generated if any of the four outputs exceeded the threshold of 0, with the movement with the highest score prevailing if two or more classes exceeded the threshold. During experimental sessions, the participant had to attempt to perform the intended hand movement in order to control the BMI system, as he did during training sessions. Due to decoder output smoothing, plus neural and acquisition noise, the delay from the go cue to decoder threshold crossing was variable and significantly larger (1.2 s ± 0.48 SD) than in natural movements. Being an expert user of the BMI system, the participant is accustomed to these long delays, allowing him to experience a strong sense of agency for BMI movements.

A custom-built Neuromuscular Electrical Stimulation (NMES) system was used to translate the decoded intentions into actual hand movements, by stimulating forearm muscles to elicit the decoded movement. The NMES system consisted of a circumferential forearm sleeve with 130 copper-coated electrodes, 12 mm in diameter. The electrodes were disposed in an array, spaced at regular intervals (22 mm longitudinally x 15 mm transversely). Stimulation was delivered through rectangular pulses of 50 Hz monophasic current (pulse width 500 μs, amplitude 0–20 mA). The stimulation patterns and

intensity were re-calibrated at the beginning of each session in order to optimise the match with the participant's intentions. Due to hardware delays, the onset of NMES stimulation followed the end of the 100 ms acquisition bin in which the neural decoder crossed the threshold by a stereotyped $70 \pm 10$ ms (SD) delay. Further details about the neural decoder and NMES system can be found in ref. 74.

## Experiment 1 – protocol

In Experiment 1, we manipulated the congruency between the participant's motor intentions and sensory feedback and assessed how this affected his sense of agency. Each trial started with a verbal cue about the hand movement to be performed (HO, HC, TE, TF), followed after a 2 s delay by a verbal go cue. The participant was instructed to start attempting the cued movement when the go cue appeared without anticipating. During the 4 s following the go cue, the BMI algorithm decoded changes in M1 multiunit activity generated by the participant as he attempted the cued movement, and translated them into visual and somatosensory feedback according to the decoded movement and the feedback congruency assigned for that trial and sensory modality. Somatosensory feedback was delivered by eliciting the target movement through the NMES sleeve and thus consisted in a functional hand movement. Visual feedback was constituted by an animation of a virtual hand performing the target movement, displayed on a screen placed horizontally to cover the participant's right hand. Note that the participant has sufficient residual proprioception to recognise the hand movement performed even with his real hand being hidden by the screen (see refs. 13,74). The hand model and the animation corresponded to the ones routinely used by the participant during BMI training sessions, and its size and position were adjusted to match the participant's real hand. In trials with congruent somatosensory (and/or visual) feedback, the decoded movement was executed through NMES (or displayed in a virtual animation). In incongruent trials, the opposite movement was executed and/or displayed, replacing HO with HC, TE with TF, and vice versa. Sensory feedback was only delivered when one of the output classes of the neural decoder reached the threshold of 0. In the 5-6 s after the sensory feedback phase, the participant answered two questions, Q1 ("Are you the one who generated the movement?") by saying "Yes" or "No", and Q2 ("How confident are you?") by reporting a number between 0 and 100. Only reports from Q1 are the object of the present analyses. The whole experiment consisted of five experimental sessions performed over different days, each consisting of four blocks of BMI decoder training and four blocks of experiment, each lasting around 15 min. Each experimental block consisted of 32 trials, where each combination of V/S feedback and cued movement was repeated twice. Therefore, the grand total of trials was 640, 160 for each feedback condition.

## Experiment 2 – protocol

Experiment 2 consisted in two sessions. In the first, voluntary movement session (high agency), the participant was cued to perform one of two possible movements through the BMI system, HO and HC. While performing the movements, the participant observed a single-hand clock on a computer screen, with numbers from 5 to 60, completing a full rotation in 2.56 s. Movements were triggered by the activation of the neural decoder, but the NMES was always activated congruently with decoded motor commands. In addition, 300 ms after HC was executed, a 1000 Hz "beep" was produced, lasting 100 ms (operant condition). No additional consequence followed HO execution (non-operant condition). Here, we focused on pre-movement signals occurring before the differentiation between operant and non-operant trials, and our key contrast is between the voluntary and involuntary sessions. Thus, we pooled trials from the operant and non-operant conditions in the present analyses. The participant was instructed to pay attention to the location of the clock hand at the time of

movement onset and to report it at the end of the trial, allowing us to measure the perceived timing of the action. Differently from Experiment 1, the movements were self-paced, meaning the participant was instructed to freely initiate the movement and encouraged to vary his waiting time, which should, in any case, exceed one full clock rotation. The second involuntary movement session (low agency) was identical to the voluntary movement session, with the only difference that the same movements were executed by activating the NMES for HO or HC at a random time while the participant was instructed to remain at rest. Each session consisted of 80 trials (40 HC, operant, and 40 HO, non-operant). Experiment 2 is part of a comprehensive set of experiments in which the intentional chain was manipulated, and reports about the perceived timing of intention, action, and effect were collected[38]. For consistency with Experiment 1, where the analysis was time-locked to action onset, we focused on conditions where action timing was reported.

## Experiment 3 - participants

Thirty healthy participants (13 females, age = 26.6 years old, SD = 3.99) took part in this study. All participants were right-handed and had no psychiatric or neurologic history. All participants were naive to the purpose of the experiment and gave written informed consent to take part in the study. They were remunerated for their time with 20 Swiss francs per hour. The ethical approval for the project was granted by the Vaud canton ethical committee.

## Experiment 3 - EEG acquisition

EEG data was collected via a 64-channel EEG eego mylab amplifier (AntNeuro, Hengelo, Netherlands) with a sampling rate of 512 Hz, referenced to the CPz electrode. Impedances were kept below 20 KOhm.

## Experiment 3 – protocol and rationale

When extending our investigation to healthy participants, we chose to focus on the paradigm of Experiment 1 rather than the one of Experiment 2, as we believe that producing genuine temporal binding effects in a BMI setup would require externally inducing a real upper limb movement following BMI decoding, which is hardly feasible in healthy participants and produces sensory feedback that is not comparable to natural movements. On the other hand, simply using virtual movements on a screen would result in a purely visual temporal judgement task between two visual events (the virtual arm and the clock), without necessarily implying any temporal estimation about actions, thus providing no information about the sense of agency. The experiment lasted a total of about 3 h and consisted in two parts: BMI training-testing and agency assessment.

## Part 1 – BMI training

After setting the EEG, participants sat at a table in a light and sound-controlled environment with their hands placed below a computer screen lying horizontally on the table. They had to alternate periods of rest and kinaesthetic motor imagery of their right hand to train the BMI algorithm in 3 blocks of 30 trials.

For motor imagery, they were asked to imagine the physical sensation of squeezing their right hand. They had to imagine a continuous and strong squeezing effort as if they were on the verge of making a movement, but without contracting the arm, shoulder, or face muscles. For rest, they were instructed to relax as much as possible and especially avoid thinking about movement. Participants were asked to keep a constant strategy throughout the training session. During each trial, a grey fixation cross appeared on the screen, showing one of two different cues in randomised order: Rest or Motor Imagery (MI). Then, a red bar started filling the fixation cross to signal the participant for how long he had to maintain the rest or motor imagery (4 s for each trial). Once the recordings were completed, the

BMI algorithm was trained on the participant's data to produce an individualised motor imagery decoder while the participant had a 5 min break. If the decoder's cross-validated accuracy on the training data was below 0.6, another block of 30 trials was collected, and training was restarted. Before moving on to the second part, it was verified that the participant was able to control the BMI. This step was also important to convince the participant that the BMI could truly detect whethershe/he is performing rest or motor imagery. Each participant performed 20 trials with either Rest or MI appearing on a grey fixation cross. This time, a red square moved left or right on top of the fixation cross displaying the decoder's output to the participant. 20 trial blocks were repeated while adjusting the motor imagery threshold until it would take ~ 3–6 s to reach it on successful trials. This had the aim to keep the task challenging but not impossible, while minimising the number of unwanted activations (see also below).

## Part 2 – agency assessment

In the second and main part of the experiment, participants saw two virtual, real sized open hands overlapping with their real hands which were kept below the screen. Participants performed motor imagery and provided agency ratings when the right hand closed. The aim of the experiment was to induce a trial-by-trial varying sense of agency, with minimal exogenous manipulations, and independently from sensory feedback, to highlight the role of endogenous neural oscillations in modulating agency ratings. A similar approach, keeping experimental stimuli constant and contrasting trials based on fluctuations in subjective reports, is the one routinely applied in similar previous studies, investigating the link between perception and pre-stimulus oscillations (e.g., refs. 29–31). We chose to use sham BMI as the main experimental condition, as it allowed us to reliably produce the illusion that the participant was controlling the virtual hand while being less subject to fatigue and decoder variability than real BMI. This allowed us to collect a large and constant number of trials regardless of each participant's proficiency with BMI. The illusion was made possible by the fact that in EEG-BMI, delays are long, and require a prolonged and continuous effort lasting several seconds to reach the decoding threshold. Therefore, even when truly controlling the BMI, the participant could not predict the exact timing of the movement, and thus habituated to experience control for delayed and temporally unpredictable movements. If the movement is provided at a randomised delay comparable to the intrinsic delay of the BMI system (hence the careful adjustment of decoding thresholds after BMI training), a sham BMI trial is hardly distinguishable from a real BMI trial. Sham BMI trials were cued and constituted 2/3 of the trials the remaining 1/3 of trials were self-paced and used real BMI, having the purpose to ensure participants remained unaware that part of the trials used sham BMI.

In sham BMI trials, participants saw a go cue appearing after a randomised delay of 1.5–2.5 s (uniform), at which point they had to start motor imagery. After a Gaussian randomised delay of $4.25 \pm 0.5$ (mean ± SD) seconds (limited between 3 and 5.5 s), the right hand closed with a continuous, pre-determined movement lasting about 0.4 sec irrespective of the decoder's output. Note that, despite the displayed movement being always spatially congruent with the imagined movement (hand closing), the temporal unpredictability induced by the randomised delay helps keeping the agency experience variable and the task meaningful. 1.4 s from the onset of the movement, they were asked to provide an agency rating "How much did you feel like you generated the movement? (From 1 to 9)". They reported their answer verbally and the experimenter entered it into the experimental software through a keyboard.

In real BMI trials, no go cue was displayed, and participants were instructed to start motor imagery whenever they wanted between about 1 and 5 from the onset of the trial. The right hand closed when the decoder reached the threshold. 1.4 s after the movement, subjects provided an agency rating like in cued trials. If the threshold was not reached within 10 s, the trial ended, and no rating was asked. Participants performed five blocks of 60 trials, separated by about 3 min of break. In total, 200 cued and 100 self-paced trials were collected.

Before the experiment, subjects were instructed to focus on pre-reflexive aspects of the control experience, and not to use cognitive reasoning to provide the ratings. They were also asked to focus on the differences between trials rather than on the absolute levels of agency, to provide variable ratings using all the available range, and considering 5 as an intermediate point to distinguish between higher and lower agency levels. Note that a potential bias in average ratings due to using 5 as a reference value cannot affect the results of our analyses, which are always based on relative agency ratings, compared within participants.

## Experiment 3 - BMI decoder

The Python based Neurodecode framework was used for EEG-BMI (https://github.com/dbdq/neurodecode). To obtain the classifier, a random forest was trained to discriminate Rest and MI trials from the training blocks (see[76]). 322 input features were used, consisting of average power values during each trial's 4 s Rest or MI period in 23 frequency bands, equally spaced from 8 to 30 Hz, measured in 14 sensorimotor channels (a 3 by 5 grid from FC3 to Cp4 excluding the reference, CPz). 8-folds cross-validation was used to test decoding accuracy, using 75% of trials as a training set and 25% as the test set.

## Experiment 4 – protocol

The principle of the experiment was to let participants perform a self-paced simple hand movement (index finger lift) and to show them, in place of their real hand, a virtual hand performing the same movement (3D animation) but at a random time. Two cases could, therefore, occur: either the motor onset precedes the visual onset, or the visual onset precedes the motor onset. For each trial, the participants answered a forced choice question about their sense of agency for the movement. The system was precisely tuned to guarantee an optimal visual correspondence between the real and virtual hands, a precise recording of the timings, and to minimise the time interval between the two events by influencing the randomisation of the animation onset time.

10 right-handed participants were recruited. The study was undertaken in accordance with the ethical standards as defined in the Declaration of Helsinki and was approved by the local ethics research committee at the University of Lausanne. Participants sat at a table and placed their right hand on a block containing a touch sensor. A monitor occluded vision of their real hand and projected a stereoscopic 3D virtual hand holding a virtual block (Supplementary Fig. S11a). Participants, wearing a pair of stereoscopic 3D glasses (nVidia 3DVision), were instructed to align their right arm and hand such that the virtual hand position corresponded to where they felt their real right hand to be. Head movements were restrained with a chin rest, and the experiment took place in a darkened room.

Participants first learned to lift their finger such that it matched the velocity and amplitude of the virtual finger in a short training block (consisting of 20 trials). They were asked to fixate on a point located between the virtual thumb and index finger to lift their right index finger from the block at a time of their choosing, and to leave it lifted for a short duration. For these trials, the visual feedback onset was synchronous to the motor onset. The subsequent trial began when the participant placed their finger back on the sensor.

The procedure for the main experiment was similar to the training block, with two exceptions. First, we experimentally manipulated the visual onset delay such that there was no longer a synchrony between the motor and visual onset times. Second, participants were asked about their sense of agency for the movement at the end of the trial (i.e., after both the virtual and real fingers were lifted): "Did the movement that you made correspond to the movement that you saw?"

Answers to the question were provided by button press with the subject's left hand. Following the response, an inter-trial interval was inserted before the next trial began (uniformly sampled from 100, 200, 300 or 400 ms). Participants completed 600 trials over 4 blocks.

## Experiment 4 – delay manipulation

For each trial of the main experiment, we defined three events: the onset of the appearance of the virtual scene ($T_0$), the motor onset ($T_M$), and the visual onset ($T_V$) (Supplementary Fig. S11b). Furthermore, we defined $\Delta T$ to be the difference between $T_V$ and $T_M$. As we were most interested in the sense of agency for $\Delta T$ values close to 0 (near-synchrony between the visual and motor movements), we employed a dual strategy. First, with the aim of providing visual consequences that closely precede the movement onset ($T_V < T_M$), we used a heuristic predictive algorithm to anticipate the participant's motor onset time for a given trial. This online algorithm set the visual onset time to be the prediction of the motor onset time from the per-subject motor onset history profile. In particular, $T_V$ was sampled from a Gaussian distribution with a mean based on a Gaussian approximation of the previous motor onset times, but with twice the standard deviation. The algorithm guaranteed that the $T_V$ could not be < 150 ms. Second, in the case that the algorithm failed to precede the participant's movement ($T_M < T_V$), the visual consequences were presented with a delay sampled from a uniform distribution in a window of interest (0 ms $\Delta T <$ 750 ms). The actual distributions of delays obtained through this procedure can be seen in Supplementary Fig. S11c.

## Experiment 4 - EEG acquisition

EEG data was collected via 64-channel EEG (Biosemi Inc, Amsterdam, Netherlands), with a sampling rate of 2048 Hz. Impedances were kept below 15 KOhm.

## Experiment 1 and 2 - data pre-processing

For LFPs, the main focus of the present study, data pre-processing, consisted of four steps: trial selection, artefact removal, downsampling, and epoching. Trial selection had the main goal of discarding trials in which the participant failed to generate any movement, or to activate the correct decoder. Therefore, we only kept trials in which the participant managed to keep the cued decoder above the threshold for at least 600 ms (6 classifier bins). In Experiment 1, we additionally required that such movement happened after the go cue, and at least 1.5 sec before the "stop", in order to ensure a sufficient time window for epoching. In addition, in Experiment 2, 5 HC trials from the high agency session had to be removed due to technical issues with the recording. After trial parsing, we retained 422 out of 640 trials for Experiment 1 (66%), and 61 out of 80 (76%) trials from the high agency session in Experiment 2 (all 80 trials were retained in the low agency session as the participant did not need to activate the decoder in this session). The parsing was relatively even across conditions of interest, with 114/160 for V + /S +, 93/160 for V + /S −, 117/160 for V −/S +, and 98/160 for V −/S −. Similarly, in Experiment 2, we retained 26/35 (HC) trials for the movement eliciting the sound, and 35/40 for the movement not eliciting the sound (HO). Artefact removal was performed before epoching, as done online for BMI decoding, with the difference that we applied an 8.7 ms blanking window, in order to be more conservative on oscillatory analyses. Then, the data was down-sampled to 1000 Hz, using a Kaiser anti-aliasing kernel. Spiking activity was extracted through the Wave_clus spike detection and sorting algorithm[77] with default settings. For the detection, a threshold was set at four times the standard deviation of baseline noise. Spikes were clustered through the superparamagnetic clustering algorithm, allowing to remove spurious signals. Since data collection was done in different sessions spanning several weeks, and the number of units in each channel fluctuated across sessions, we did not attempt to match units across recording sessions and considered each unit in each session separately (e.g., in Fig. 5c). For both LFP and multiunit activity,

the data was epoched by time-locking to the onset of hand movements (sensory feedback). The exact timing of the onset of hand movements (70 ± 10 ms after the neural decoder crossing threshold) was determined by detecting the 50 Hz stimulation artefact induced by the NMES system.

## Experiment 3 - data pre-processing

EEG analyses were performed using FieldTrip[78] and custom MATLAB code. Data was filtered in the 0.5–45 Hz range, a 50 Hz notch was applied, and epochs going from − 2 to 2 s from movement onset were created in FieldTrip, based on the precise timing of a photodiode detecting the onset of hand movement. At this stage, data was re-referenced from the original CPz reference to the average reference. Bad epochs presenting movement or muscular artefacts were manually rejected, and bad channels were selected and removed prior to independent component analysis (ICA). ICA was then run on the already epoched data with FieldTrip, with bad epochs and channels removed. Components corresponding to ocular movements were manually identified and removed before projecting the signal back to electrode space. Bad channels were re-inserted at this stage by interpolating neighbouring channels.

## Experiment 3 - source reconstruction and templates

Source reconstruction was performed in FieldTrip using a standard head template. The inverse solution was computed on a grid of 3835 points regularly spaced by 7.5 mm through eLORETA. Then, source space data was projected to a set of regions of interest derived from the Lausanne atlas[79] (scale 2, 129 ROIs), from which subcortical regions were excluded obtaining a total of 114 ROIs. To obtain the signal of each ROI, singular value decomposition was performed on all the signals from solution points falling within the ROI. For connectivity analyses, the dimensionality of EEG data was further reduced by grouping together ROIs by hemisphere (left or right) and lobe (frontal, parietal, temporal, occipital). Connectivity values were first computed at the ROI level and then averaged within each of the eight macro-regions corresponding to the left and right portion of each of the four cerebral lobes (see Supplementary Fig. S13 for the anatomical map).

## Experiment 4 – data pre-processing

Data pre-processing was the same as for Experiment 3, with the only difference that epochs were adapted to the shorter duration of trials, extending from 700 ms before the onset of the visual movement to 300 ms after.

## Experiment 1 - Multiunit analyses

Analyses on multiunit activity consisted of the evaluation of the coherence between spiking activity in M1 and LFP oscillations at different frequencies. For each frequency, the spike-LFP phase-locking value (PLV) was calculated by extracting the phase vectors for LFP oscillations at each spike location (through the procedure described above) and then applying formula (1) to the ensemble of phase vectors. For this analysis, all units from all channels were pooled. The analysis was re-applied separately for each unit to obtain Fig. 5c.

## Experiment 2 - statistical analysis of behavioural data

Due to the strongly non-normal distribution of responses with frequent outliers (Kolmogorov-Smirnov test $p < 0.0001$ in both voluntary and involuntary conditions, K-statistic = 0.80 and 0.99, respectively), we used the median as a robust indicator of the subject's perceived movement timing and performed a Wilcoxon rank sum test to compare the two conditions.

## Phase opposition analyses

Instantaneous values for power and oscillatory phase were obtained by convolving the signal with Morelet wavelets over 10 linearly spaced frequencies between 4 and 13 Hz, setting the number of cycles at 2π

(4π for EEG signals to compensate for the lower signal-to-noise ratio). Our main analysis focused on quantifying phase opposition in time and frequency between conditions of interest (high-low levels of explicitly or implicitly assessed agency). As a measure of phase opposition, we use the phase opposition product (POP)[39]. To compute the POP, first, we computed the inter-trial phase coherence (ITC) for all the trials pooled together, and for the two conditions separately. The ITC at a given time and frequency was defined as follows

$$ITC_{ALL} = \left| \sum_{all\ trials} \omega_i / |\omega_i| \right| / n \qquad (1)$$

$$ITC_A = \left| \sum_{condition\ A} \omega_i / |\omega_i| \right| / n_A \qquad (2)$$

$$ITC_{ALL} = \left| \sum_{condition\ B} \omega i / |\omega_i| \right| / n_B \qquad (3)$$

Then, the POP is simply obtained as follows

$$POP = ITC_A ITC_B - ITC^2_{ALL} \qquad (4)$$

where $n_A$, $n_B$ and $n$ respectively represent the number of trials for condition A, condition B, and total number. The notation $\omega_i$ indicates the analytic signal for trial i, the complex number whose absolute value is the instantaneous amplitude, and whose argument is the instantaneous phase at a given time and frequency. For the different methods used to extract analytic signals, see further paragraphs. The underlying idea is that, if trials within a condition are clustered around some angle, and trials in the other condition are clustered around an opposed angle, then the inter-trial coherence within conditions is going to be higher than when pooling the trials together. Therefore, higher values of POP indicate a stronger phase opposition between conditions. Instantaneous values of oscillatory power were simply computed as the squared absolute value of the wavelet convolution. Since in our analyses we focus on low frequencies, which are expected to be highly coherent on the small spatial scales of a Utah array, all analyses were performed on the mean LFP across all channels for Experiment 1 and 2. The high coherence of oscillations in the 4–13 Hz range was confirmed by supplementary analyses (Supplementary Fig. S14). The analysis was performed independently for each channel or ROI in Experiment 3. We also separately assessed the role of the phase of higher frequency oscillations (15–40 Hz) in supplementary analyses (Supplementary Fig. S2).

## Oscillatory power analyses
In addition to the effect of the phase, we tested whether oscillatory power could predict agency ratings. We used the same Morelet wavelets as for the phase to extract instantaneous power values in the 4–40 Hz range and compared it between the high and low agency conditions by means of a T-test, applying the cluster-based correction for multiple comparisons (Supplementary Fig. S1).

## Experiment 1 and 2 - statistical analysis of phase opposition
Statistical analyses of the time-frequency distribution of POP values were performed through cluster-based permutation tests to address the multiple comparison problem[80]. In order to run the permutations, we started by defining a suitable statistic for POP values, as with a single subject it is not possible to simply run a T-test on POP values across subjects. To the best of our knowledge, the analytical form for the null distribution of POP values is not known, so we obtained p-values by comparing true POP values to POP values obtained by shuffling the labels of high and low agency trials over 10000 permutations. The *p*-value was then simply given by counting the number of

permutations with a higher POP value. The same procedure was applied to all permutations so that a time-frequency map of p-values was available for each permutation. P-values obtained this way were then transformed into T-values, and a threshold of 2 (corresponding to *p* - 0.05) was set to define the clusters. The total value of each cluster was then defined as the sum of the T-values of all time-frequency points composing it. Importantly, the exact nature of the statistics used at this stage to define cluster scores does not influence the test's ability to appropriately control for type I errors, as this is addressed by the permutations performed subsequently[80]. The final p-value for each cluster was defined as the probability of finding a cluster with a larger score over the 10000 permutations. The analysis was performed over a 1 s window ending at the time of movement onset.

To rule out that the relation between the pre-movement phase and sense of agency could be due to differences in BMI decoding delays for high and low agency trials, we compared the delay between go cue and NMES movement onset between high and low agency trials, and between trials with optimal vs. non-optimal phase. We found no difference (t(29) = 0.15, *p* = 0.84 and t(29) = 0.71, *p* = 0.48 respectively, Supplementary Fig. S15). We also tested whether phase opposition results could be replicated by time-locking the LFPs to the crossing of a simple threshold on average M1 firing rates rather than to the onset of the actual hand movement, and found no significant result (Supplementary Fig. S16). We believe this was the case because the BMI decoder does not produce a movement by threshold crossing of population activity, but rather by decoding specific activity patterns across neurons[74]. Thus, population rate threshold crossing is an internal event which, unlike the BMI decoder crossing threshold, does not produce movements or salient subjective percepts. To simultaneously account for the amplitude and phase of 8 Hz LFP oscillations, we compared the analytic signals ($\omega_i$) for high and low agency trials at 8 Hz, − 256 ms, the time and frequency of strongest phase opposition, without normalising them to unit length to include amplitude information. We used a Hotelling $T^2$ to simultaneously compare the phase and amplitude of the LFP. The test was significant ($T^2(208) = 21.4$, *p* < 0.0001, Supplementary Fig. S17), indicating that phase differences were significant also when considering 8 Hz LFP amplitude. A further T-test on the sole amplitude was not significant (T(208) = 1.03, *p* = 0.31), confirming that the observed differences emerge from phase rather than amplitude differences. To confirm that, due to the non-causal nature of the wavelet transformation, the pre-movement effect was not induced by post-movement signals, we repeated all phase opposition analyses by extracting the phase through a Hilbert transform coupled with a causal filter[40]. We first filtered the signal in the frequency range of the effect (6–10 Hz for the implanted participant, 8–13 Hz for healthy subjects). Then, we applied a Hilbert transform to the filtered signal, and the analytic signal $\omega_i$ was computed from the signal as the complex number whose real part is the original signal, and the imaginary part is the Hilbert transform. The analyses showed significant phase-opposition before movement onset for Experiments 1–3 (Supplementary Fig. S18).

## Experiment 3 – validation of sham-BMI and statistical analysis of behaviour
We assessed the validity of the sham-BMI setup, by testing whether sham and real BMI trials could be distinguished by participants and led to different agency levels. We found that sham BMI and real BMI trials elicited comparable agency levels (mean sham BMI 5.75 ± 0.13 SEM, mean real BMI 5.57 ± 0.21 SEM, t(29) = 0.99, *p* = 0.33, Supplementary Fig. S9). In addition, when told that they were truly in control of the virtual hand only in self-paced trials, at the end of the experiment, all subjects reported that they were not aware of it. Thus, participants were not aware of the sham BMI setup, and believed to be causing the hand movement. The analysis was then performed only on sham BMI trials, reducing variability due to decoder accuracy (see protocol for the rationale).

Further analyses showed that the delay between go cue and movement significantly influenced agency ratings, with 17 out of 30 subjects showing a significant negative correlation and 1 subject showing a significant positive correlation (see Supplementary Fig. S19). To remove the effect of such an exogenous factor, we regressed the effect of delay on agency ratings within each participant before splitting trials and computing POP values.

## Experiment 3 – statistical analysis of phase opposition

We contrasted the highest and lowest 33% of detrended agency ratings of each participant, in order to exclude uninformative intermediate values. The POP analysis was performed similarly to the intracranial data, using permutations of trial labels to obtain p-values for each frequency and time point. The main difference in the statistical analysis depended on the presence of a pool of multiple subjects instead of a single participant. P-values at the group level were obtained by permuting trial labels within each subject, computing POP values, and then averaging the POP values across subjects for real and permuted data. Again, p-values were computed by counting the number of occurrences of larger average POP values in permuted data. The procedure was performed on the average POP values in the 8–13 Hz,− 0.5-0 s range. To demonstrate the robustness of our results to the specific choice of such frequency range, we performed supplementary analyses showing that the main results hold for a broad set of frequency ranges which include the alpha band (Supplementary Fig. S11).

## Multiple comparison corrections and conventions for reporting *p*-values

Unless otherwise specified, all p-values reported in the main text are corrected for multiple comparisons. Namely, we used cluster-based permutation tests (see sections above) whenever correcting across frequencies, time points, time-frequency points or ROIs for the pre-movement phase effect and connectivity analyses. Two ROIs were defined as neighbouring if the minimal distance between any two voxels belonging to the two regions was smaller than 5 mm. When correcting across cerebral lobes, due to the small number of comparisons, we used Bonferroni correction to be maximally conservative. Also, unless otherwise stated, all *p*-values are two-tailed. The only one-tailed comparisons are those for the finer spatial resolution analysis of information directionality (Fig. 7g) since the direction of the effect was pre-selected based on results at the coarser spatial resolution.

## Experiment 3 – individual alpha peak analysis

To test the hypothesis that the phase opposition effect frequency may be linked to the individual SMA alpha peak, we correlated each participant's alpha peak to the frequency of the maximal SMA phase opposition. To extract the individual alpha peak, we started by regressing out from the left SMA power spectrum the background 1/f component, extracted by fitting a power law on the 5–30 Hz power spectrum. Then, we defined the individual alpha peak as the point of maximum convexity in the alpha-band (8–13 Hz) power spectrum, i.e., the sharpest local peak within the alpha band. To define the maximal phase opposition frequency, we averaged POP values across time points in the − 0.5/0 s window and took the frequency of minimum *p*-value across the 10000 permutations. We then correlated individual alpha peak and maximal phase opposition frequencies with a linear regression.

## Experiment 3 - Determination of optimal phase angles and trials

The analysis was performed on ROI level data, focusing on the left SMA, where the phase-driven modulation of agency ratings is strongest. First, we tested whether the optimal phase angles at the group-level highest POP time-frequency point (9 Hz, − 304 ms) were comparable for all participants. For each participant, we computed the optimal phase angle by taking the average phase angle of high-agency trials and averaging it with the flipped average phase angle of low-

agency trials, at 9 Hz and − 304 ms in the left SMA. We tested whether the distribution of such optimal phase angles was significantly different from a circular (i.e., random) distribution through a Raleigh test, and the test was not significant at *p* = 0.13. We concluded each subject may in principle, have a different optimal phase angle. This is in line with what observed in similar works[32] and may be exacerbated by individual differences in the frequency of the strongest effect. We then decided to define optimal phase angles (and trials) individually. For each subject, we first identified the frequency at which the strongest phase modulation of agency ratings was observed within the 8–13 Hz range. Then, we extrapolated and averaged phase angles at that frequency in the 500 ms preceding the movement, and computed the optimal angle as described above. We show that the main results of the subsequent connectivity analysis are not sensitive to the specific choice of the frequency range for defining the optimal angle, as they hold for a broad set of frequency ranges which include the alpha band (Supplementary Fig. S9). We excluded from further analyses 5 subjects showing no significant phase opposition in any time-frequency point in the 8–13 Hz, − 0.5-0 s range, as the optimal phase angle cannot be meaningfully defined. We performed supplementary analyses to show that the main results on connectivity hold also without excluding the 5 participants not having significant phase opposition (Supplementary Fig. S20), or using the global optimal angle rather than the individualised optimal angle (Supplementary Fig. S21).

## Experiment 3 - Connectivity analyses

Connectivity analyses were performed in FieldTrip on ROI signals, using the debiased weighted phase-lag index (debiased WPLI)[51]. We focused on the temporal window from 0.2 to 1.2 s post-movement, spanning symmetrically the temporal interval between movement onset (0 s) and the behavioural response (the agency rating was requested at 1.4 s), where we expect phase-related connectivity changes to be strongest. For each subject, trials were split in optimal or non-optimal, depending on whether the phase was closer than π/3 to the individual optimal phase or to the non-optimal phase, selecting approximately 33% of the trials within each group. Then, average connectivity values were computed for each subject and pair of ROIs for optimal and non-optimal trials, for 42 linearly spaced frequencies in the 4–45 Hz range. We then averaged connectivity values from the left SMA to all other regions and contrasted them between optimal and non-optimal trials by mean of paired t-tests and corrected for multiple comparisons across frequencies using a cluster-based permutation test. We analysed connectivity values in the frequency range, which showed significant changes in this first analysis (9-12 Hz) separately for each of the eight macro-regions, by averaging all connections from the left SMA to all other ROIs belonging to each macro-region. Furthermore, we checked that the effect on connectivity may not be simply driven by changes in oscillatory power at the level of SMA, by comparing oscillatory power for optimal and non-optimal trials (Supplementary Fig. S22).

The directionality analysis was performed using custom MATLAB code and using a method inspired by previous work on hippocampal theta oscillations[52]. The key idea is to look at time-delayed correlations between the phase of oscillations of the seed region (left SMA) and of ROIs belonging to the target region, within the usual time-frequency range (0.2–1.2 s, 9–12 Hz). For each pair of analysed regions and frequency (9, 10, 11 and 12 Hz), we computed the difference of oscillatory phase angles (extracted through Morelet wavelets as described above), while introducing a variable delay in the signal of the target region. Then, we computed the inter-trial coherence (ITC) of phase differences as a function of the temporal lag. The idea is that the delay at which maximal coherence is observed indicates the preferred direction of information flow. If the delay on the target region is positive, it means that the present of the seed region best predicts the future of the target, and that information flows from the former to the latter. The difference was computed separately for each temporal

delay, trial, frequency and ROI of a macro-region, and subsequently averaged to keep only its dependency on temporal lag. To perform statistical analyses on the effect of the pre-movement phase, we extracted the peak of the lag-ITC curve in optimal and non-optimal phase trials for each subject and compared them using a paired t-test. We checked that our results were not sensitive on the range of temporal lags used in the analysis by repeating our main analyses on different sets of temporal lags (Supplementary Fig. S23).

## Experiment 4 – data analysis

We aimed to replicate the main results of Experiment 3 by contrasting high and low agency trials in the left M1 and SMA. As for all previous experiments, to enhance the contribution of endogenous oscillations, it is necessary to reduce the contribution of exogenous factors, in this case, visuomotor delays. We thus excluded trials with visuomotor delays yielding an average rating above 80% or below 20% (Supplementary Fig. S11d). This way, we excluded trials where agency is constantly very low or very high, and focused on those where agency ratings were more variable and thus potentially explained by endogenous oscillations. Then, we extracted instantaneous phase angles and computed the phase opposition product between high and low agency trials exactly as in Experiment 3. Statistical analyses were also the same as in Experiment 3, using 10000 permutations to compute p-values for phase opposition values. This was done for individual time-frequency points, subsequently applying cluster-based correction for multiple comparisons and on the average phase opposition values in the 8/13 Hz, Supplementary Fig.0.5/0 s region (Supplementary Fig. S11e, f).

## Reporting summary

Further information on research design is available in the Nature Portfolio Reporting Summary linked to this article.

## Data availability

Data from the implanted participant cannot be made available due to privacy reasons. Pre-processed EEG data is available at https://osf.io/4w3t6/?view_only=c5c108ae37304e8f9ed9a9aad61bd763. Source data are provided in this paper.

## Code availability

Commented MATLAB code for replicating EEG phase opposition analyses is available at https://osf.io/4w3t6/?view_only=c5c108ae37304e8f9ed9a9aad61bd763. For further information, contact the corresponding author (T.B.).

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

## Acknowledgements

T.B. was supported by a fellowship of the MINDED Marie Skłodowska-Curie COFUND Action (grant agreement No. 754490), within the European Union's Horizon 2020 research and innovation programme. In addition, this work was supported by a Swiss National Science Foundation Professorship grant, grant number 163951, attributed to A.S. This research was further supported by two generous donors advised by CARIGEST SA (Fondazione Teofilo Rossi di Montelera e di Premuda and a second one wishing to remain anonymous), the Bertarelli Foundation, the Empiris Foundation, and Parkinson Suisse to O.B.

## Author contributions

T.B., J.P.N., N.E., B.H., A.R., A.S. and O.B. designed the experiments, T.B, M.B., S.C., B.O., N.E., B.H. and A.R. implemented the experimental set-ups, T.B., J.P.N., M.B., C.F., S.C., B.O., N.E. and B.H. collected the data, T.B., C.F. and J.P.N. analysed the data, T.B. produced the original draft and figures, S.P., C.B, O.B. and A.S. supervised the work. All authors edited the manuscript.

## Competing interests

The authors declare no competing interest.
