## [Transparent Peer Review file · Nature Communications]

Pre-movement sensorimotor oscillations shape the sense of agency by gating cortical connectivity

Corresponding Author: Dr Tommaso Berton

Version 0:

Reviewer comments:

Reviewer #1

(Remarks to the Author)

Berton and colleagues investigated the relationship between “alpha oscillations” at the level of M1 and the sense of agency perceived by a tetraplegic individual over specific movements mediated by a brain-machine interface (BMI). In particular, the authors manipulated the congruence between intended and executed actions. The results suggest that “low-alpha” oscillations recorded in M1 during the pre-movement phase were associated with the level of (explicit and implicit) sense of agency (SOA) in the patient during the execution of the experimental tasks.

This link between pre-movement alpha oscillations and the sense of agency was further supported by an EEG study on healthy controls using a (sham) EEG-BMI setting: this highlighted the role of SMA in the agency experience rather than area M1.

There are interesting elements of novelty in the Ms. Yet, there are a few problems as well.

My general impression, also considering recently published work from this group, is that the Authors might be working on a large data-base from multiple experiments, not always preplanned in a concerted manner: some data were probably described also in reference #30. For example, Experiment 1 has already been described in a previous paper, where the authors described the role of area M1 in the agency experience. The authors should therefore better specify the novelty of the present data with respect to the previous ones and how these results might be integrated with the previous paper.

Coherence. In the present Ms, the reader is forced to learn from multiple different -unvalidated- paradigms tapping the SOA rather than, for example, an orderly attempt to measure the same behavioural variables in a very interesting patient and in a group of normal controls. This is inevitable when moving from implicit to explicit SOA; yet the implicit SOA paradigm could have been the same for the patient and the controls. The implications are that the importance of the non-negligible differences in the results cannot be assessed well.

The presentation of the data also suffers from a myriad of control analyses on the data, apparently not planned in the methods, that one finds unexpectedly: this makes reading the MS quite cumbersome and a detailed commentary on each and every aspect impossible in a single run of review. I will therefore concentrate on the main points that require clarification before any further consideration.

I found stylistic, theoretical and methodological issues that should be considered. I'll start with the methods.

Methodological issues.

As said, overall, I have the impression that the present drawing from a larger data set might have created some confusion while drafting the MS and surely in the reader as the methods are described sometimes in an incomplete manner: for example, some experimental manipulations or abbreviations are never explained, and so forth. The authors should double-check that all details are provided for a clear comprehension of what was done and for the reproducibility of the procedures and replicability of the results.

Experiment 2. The authors adopted a Libet-like paradigm to measure the implicit component of the sense of agency. The patient performed a movement through BMI, which was followed by a sound. According to the intentional binding phenomenon (Haggard et al., 2002), voluntary movements are perceived as shifting in time towards the generated sound, while the sound is perceived as occurring earlier, closer to the action. Based on this phenomenon, the higher this "time compression" that "binds" the voluntary action to its outcome, the stronger an implicit sense of agency for the action and the produced physical outcome. Here, the authors only measured the perceived time of the movement onset while ignoring the timing of sound perception. More importantly, they observed the opposite result with respect to the intentional binding phenomenon since voluntary movements were perceived EARLIER in time. This would imply a lower implicit sense of agency, according to the intentional binding phenomenology. The authors discussed this result as a temporal binding between intentions and actions, but they cannot provide any evidence for this explanation; moreover, they do not explain how this binding can be considered a marker of increased implicit sense of agency (the cited reference #30 is a preprint on the very same data).

Incomplete methods description. There is some missing information in the MS. Figure 3a mentions an "operant movement" condition, which is not described in the text. I assume that there might be a baseline "non-operant movement" condition, in line with the intentional binding paradigm, but I cannot find any information about this. These conditions are mentioned but not described also in Figure S15.

Experiment 3. According to the main text, participants were instructed to imagine a specific (unknown) right-hand movement, followed by (congruent?) visual feedback. At the end of each trial, they were asked to rate their perceived explicit agency. First, I was wondering how there could be an agency modulation if the movement to imagine and the displayed feedback were always the same. Maybe I missed some details here. However, the main issue here is related to the behavioural dependent variable that (i) should validate the use of sham EEG-BMI trials and (ii) should guide the EEG analyses: the agency ratings. These agency judgments were inevitably biased by the experimenters' who warned participants "to keep the mean rating around 5" in the two conditions; for this reason, they cannot represent a valid measure of agency.

Data interpretation

Puzzling interpretation of the LFP data. Exploration of the LFP data, in Figure 3c, suggests that the two curves clearly dissociate AFTER movement execution. I was wondering why the authors concentrate only on minute differences before Time 0 and do not discuss this result.

The "alfa band" is really the same in all experiments?

For some reason, the Authors have decided that signals coming from different frequency ranges are the same thing. In the patient, the frequency of the electrical signal measured from area M1 peaks at 6.2 Hz (below the canonical alfa); in the normal controls the peak of the electrical signal measured this time from area SMA peaks at 9-12 Hz. I am not convinced by this analogy. As said, and unfortunately, because the paradigms used in the patients and in normal controls are not the same these differences are not interpretable.

Coherence of the results

Contrary to what was seen in the patient, the normal control EEG data do not reveal activity in area M1 but in the SMA. The authors do not seem to be bothered by this. Yet this discrepancy deserves more than a comment. Indeed, as the two signals come from different areas, measured with different techniques, some differences should be expected. The key common thing here is that the crucial -small- EEG signal that correlates with the agency measures comes before action. This is the more solid cross-validation from the patient to the normal controls.

Theoretical issues.

A paper of this magnitude cannot be neutral towards the previous theories on the sense of agency. There are at least three theoretical models proposed to address the arising of the agency experience. The authors seem to consider only one of these nor they do make an effort to discuss whether and why their findings are in support of this specific theory. The "Comparator Model", the only model mentioned in the paper, postulates that the sense of agency arises from a series of comparator processes within the motor control system. However, in an alternative perspective, the "Apparent Mental Causation Theory" proposes that the emergence of the sense of agency is a post hoc and retrospective phenomenon. Accordingly, the sense of agency arises through an inferential sense-making process that occurs after the completion of a movement. Finally, in the "Cue Integration Theory", the sense of agency should arise from integrating low-level cues (sensorimotor proprioceptive and exteroceptive cues) and high-level cues (cognitive cues). The weight assigned to these different cues varies for the implicit and explicit components of agency. Hence, I feel that the authors should try to frame their data considering the previous theoretical and experimental efforts on the very same topic over the last 20 years.

Style.

For sure, there is already a broad body of literature investigating the neural correlates of the sense of agency that makes an opening statement in the abstract ("..... the underlying neural mechanisms of the sense of agency are still unknown") too

strong for two reasons: (1) the previous evidence connecting brain physiology and the SOA (2) the correlational nature of the evidence provided; this lacks the identification of a clear "mechanism" that causally connects brain physiology and the complex phenomenology of the SOA.

Reviewer #2

(Remarks to the Author)

The manuscript by Berton et al reports very interesting data and experimental designs. The authors recorded single neurons and LFP from M1 of a tetraplegic patient who, with the help of a prosthetic device, is able to guide fake-hand reaching movements. The authors correlated the LFP with the reported simulated action of the subject and specifically with his sense of agency, demonstrating a phase dependence of alpha band endogenous oscillation with the sense of agency. To complete the study the authors measured the effect in typical subjects, and used high density EEG coupled with simulation of the moving hand to demonstrate again a phase dependence. Unfortunately and not surprising this data are less compelling with a small and just significant effect.

While I like the paper and the material presented, I have several unclear and not-discussed technical problems that make me uneasy to give a strong recommendation at this stage. The main one being the synchronization with the prosthetic device. For this reason I would like to postpone my acceptance/rejection recommendation after a major revision giving the possibility to the authors to explain better their technique and rationale and write a more clear and critical manuscript.

Main problem

Phase depends on the time origin of the synchronisation trigger. Here the authors trust the prosthetic algorithm and set the synchronisation with the onset threshold for the fake-hand movement. However, this relies on a very strong assumption: the algorithm should operate with the same delay for the high and low agency perception. This may be wrong and would be mainly CONTRARY to their data. They show that the agency state signal can be derived from the phase of the LFP and supposedly also the decoding algorithm is using this signal, possibly by introducing a delay (50 ms would be sufficient) to give an apparent phase change. In other words, the argument to me is circular!

However, the authors have access to single cell firing. A simple threshold to the ensemble recording firing could be used for synchronize the LFP and analyse the change in phase. It is true that they found a strong correlation between pre-movement of fake-hand LFP and neuronal firing. However, the amount of firing may be independent of onset phase. Maybe I misunderstood something, and the tetraplegic patient does not modulate at all M1 firing during Motor Imagery. But if so, this needs to be explained in detail, at least in the methods. In addition the variability of the decoding delay to generate the phantom hand movement needs to be reported!

Again, I am surprised that the Readiness Potentials, usually associated with SMA activity, are never present in recordings, either in the patient or in the typical subjects. Probably they have been filtered out given the high-pass with 0.2Hz. If so I advise to analyse and show them. These potentials are so strong and rich of information about the internal motor state and motor intention of the participants that it may contain already important information about agency.

Figure 5. Could you split the M1-LFP correlation between high and low agency trials? Did you obtain a difference in the correlation? Low agency may reduce correlation and this would be a stronger and physiological more important effect that the phase difference of LFP.

Results of Fig 6 are somewhat disappointing. If anything, the phase opposition seems from Fig 6e stronger around 500 ms post movement. The scale of Fig 6d is not really readable to judge the robustness of the effect.

Why has figure 7f directionally phase effects only on temporal cortex? Blue means that that ROI was not analysed? Please explain.

Please make clear if 8Hz is theta or alpha for you.... To me looks like a typical subject alpha band LFP.

Version 1:

Reviewer comments:

Reviewer #1

(Remarks to the Author)

I thank the authors for their response and the clarifications they gave about their experiment. Despite these clarifications, I maintain reservations regarding the experimental paradigm used and the conclusions that can be drawn from it.

Experiment 2. The authors compared action timing judgments that were given after voluntary and involuntary actions, pooling together operant (Hand closing movement followed by sound) and non-operant (Hand opening movement alone) conditions (page 7 of the rebuttal letter).

1st issue: pooling together different actions and different conditions. The sense of agency is defined as "the feeling of

making something happen" (Haggard, 2017). Here, the authors are mixing together conditions in which (different) actions caused a sound (i.e., the action makes happen a tone) with conditions in which the action did not produce any kind of external feedback (i.e., the action does not make happen a tone). In other words, they considered two very different conditions as equivalent, according to the literature on the sense of agency (see also the next comment).

2nd issue: Comparing voluntary vs involuntary actions. The authors compared action timing judgments after voluntary and involuntary (unpredictable) actions. The two conditions clearly differ based on the fact that the passive movement cannot be predicted: "In the involuntary session, the movement was randomly generated via the NMES system". This may explain the delay in action timing judgment observed with respect to voluntary movements.

The authors correctly cited Haggard et al., 2002 mentioning that this paper compared active and passive movements: indeed, Haggard et al. compared action timing judgements in voluntary vs involuntary actions yet this was done AFTER subtracting for the same measures collected in control conditions: timing judgment for voluntary actions recorded in the action+sound trials compared with timing judgment for voluntary actions recorded in the action-alone trials OR timing judgments for TMS-induced actions recorded in the action + sound trials compared with timing judgments for TMS-induced actions recorded in the action-alone trials, see Table 1 of Haggard et al., 2002).

In the present Ms, the authors adopted a different approach that cannot control for possible intervening factors not directly related to the agency dimension (e.g., surprise effect by unexpected externally generated muscle twitches).

Figure 2, shown in Haggard et al., 2017 may help in clarifying my comment (below called "The Figure").
<https://www.nature.com/articles/nrn.2017.14>

Regarding the 1st issue, the authors here are pooling together conditions represented by the fourth (operant condition) and second (non-operant condition) rows of the upper part of the Figure (voluntary-action condition).

For what concerns the 2nd issue, the authors are comparing the conditions illustrated in the second/fourth row of the upper part of TheFigure with the second/third row of the lower part of The figure.

To sum up, the authors of the present Ms are mixing together conditions that, in the intentional binding literature, refer to experimental and control conditions. Accordingly, there is no isolation of the crucial judgements of intentional acts and ensuing consequences.

Experiment 3. In the first version of the paper, the authors acknowledged that participants were invited to "keep the mean (agency) rating around 5". I criticised this approach since it seems that they were actively inviting their subjects to give specific rating values and thus manipulating the dependent variable of the experiment.

In the revised version of the paper, they now write that participants: "were also asked to focus on the differences between trials rather than on the absolute levels of agency, to provide variable ratings using all the available range, and consider 5 as an intermediate point to distinguish between higher and lower agency levels."

This is something very different, and I am not sure that such changes from the original to the revised version are acceptable. On a related note, they confirmed that no agency manipulation was applied in this experiment. This was aimed at measuring a sort of "intrinsic noise" in agency ratings. I am not sure that such noise can be considered related to the agency dimension since it does not vary according to a specific agency manipulation.

Finally, in my previous comment, I mentioned: "Contrary to what was seen in the patient, the normal control EEG data do not reveal activity in area M1 but in the SMA. The authors do not seem to be bothered by this. Yet this discrepancy deserves more than a comment.

Indeed, as the two signals come from different areas, measured with different techniques, some differences should be expected. The key common thing here is that the crucial -small- EEG signal that correlates with the agency measures comes before action. This is the more solid cross-validation from the patient to the normal controls."

The authors replied: "Based on the previous literature about the neural bases of the sense of agency (see e.g., Haggard, Nature Neuroscience Reviews, 2017) and our results in the implanted patient (Experiment 1 and 2), we expected to observe phase opposition in motor and premotor areas. Consequently, we targeted our search for phase opposition on motor and premotor regions (n = 12), applying FDR correction for multiple comparisons within this network. This analysis revealed the strongest phase opposition in SMA, and that the second strongest phase opposition in M1 (p = 0.023). If focusing purely on the left M1, with the purpose of replicating the effect observed in the implanted participant, and thus dropping multiple comparisons correction, the uncorrected M1 effect is highly significant with p = 0.003. The uncorrected p-value is not reported for in the text, for being maximally conservative. In sum, phase position was significant in both SMA and M1. For completeness, we also reported the results of a whole-cortex search for phase opposition across all 114 ROIs of the cortex. This whole-cortex search confirmed significant phase opposition in the SMA after adjusting for 114 comparisons; in contrast, M1 did not survive multiple correction. We refrain from making interpretations of the null result for M1 in this whole-cortex search as this analysis is underpowered".

I have the following comments:

1. Haggard 2017 did not mention M1, but temporo-parietal and premotor regions.
2. The whole brain analysis cannot survive a formal multiple comparison correction, and the authors commented on this result as a consequence of low power: I am not sure that this kind of comment is acceptable; given the simplicity of running an EEG experiment in normal controls, there is no excuse for low-powered experiments. Given that expanding the same sample of subjects at this stage is not acceptable, a replication of this experiment with a fresh sample of subjects could make the whole story more tenable.

On a final note, the authors mentioned Wegner as a support to their approach: "Finally, concerning the validation of our Sham BMI approach. As correctly pointed out by Wegner in his mental causation theory, it is entirely possible for healthy individuals to experience a genuine sense of agency even when not truly in control of an external event, as we believe was

the case in our sham BMI setup.”

However, please note that this model was based on a “reconstructive” vision of the sense of agency, whereby the agency experience is formed AFTER the execution of the action. This is something different from what the authors are supporting in this paper.

Reviewer #2

(Remarks to the Author)

The authors addressed my main concerns about the technical issue very thoroughly and clearly. I now appreciate well the design of the experiment and I found that the additional analysis that they performed dismissed all the possible criticisms that I raised in the previous version, and I believe that the data strongly support the fact that the phase of pre-movement low-alpha rhythms in M1 and SMA encodes the sense of agency. Given the importance and the novelty of the message, the originality of the technical design and the fact that the intracortical recording in M1 human are rare and precious, I strongly support publication of the manuscript in Nature Communication.

However, in the present form the manuscript still needs an additional revision to meet the standard of the journal.

Clarity of the writing:

1) The introduction is very general and not informative about the state of art of mechanisms and circuits of sense of agency; the same criticism applies to the literature on phase of endogenous oscillations encoding motor information. Also about half of the introduction is about the presentation of the logic and result of the experiments, that is not useful for the reader given the complexity of the experiments. In addition, the same data from the same patients have already been published. It is important that the readers know about this in the introduction. The authors should clearly state and describe the results already obtained and published and, if possible, these should be used to motivate this new research. This means a fresh rewrite of most of the introduction.

2) The language used in the paper to explain analysis procedures is not appropriate, using many technical terms that are used currently in the EEG laboratories, but often mathematical incomplete. One author is an excellent mathematician, and he should check accurately the language. For example equation 1, line 891, they not report what is W_i and in any case as it not mathematically correct. If W_i is the complex number associated with the frequency it should be added the term Arctangent of the imagery/real part...if it is already phase, it cannot be divided by the norm of the phase! Similarly, no indication of the other term in the equation, like the number of trials. Another example in lines 944 and 1335. The Hilbert transform is not needed to calculate the phase at the various frequency. What exactly has been done to calculate it? Another example appearing many times, time-frequency point is a colloquial term: there are maxima energy points or other point that can be marked on other specifications in the time-frequency domain. Please correct. Another example in line 1325. Normalization is associated mathematically to a division, while inspecting the figure there is clearly also a subtraction. Line 243 very mathematically unclear: do you mean that phase is expressed with respect to a different origin? In other words, that the phase has been rotated to take into account the difference in delay respect to experiment 1? In most cases I understood what the authors have done, because it is what normally it is done in EEG or LPF analysis: but please express the procedure in mathematically correct language in these examples and in many other instances.

Data analysis:

The authors assess differences in phases between the two conditions using inter-trial phase coherence and apply an equation that maximize anti-phase difference. However, in principle the system could work in quadrature phase that offers other advantages. It is essential to show that a difference in phase of the 8Hz component of the average LFP between the two conditions is statistically significant and report the value. This can be obtained simply by applying circular statistics across the vector (amplitude and phase) cluster for the two conditions. I think that it is important to consider simultaneously amplitude and phase of the 8 Hz oscillation.

Figure 3h reports LFP amplitude, but it is never defined what exactly it is measured. Is it the amplitude at 8 Hz, so it becomes negative due to the phase? Or it is the value in voltage of the LFP at the time point chosen? Please explain also in the text.

Concetta Morrone

Version 2:

Reviewer comments:

Reviewer #2

(Remarks to the Author)

The authors have dealt adequately to all my criticisms, and I strongly support publication in the present form

Reviewer #1 (Remarks to the Author):

Bertoni and colleagues investigated the relationship between “alpha oscillations” at the level of M1 and the sense of agency perceived by a tetraplegic individual over specific movements mediated by a brain-machine interface (BMI). In particular, the authors manipulated the congruence between intended and executed actions. The results suggest that “low-alpha” oscillations recorded in M1 during the pre-movement phase were associated with the level of (explicit and implicit) sense of agency (SOA) in the patient during the execution of the experimental tasks.

This link between pre-movement alpha oscillations and the sense of agency was further supported by an EEG study on healthy controls using a (sham) EEG-BMI setting: this highlighted the role of SMA in the agency experience rather than area M1.

There are interesting elements of novelty in the Ms. Yet, there are a few problems as well.

We thank the reviewer for their careful revision of our work. Please find our point-by-point response below. We highlighted our responses in bold, and indicated with quotation marks and *italics* passages reported from the revised text, highlighting changes in red. To allow referring to specific points, we have numbered reviewer comments and our responses (e.g., comment 1: C1; response 1: R1). The numbering continues across comments from the two reviewers.

C1)

My general impression, also considering recently published work from this group, is that the Authors might be working on a large data-base from multiple experiments, not always preplanned in a concerted manner: some data were probably described also in reference #30. For example, Experiment 1 has already been described in a previous paper, where the authors described the role of area M1 in the agency experience. The authors should therefore better specify the novelty of the present data with respect to the previous ones and how these results might be integrated with the previous paper.

R1)

Experiment 1 and 2 represent separate analyses of a comprehensive set of experiments performed in a BMI participant with an M1 implant, and the present work also includes a large separate study in healthy participants, Experiment 3. Some data from Experiment 1 and 2 have been described in two previous studies. The first focused on the role of sensory feedback in explicitly assessed sense of agency and its encoding in *post-movement* LFPs and multiunit activity (Serino et al., *Nat Hum Beh*, 2022). The second focused on temporal perception of BMI-mediated actions used as an implicit measure of agency and how this is reflected in M1 activity (Noel et al., *Biorxiv*, 2023).

While the two previous works studied the effect of *exogenous* manipulations on the sense of agency and its neural encoding, the present manuscript focuses on the role of *endogenous* signals, i.e., the state of the brain *before* movement onset, which is determined by internal fluctuations rather than external sensory inputs. While *exogenous* contributions have been extensively studied, reports on the *endogenous* factors determining the sense of agency are much scarcer in the literature. Specifically, here we investigated the role of *pre-movement* theta-alpha oscillations in the sense of agency, and their link with M1 spiking activity and whole brain connectivity. To this aim, we applied to Experiment 1 and Experiment 2 completely new analyses on the power and phase of pre-movement oscillations, while previous works focused on LFP amplitude and multiunit activity. The scientific questions and analyses in Experiment 1 and Experiment 2 are thus entirely novel with respect to previously published data.

Additionally, we designed and performed Experiment 3 specifically to corroborate our findings from the implanted patient in a cohort of healthy participants, to extend our investigation to whole brain signals and to study our hypothesis on brain connectivity, which could not be investigated in the patient data where we recorded only from M1.

We have revised our manuscript to clearly acknowledge the use of previously collected data and better explain the novelty of our hypothesis and analysis approach with respect to both previous studies, as visible on pages 3 and 5:

“Nevertheless, we still lack specific knowledge of the mechanisms leading to the integration of endogenous pre-movement signals with post-movement reafferent information.” (p. 3)

“Data analysed in Experiment 1 were collected as part of a comprehensive set of experiments performed in a BMI participant with an M1 implant. In a previous work, we investigated how post-movement LFP amplitude and multiunit activity in M1 encode exogenous sensory feedback congruency, and how these signals covary with agency judgements for BMI actions¹³. Here we conducted novel analyses on these previously collected data focusing on endogenous signals, to test the hypothesis that pre-movement theta-alpha oscillations in M1 predict agency judgements.” (p. 5)

We also now discuss the relation between present results and the previously published findings on page 19:

“In our previous work¹³ (including data from Experiment 1), we showed how post-movement LFPs and multiunit activity in M1 encode congruency between motor commands and sensory feedback, a key aspect in the sensorimotor comparisons underlying the sense of agency. Our findings in Experiment 1 extend these previous results, suggesting that M1 and SMA also play pivotal roles in the sense of agency at an earlier stage, when sensorimotor predictions are computed during motor preparation and before execution.”

This way, we hope to have clarified that the approach, analyses, and results of the present study are entirely novel.

C2)

Coherence. In the present Ms, the reader is forced to learn from multiple different -unvalidated- paradigms tapping the SOA rather than, for example, an orderly attempt to measure the same behavioural variables in a very interesting patient and in a group of normal controls. This is inevitable when moving from implicit to explicit SOA; yet the implicit SOA paradigm could have been the same for the patient and the controls. The implications are that the importance of the non-negligible differences in the results cannot be assessed well.

R2)

Thank you for giving us the opportunity to better clarify these important aspects. In the present study, we aimed at investigating the role of pre-movement oscillations in modulating the subsequent sense of agency. Although the paradigms used present some differences, mainly due to the necessity of adapting our investigation from an implanted patient to healthy controls, the principle underlying our key phase opposition analysis is applied coherently across these experiments. Across three experiments, we divided trials based on explicit or implicit measures of agency (and not on experimental manipulations, whose impact we tried to minimise or regress out), and studied how the pre-movement oscillatory phases were clustered in the high and low agency groups of trials.

Both Experiment 1 (whose paradigm is presented in our previously published study, Serino et al., *Nat Hum Beh*, 2022) and Experiment 3 use classic agency judgements/ratings as measures of agency. Thus, we believe that they measure the same behavioural variable. The main difference between Experiment 1 and Experiment 3 is that the sensory feedback presented to the tetraplegic participant (visual and somatosensory) and able-bodied participants (visual only) are different, due to the difficulty of reliably implementing somatosensory feedback in able-bodied individuals. Any externally implemented movement of the upper limb of a healthy participant, via, for instance, a robotic exoskeleton or functional electrical stimulation, would result in completely different sensory feedback than that produced by the NMES system in our tetraplegic participant (see also below). Said that, importantly our key analysis, phase opposition, was based on a contrast between trials with different subjective reports, and not between experimental conditions. Furthermore, our main result was found in the pre-movement period, so it should be

independent from the specific type of sensory feedback used to elicit a sense of agency. Thus, within the scope of the present results, we believe that Experiment 3 represents the closest conceptual replication of Experiment 1 which could be performed in able-bodied individuals.

Concerning another point made by Reviewer: -“the implicit SOA paradigm could have been the same for the patient and the controls”:

The possibility of reproducing our BMI-based temporal judgement task (Experiment 2) in healthy participants was indeed the object of in-depth discussion in our group, but was finally discarded as it was deemed technically impossible in healthy participants. The crucial point is that, while visual feedback alone can be used to elicit a sense of agency and is thus suitable for explicit judgements (used in Experiment 1 and 3), a combination of visual and somatosensory feedback would be needed for a paradigm based on action timing perception, like in Experiment 2. Indeed, simply judging the timing of BMI-generated virtual hand movements, while observing a clock on the same screen, would result in a purely temporal visual task between the two visual cues (the virtual hand and the clock), independently from judging the time of an action. On the other hand, implementing actual hand movements in healthy participants based on the decoding of motor imagery would instead be technically challenging, requiring either FES stimulation, or using an external exoskeleton. More importantly, these solutions provide sensory feedback that is not comparable to that of a natural arm movement, thus potentially biasing the sense of agency associated to temporal judgement. Finally, these methods are uncomfortable and potentially painful, and may interfere with motor imagery and BMI decoding. Based on these considerations, we now explain our choice to extend Experiment 1 to Experiment 3, rather than Experiment 2, in more detail, on page 14 and 27:

“To investigate the potential contribution of areas beyond M1, in Experiment 3, we devised an EEG-based version of Experiment 1, which we believe to be the closest conceptual extension of that paradigm achievable in healthy participants.” (p. 14)

“When extending our investigation to healthy participants, we chose to focus on the paradigm of Experiment 1 rather than the one of Experiment 2, as we believe that producing genuine temporal binding effects in a BMI setup would require externally inducing a real upper limb movement following BMI decoding, which is hardly feasible in healthy participants and produces sensory feedback that is not comparable to natural movements. On the other hand, simply using virtual movements on a screen would result in a purely visual temporal judgement task between two visual events (the virtual arm and the clock), without necessarily implying any temporal estimation about actions, thus providing no information about the sense of agency.” (p. 27)

For the validation of the implicit task described in Experiment 2, and of the sham BMI paradigm of Experiment 3, please refer to our detailed responses in subsequent points of the present response letter (R5 and R9).

We further conducted new analyses and rephrased passages in the results and discussions to better support the coherence of our results between the implanted patient and healthy controls, namely the frequency and anatomical localisation of the phase opposition effect. Please refer to the detailed responses to the relative comments (R11 and R12).

To conclude, we agree with the reviewer that differences in our setups, mainly required by the uniqueness of our implanted participant, may have made the comparison of our results more difficult, and we thank the reviewer for giving us the opportunity to better address this point as presented in R11 and R12. On the other hand, we believe that these differences allowed us to show that pre-movement oscillations consistently predicted the subsequent sense of agency in different cohorts of subjects, using different recording techniques and behavioural paradigms, speaking in favour of the generality of our results.

C3)

The presentation of the data also suffers from a myriad of control analyses on the data, apparently not planned in the methods, that one finds unexpectedly: this makes reading the MS quite cumbersome and a detailed commentary on each and every aspect impossible in a single run of review. I will therefore concentrate on the main points that require clarification before any further consideration.

R3)

To avoid that control analyses impact the readability of the main text of the manuscript, we have streamlined their presentation, describing control analyses and their rationale in the methods (pp. 31, 32, 33, 34, 35) and removing unnecessary references to supplementary figures from the results section. Since we believe that these control analyses still represent an added value, demonstrating the solidity of our results, we have chosen to keep the original control analyses in the supplementary figures.

I found stylistic, theoretical and methodological issues that should be considered. I'll start with the methods.

Methodological issues.

C4)

As said, overall, I have the impression that the present drawing from a larger data set might have created some confusion while drafting the MS and surely in the reader as the methods are described sometimes in an incomplete manner: for example, some experimental manipulations or abbreviations are never explained, and so forth. The authors should double-check that all details are provided for a clear comprehension of what was done and for the reproducibility of the procedures and replicability of the results.

R4)

Indeed, we have realized that some details of experimental procedures and methods were not sufficiently clear and improved their presentation. We have improved the revised methods, figures and figure captions to better explain these details (pp. 7, 10, 24, 25, 26, 27, 28, 29, 30).

C5)

Experiment 2. The authors adopted a Libet-like paradigm to measure the implicit component of the sense of agency. The patient performed a movement through BMI, which was followed by a sound. According to the intentional binding phenomenon (Haggard et al., 2002), voluntary movements are perceived as shifting in time towards the generated sound, while the sound is perceived as occurring earlier, closer to the action. Based on this phenomenon, the higher this "time compression" that "binds" the voluntary action to its outcome, the stronger an implicit sense of agency for the action and the produced physical outcome.

Here, the authors only measured the perceived time of the movement onset while ignoring the timing of sound perception. More importantly, they observed the opposite result with respect to the intentional binding phenomenon since voluntary movements were perceived EARLIER in time. This would imply a lower implicit sense of agency, according to the intentional binding phenomenology. The authors discussed this result as a temporal binding between intentions and actions, but they cannot provide any evidence for this explanation; moreover, they do not explain how this binding can be considered a marker of increased implicit sense of agency (the cited reference #30 is a preprint on the very same data).

R5)

Here the Reviewer is raising a few related points, and we are grateful as it would allow us to clarify our approach.

First, and main point: while our paradigm is inspired by the intentional binding task, as well as to the Libet experiment, the uniqueness of our BMI setup and design resulted in a new paradigm, allowing us to investigate a new form of binding between intention and action that differs from the classical intentional binding between action and effect. A full analysis and discussion of the intentional binding effect in this setup are the focus of the previous work by our group cited in the present manuscript (Noel *et al.*, *Biorxiv*, 2023, currently under review at *Current Biology*), and goes beyond the aim of the present study. To better reflect the uniqueness of our setup, we have rephrased the introduction of Experiment 2, and now describe our paradigm as an implicit agency paradigm based on time perception. In particular, the difference between the perceived time of actions preceded by subject's intention to move and those passively implemented is taken as an index of the sense of agency. See page 8:

“The previous analyses establish a relationship between pre-movement low-alpha oscillations and explicit agency judgements. We next tested whether the same phase opposition can distinguish between high vs. low agency actions as defined from an implicit marker of agency based on the subjective perception of the timing of self-initiated movement³². Leveraging our BMI setup, in a previous study, we have shown that temporal judgements of voluntary actions triggered by the participant's intention to move are anticipated compared with involuntary actions triggered by NMES, resulting in a temporal compression between the intention to move and the action³².

The experimental paradigm and temporal compression results are extensively reported in our previous work³². Below we provide a brief summary of the methods and findings in the subset of conditions relevant to this study. A rotating clock was displayed on a screen, and the participant was asked to report the position of the clock at the onset of a hand movement triggered by the NMES system (Fig. 3a). In the voluntary session, the action was triggered by the participant's intention to move as decoded by the BMI system. In the involuntary session, the movement was randomly generated via the NMES system without motor intention. The participant perceived voluntary BMI-generated movements as occurring earlier relative to their actual timing than involuntary movements (median voluntary = -497.8 ± 299 ms interquartile range, median involuntary = -384 ± 185 ms, Wilcoxon $p = 0.033$, Fig. 3b). We thus leveraged this finding to classify trials associated with putatively higher agency, as those showing stronger intention-action temporal compression, and trials with putatively low-agency as those showing weaker temporal compression. To this aim, we performed a median split of trials within the voluntary session based on the amount of anticipation and computed the phase opposition product between trials in which the movement was perceived earlier (high agency) and trials in which the movement was perceived later (low agency).”

A second point by the Reviewer is that we focus our analysis on movement timing, not analysing effect timing, i.e., the tone. Please note that the perceived time of the tone and intention was also collected in this participant as reported in Noel *et al.*'s paper. However, to be consistent with Experiment 1, the analysis on pre-movement phase has to be time-locked to action onset. Therefore, also in Experiment 2 we focused on trials where action timing was reported. Indeed, time-locking the phase opposition analysis to the effect would include post-movement signals. On the other hand, it would be arbitrary to time-lock the analysis to the intention, as there is no objective timing for it. In sum, reports about intention and event go beyond the scope of the present manuscript, but they are the focus of Noel *et al.*'s paper. Thus, we prefer to cite these results in our manuscript rather than reporting them again in full. We now motivate our choice to focus on action timing in the methods, on page 26:

“Experiment 2 is part of a comprehensive set of experiments in which the intentional chain was manipulated, and reports about the perceived timing of intention, action and effect were collected³². For consistency with Experiment 1, where the analysis was time-locked to action onset, we focused on conditions where action timing was reported.”

A third point concerns the different effects between the classic intentional binding and our new intention-action compression. The paper by Noel *et al* focuses on a full analysis and discussion on the full “intentional chain”, i.e., the relationship between intention-action-effect as highlighted by temporal judgement. For convenience, we summarize their findings and interpretation in the following lines.

As highlighted above, with our tetraplegic participant, we observed that voluntary actions are perceived as happening earlier in time than involuntary ones. This was true regardless of whether these actions were or were not followed by a tone (Row I and IV vs II in Fig. 1 from Noel's paper, reported below). Importantly, the perceived time of intention was instead postponed when motor commands led to a bodily action compared to no action (Rows I and IV vs III). Thus, perceived timing of will and action seemed to be pulled towards each other when they happened in the same timeline, similarly to what observed for action and effect in classical intentional binding effects.

We report here Fig. 1 from Noel et al.'s paper, showing the perceived time of intention (red), action (green) and effect (blue) in six different types of intentional chain:

As reported in Noel's paper, this result was interpreted in the perspective of a recent computational accounts of intentional binding (Legaspi et al., *Nature Communications*, 2019), suggesting that perceived timing compression should occur not only between actions and their consequences, but between any pair of events that the brain determines to be causally related, such as intentions/motor commands and actions. Due to the inherent noise of neural recordings and decoding, BMI movements show a significant and variable delay ($\sim 1.2 \pm 0.48$ SD, see p. 24 of revised text) between motor commands and actions, so that the exact timing of the movement is subjectively hard to predict. When our patient executed voluntary movements, such delay got possibly "compressed" by perceived causality, as compared to involuntary movements. That this action-intention compression was not reported in classical intentional binding studies is unsurprising. First, no previous study systematically assessed the perceived time of intention and action in able bodied individuals. Second, the action anticipation effect might not be evident in healthy participants because the delay between motor commands and movements is arguably much shorter and subjectively more predictable than in our BMI setup. This possibly results in a smaller binding compared to the one occurring between action and effect, leading to perceive the action as occurring later. In our patient instead, due to the longer and unpredictable delays, intention-action compression may dominate over action-effect binding, leading to overall action anticipation.

In sum, we believe Noel's paper to be the most appropriate source of validation for our implicit index, and we have rephrased the presentation of Experiment 2 to better introduce such previous validation on page 8:

"Leveraging our BMI setup, in a previous study, we have shown that temporal judgements of voluntary actions triggered by the participant's intention to move are anticipated compared with involuntary actions triggered by NMES, resulting in a temporal compression between the intention to move and the action ³²."

Finally, the Reviewer expresses concern for the lack of an external validation as a marker of agency. In this regard, we wish to highlight that our approach is aligned with that adopted in the original intentional binding paper by Haggard et al., *Nature Neuroscience*, 2002. In that study, the conclusion that the temporal shift is an implicit marker of agency was drawn based on the observation that such a shift occurred in voluntary vs. involuntary movements. Here, we applied the same logic, taking the temporal shift occurring

in voluntary movements compared to involuntary ones (action anticipation) as an implicit marker of the sense of agency. Importantly, not only did we observe the same effect of 8 Hz pre-movement phase as in Experiment 1, but the phase relation was the same. In other words, restricting ourselves to facts without any interpretation, anticipated action perception was associated with precisely the same 8 Hz phase preceding explicit judgements of high agency in Experiment 1, *and* with voluntary movements compared to involuntary ones in Experiment 2. Although internal to our participant (due to the uniqueness of the setup), we believe the coherence of these results supports the interpretation that, in our setup, action anticipation is an implicit marker of higher agency. Still, we agree that the lack of external validation, due to the uniqueness of our setup, constitutes a limitation, as now acknowledged in a new limitations paragraph on page 22:

“Our unique BMI set up enabled us to relate pre-movement phase opposition to the temporal compression between intention and action observed for voluntary actions. More studies are needed to validate this intention-action compression as an implicit marker of agency.”

We have further toned-down the following passages in the introduction and results to reflect the putative state of such marker of agency, as visible on pages 4 and 8:

“In Experiment 2, we showed that these oscillations also predicted the temporal binding between intentions and actions, a putative implicit measure of the sense of agency³²” (p. 4)

“We thus leveraged this finding to classify trials associated with putatively higher agency, as those showing stronger intention-action temporal compression, and trials with putatively low-agency as those showing weaker temporal compression.” (p. 8)

C6)

Incomplete methods description. There is some missing information in the MS. Figure 3a mentions an “operant movement” condition, which is not described in the text. I assume that there might be a baseline “non-operant movement” condition, in line with the intentional binding paradigm, but I cannot find any information about this. These conditions are mentioned but not described also in Figure S15.

R6)

We apologize for the lack of clarity. Indeed, to study the full intentional chain (as done in Noel *et al.*, *Biorxiv*, 2023) two types of movements were used. One (hand closing, operant movement) led to a tone being produced 300 ms later, and the other (hand opening, non-operant movement) was not followed by a tone. Since the focus of the present manuscript is pre-movement signals, occurring before the timeline of operant and non-operant movements bifurcate, we pooled operant and non-operant movements in our analyses. The differential effects of operant and non-operant movements in temporal judgements are presented Noel *et al.* (*Biorxiv*, 2023). We now have better explained these aspects in the methods on page 26:

“Movements were triggered by the activation of the neural decoder, but the NMES was always activated congruently with decoded motor commands. Additionally, 300 ms after HC was executed, a 1000 Hz “beep” was produced, lasting 100 ms (operant condition). No additional consequence followed HO execution (non-operant condition). Here, we focused on pre-movement signals, occurring before the differentiation between operant and non-operant trials, and our key contrast is between the voluntary and involuntary session. Thus, we pooled trials from the operant and non-operant conditions in the present analyses.”

To increase the readability of the results section, we removed references to operant and non-operant conditions from Fig. 3, as the two conditions were pooled in the analyses and the purpose of the figure is to present the behavioural contrast between voluntary and involuntary movements.

C7)

Experiment 3. According to the main text, participants were instructed to imagine a specific (unknown) right-hand movement, followed by (congruent?) visual feedback. At the end of each trial, they were asked to rate their perceived explicit agency.

R7)

As described in the methods, participants had to imagine a continuous squeezing effort of their right hand. Visual feedback showed the virtual hand closing in all trials, so the feedback was always “spatially” congruent with the imagined movement. The text has now been revised to improve clarity (page 27):

“For motor imagery, they were asked to imagine the physical sensation of squeezing their right hand. They had to imagine a continuous and strong squeezing effort, as if they were on the verge of making a movement, but without contracting the arm, shoulder, or face muscles.”

C8)

First, I was wondering how there could be an agency modulation if the movement to imagine and the displayed feedback were always the same. Maybe I missed some details here.

R8)

Experiment 3 was designed based on our hypothesis on pre-movement oscillations and findings from Experiment 1 and 2, suggesting that even in the absence of external manipulation, variability in agency ratings for BMI actions is to be expected, and such variability should be (at least in part) explained by the pre-movement endogenous phase, which is independent from sensory feedback. In this sense, note that the phase effect in Experiment 1 held even at fixed sensory feedback congruency (Fig. 2g) and is thus purely endogenous. Thus, the goal of Experiment 3 was not to measure the effect of exogenous manipulations on the sense of agency induced by sensory feedback, already broadly studied (e.g., Serino et al., *Nature Human Behaviour*, 2022, Marchesotti et al., *Human Brain Mapping*, 2017; Evans et al., *PLoS One*, 2015), but to study the link between “residual” variability in agency ratings which does not depend on external manipulations but on endogenous pre-movement oscillations. To effectively do this, we aimed creating a setup in which, at a given sensory feedback, the sense of agency is uncertain and variable, and thus more likely to depend on endogenous factors such as the pre-movement phase. If we simply asked participants to perform actual movements and rate their sense of agency, agency ratings would be constantly very high and no meaningful variability would be observed, therefore we used a BMI setup. We expanded the description of the task in the methods to better explain its rationale on page 28:

“The aim of the experiment was to induce a trial-by-trial varying sense of agency with minimal exogenous manipulations, and independently from sensory feedback, to highlight the role of endogenous neural oscillations in modulating agency ratings.”

For these reasons, participants were asked to imagine squeezing their right hand to generate virtual movements of a hand closing. Visual feedback was always the same and “spatially” congruent with the imagined movement. To keep agency uncertain and not have a stereotyped task, we randomly manipulated the temporal delay between go cue and sensory feedback. Importantly, our key analysis was based on contrasting high and low agency trials based on participant’s reports, and not on the exogenous delay manipulation, to maximize the effect of endogenous factors such as the pre-movement phase. Indeed, since the pre-movement phase is endogenous and cannot depend on externally determined post-movement conditions, external manipulations may actually confound the phase effect, and were thus kept at a minimum and regressed out from the agency ratings before the phase opposition analysis, as stated in the methods on page 33:

“To remove the effect of such exogenous factor, we regressed out the effect of delay on agency ratings within each participant before splitting trials and computing POP values.”

C9)

However, the main issue here is related to the behavioural dependent variable that (i) should validate the use of sham EEG-BMI trials and (ii) should guide the EEG analyses: the agency ratings. These agency judgments were inevitably biased by the experimenters' who warned participants "to keep the mean rating around 5" in the two conditions; for this reason, they cannot represent a valid measure of agency.

R9)

We are sorry that the too original description of our instructions, which was too concise, has generated a misunderstanding. Our instructions clearly explained to the participants to use 5 as an "anchor" between the highest and the lowest agency trials, and to focus on the differences between trials to provide variable ratings spanning the whole scale. This was to avoid having participants constantly reporting the extreme value of the scale, which was noted in few pilot subjects who always reported very high agency, hindering meaningful splitting of trials for phase opposition analyses. We have now rephrased the passage to better reflect the detailed instructions provided to participants on page 29:

"Before the experiment, subjects were instructed to focus on pre-reflexive aspects of the control experience, and not to use cognitive reasoning to provide the ratings. They were also asked to focus on the differences between trials rather than on the absolute levels of agency, to provide variable ratings using all the available range, and considering 5 as an intermediate point to distinguish between higher and lower agency levels."

Importantly, even if the instructions had affected the *average* agency rating (anchoring it for instance around 5), this would be irrelevant for our analysis, as it would not affect the subset of trials being assigned to the "high" or "low" agency condition based on participant's rating. Thus, we believe that the agency ratings collected during our experiment are appropriate for our EEG analyses. We included these considerations to the methods on page 29:

"Note that a potential bias in average ratings due to using 5 as a reference value cannot affect the results of our analyses, which are always based on relative agency ratings, compared within participants."

Finally, concerning the validation of our Sham BMI approach. As correctly pointed out by Wegner in his mental causation theory, it is entirely possible for healthy individuals to experience a genuine sense of agency even when not truly in control of an external event, as we believe was the case in our sham BMI setup. To validate this, we compared agency ratings between sham and actual BMI trials, and found no significant difference (Fig. S7). The reasonings reported above concerning the instructions given to the participants apply also in this case. Indeed, even if the instructions biased average ratings towards 5, there would be no reason to expect that such bias would be different between sham and BMI trials. Since the comparison of ratings to validate the sham setup is done within participants, we believe it cannot be affected by such potential bias.

We expanded and clarified passages about the rationale and validation of our sham setups in the methods on pages 28 and 33:

"We chose to use sham BMI as the main experimental condition, as it allowed us to reliably produce the illusion that the participant is controlling the virtual hand, while being less subject to fatigue and decoder variability than real BMI. This allowed us to collect a large and constant number of trials, regardless of each participant's proficiency with BMI. The illusion was made possible by the fact that in EEG-BMI delays are long, and require a prolonged and continuous effort lasting several seconds to reach the decoding threshold. Therefore, even when truly controlling the BMI, the participant could not predict the exact timing of the movement, and thus habituated to experience control for delayed and temporally unpredictable movements. If the movement is provided at a randomized delay comparable to the intrinsic delay of the BMI system (hence the careful adjustment of decoding thresholds after BMI training), a sham BMI trial is hardly distinguishable from a real BMI trial." (p. 28)

"We assessed the validity of the sham-BMI setup, by testing whether sham and real BMI trials could be distinguished by participants and led to different agency levels. We found that sham BMI and real BMI

trials elicited comparable agency levels (mean sham BMI 5.75 ± 0.13 SEM, mean real BMI 5.57 ± 0.21 SEM, $p = 0.33$, Fig. S7). In addition, when told that they were truly in control of the virtual hand only in self-paced trials, at the end of the experiment, all subjects reported that they were not aware of it. Thus, participants were not aware of the sham BMI setup, and believed to be causing the hand movement.” (p. 33)

Data interpretation.

C10)

Puzzling interpretation of the LFP data. Exploration of the LFP data, in Figure 3c, suggests that the two curves clearly dissociate AFTER movement execution. I was wondering why the authors concentrate only on minute differences before Time 0 and do not discuss this result.

R10)

We thank the reviewer for giving us the opportunity to discuss these important aspects. The pre-movement phase opposition appears small to visual inspection in the plots reporting trial-averaged traces (in Figs. 2c and 3c) because the trial-averaged LFP trace is not suited to visually highlight phase opposition effects, as it conflates in a non-separable way the effects of phase and amplitude of the LFP. To provide a better visualization of the contribution of phase and amplitude in discriminating between movement perceived early and perceived late, we now complement the trial-averaged LFP plot of Fig. 3c (and 2c) with plots of single-trial phase and amplitudes. To visually appreciate the phase effect, in Fig. 3d (reported below, also see the analogous Fig. 2d for Experiment 1), we plot the instantaneous phase for individual trials throughout the entire trial, highlighting (black dashed lines) the pre-movement phase opposition window of the alpha band for early vs. late movement perception. To illustrate the effects of amplitude, we also plot in Fig 3h individual-trial LFP amplitude, showing that early vs late differences are larger and more reliable in the post-movement than in the pre-movement period. However, these differences is less strong and reliable than differences in the pre-movement phase. In Fig. 3i we plot the histograms of post-movement LFP amplitude values across all trials for the early and late condition, at the timepoint of maximal difference between early and late trials. Further, there is considerably more overlap between the distributions of post-movement LFP amplitude (Fig. 3i) at the post-movement time with the strongest early-late discriminability than between the distributions of 8 Hz phase at the pre-movement time with highest early-late discriminability (Fig. 3f). Thus, pre-movement phase discriminates between trials with early and late perception better than post-movement amplitude does.

The visual inspection is confirmed by rigorous statistical analysis of the discriminability between high and low agency movements from the instantaneous phase and amplitude across the movement time, showing that pre-movement phase opposition is statistically stronger than post-movement amplitude differences. In Experiment 1, when movements started in the optimal phase, 74.8% of trials had high agency, while only 44.9% had high agency when movements started in the non-optimal phase. In Experiment 2, 75.8 % of trials in the optimal phase vs. only 21.2 % of trials in the non-optimal phase were associated with an early perception of the movement. This results in a p-value of 0.0001 for the phase opposition at peak time, with p being below 0.001 between -0.6 and -0.2 s pre-movement. For comparison, the minimum p-value in the amplitude comparison of the two LFPs was 0.00035, reached for a very short post-movement interval (we use p-values to compare effect sizes because there is no standard method to determine effect size for phase opposition). Thus, unlike the phase opposition effect, the LFP amplitude difference did not survive cluster correction for multiple comparisons, as shown in Fig. 3g.

We thank the reviewer for giving us the opportunity to clarify these aspects, which are now detailed in the results on page 10:

“Importantly, the early vs late differences in pre-movement LFP phase were far stronger and more significant in pre-movement LFP phase than in either pre- or post-movement LFP amplitude. LFP amplitude discriminated maximally between the two conditions at 544 ms post-movement (minimal p-value, t-test, Fig. 3g). Such difference was not significant after cluster correction for multiple comparisons across timepoints ($p > 0.08$). Accordingly, phase distributions within the pre-movement phase opposition cluster

(-256 ms, Fig. 3f) had much less overlap than LFP amplitude distributions at the post-movement timepoint of maximal amplitude-based discriminability across early and late movement perception (Fig. 3i)."

We report below the changes in Fig. 3 and the relative caption, visible on page 11:

“(c) Trial-averaged LFP for trials with early (blue) and late (red) perception of movement (median split). Shades denote standard errors. (d) Instantaneous pre-movement phase (as in Fig. 2d) for individual early (top) and late (bottom) movement perception trials, showing clustering of the two subsets around opposite phases. Black dashed lines indicate the time limits of the cluster shown in panel (e), the white dashed line indicates the timepoint shown in panel (f). (e) P-values for the phase opposition product between trials with early and late movement perception. The red contour delimits the significant cluster after a cluster-based permutation test. The black bracket above the plot indicates the time window of interest for cluster-based correction (-0.5/0 s). (f) Histograms of phase angles for individual trials at the time-frequency point of maximal phase opposition in Experiment 1 (-256 ms, 8 Hz, red cross in panel d) to allow comparison with Figure 2e. Blue/red lines indicate the preferred angles for high/low implicit agency respectively, and their length is proportional to the ITC. (g) Comparison of the statistical significance of phase differences (red, 10000 permutations) and LFP amplitude differences (black, t-test). Solid horizontal lines indicate time windows surviving cluster correction for multiple comparisons across timepoints. No significant cluster was found for LFP amplitude differences. (h) Instantaneous post-movement amplitude for individual early (top) and late (bottom) movement perception trials. (i) Histogram of LFP amplitudes at the timepoint of maximal dissociation between early (blue) and late (red) trials.”

In sum, besides the theoretical interest of this study on pre-movement neural oscillations, there is also a quantitative justification in focusing on the pre-movement phase. We hope the new plots and analyses following this important comment help illustrate the reasons for this choice.

That being said, an effect of pre-movement oscillations does not exclude that post-movement signals may be also contributing to the final reported sense of agency or movement timing, as now mentioned in the discussion on page 20:

“Our results do not rule out the contribution of post-movement signals to the sense of agency. These may be relevant for postdictive inference of causality (apparent mental causation theory ⁷), or for sensorimotor comparisons, whereby sensorimotor oscillations may be integrated with higher level cognitive cues to determine the final experience of agency (cue integration theory ⁶). This contribution may be reflected in a dissociation of post-movement LFP between high and low agency trials, which was here observed in Experiment 2 and extensively investigated in ¹³. Pre-movement sensorimotor oscillations may also serve as a trigger for apparent mental causation, and gauge the integration between low- and high-level cues in determining the sense of agency.”

C11)

The “alfa band” is really the same in all experiments?

For some reason, the Authors have decided that signals coming from different frequency ranges are the same thing. In the patient, the frequency of the electrical signal measured from area M1 peaks at 6.2 Hz (below the canonical alfa); in the normal controls the peak of the electrical signal measured this time from area SMA peaks at 9-12 Hz. I am not convinced by this analogy. As said, and unfortunately, because the paradigms used in the patients and in normal controls are not the same these differences are not interpretable.

R11)

Thank you for pointing out this aspect which deserves more in-depth discussion. There is a considerable variability across studies in the definition of frequency bands, and the neural origin of EEG or LFP activity at each frequency is often unclear. Reviewer 2 commented that the 8 Hz oscillations in the implanted patient “looks like a typical subject alpha band LFP” (see C21). However, we acknowledge that in the previous versions of the manuscript we have not been sufficiently clear in noting that we cannot and do not wish to assume or conclude that neural phenomena observed with different techniques in different paradigms, different subjects, and different frequencies are analogous or are generated by the same neural processes. We can only unbiasedly document where frequency information about agency is encoded in each experimental paradigm. We have revised discussion (page 22) to emphasize this:

“Furthermore, differences in techniques and experimental paradigms applied in one implanted participant and healthy controls make it difficult to infer whether comparable phase opposition effects observed at different frequencies across different experiments are related to the same or different neural mechanisms. Because of this, and of the general difficulties in imputing neural phenomena to frequency bands ⁴¹, here we refrained from making such inference, and unbiasedly reported the precise frequency of the observed effects in each subject and experimental paradigm.”

Said that, we run further analyses to characterize and understand the subject-to-subject variability within the same experimental paradigm in the frequency of maximal agency encoding in the phase. In particular, we analysed individual spectral peak of low frequency oscillations (for brevity: individual alpha peak). Since many behavioural effects seem to be tied to the individual alpha peak frequency (Haegens et al., Neuroimage 2014), or even to its fluctuations across trials within the same participant (Drewes et al., Cerebral Cortex, 2022), we hypothesized that also variations across individuals in the frequency of the strongest phase opposition could be reflect variations across individuals of their idiosyncratic alpha peak. The new analyses indeed showed that the individual frequency of maximal phase opposition correlates with the individual alpha peak in SMA. We now present and discuss this novel finding in the results section (page 14) and in Fig. 6f:

“The SMA effect peaked at 9 Hz, close to what was observed in our implanted participant, but was relatively spread across the whole alpha band (Fig. 6d). Since alpha-band peak frequencies vary across individuals ⁴⁵ and correlations of alpha-band activity with behaviour are stronger at frequencies closer to the individual

*alpha peak*⁴⁶, we predicted that individual variations in agency-related phase opposition might reflect individual variations in alpha peak frequency. Confirming this prediction, the frequency at which maximal SMA phase opposition was found for each subject correlated with their individual SMA alpha band frequency of maximal power ($R = 0.46$, $p = 0.011$, Fig. 6f). This suggests that individual variations in the frequency at which the phase better predicts agency depends on individuals' idiosyncratic alpha band peak." (p. 14)

We report below Fig. 6f and the relative caption:

“(f) Correlation between the individual alpha peak and the frequency of strongest phase opposition in the left SMA. Data of the tetraplegic participant from Experiments 1-2 (not included in the regression) is shown for comparison (red dot).”

Crucially, a spectral peak below the conventional alpha band has been previously and commonly observed in tetra or paraplegic participants (Foldes *et al.*, *J Neurophysiol*, 2017), including our implanted patient (6.2 Hz). Thus, although lower in absolute frequency with respect to healthy controls, the phase opposition observed in the implanted participant also peaked close to his individual alpha peak (see red dot in Fig. 6f above). As visible in the figure, most participants with a low individual alpha peak (around 8 Hz) presented a phase opposition effect peaking close to the one observed in the implanted participant.

To address the difference between M1 and SMA, we now also report the spectral peak in M1 for healthy participants, which is very close to the SMA spectral peak (Fig. S3). Therefore, we believe it is unlikely that the difference in frequency can be explained by the different anatomical location, but rather by the different frequency of the peak of power of oscillatory activity.

In sum, our new analyses suggest that the apparent discrepancy in frequency can be resolved by considering that: (i) the phase effect is tied to the individual low-frequency spectral peak and that (ii) the spectral peak in the implanted participant is lower than in healthy participants, likely due to the patient's chronic paralysis. This is now mentioned in the discussion on page 22:

“We could, however, gain some intuition about the individual variability of effects within Experiment 3. Namely, individual differences in the peak phase opposition frequency could be accounted for by individual variations in the individual alpha peak (Fig. 6f). Compatibly with other reports in chronic paralysis³⁹, the individual alpha peak in the implanted participant was lower than in healthy participants, possibly explaining the lower frequency of the phase opposition effect observed in Experiment 1 and 2.”

C12)

Coherence of the results.

Contrary to what was seen in the patient, the normal control EEG data do not reveal activity in area M1 but in the SMA. The authors do not seem to be bothered by this. Yet this discrepancy deserves more than a comment.

Indeed, as the two signals come from different areas, measured with different techniques, some differences should be expected. The key common thing here is that the crucial -small- EEG signal that correlates with the agency measures comes before action. This is the more solid cross-validation from the patient to the normal controls.

R12)

The different source of the effect from the patient and healthy individuals indeed deserved more clarification. We did observe an effect in M1 in healthy participants. We apologize if this was not reported and analysed with sufficient clarity in the original manuscript. We have rephrased this part to clarify our hypotheses and results.

Based on the previous literature about the neural bases of the sense of agency (see e.g., Haggard, *Nature Neuroscience Reviews*, 2017) and our results in the implanted patient (Experiment 1 and 2), we expected to observe phase opposition in motor and premotor areas. Consequently, we targeted our search for phase opposition on motor and premotor regions ($n = 12$), applying FDR correction for multiple comparisons within this network. This analysis revealed the strongest phase opposition in SMA, and that the second strongest phase opposition in M1 ($p = 0.023$). If focusing purely on the left M1, with the purpose of replicating the effect observed in the implanted participant, and thus dropping multiple comparisons correction, the uncorrected M1 effect is highly significant with $p = 0.003$. The uncorrected p-value is not reported for in the text, for being maximally conservative. In sum, phase position was significant in both SMA and M1. For completeness, we also reported the results of a whole-cortex search for phase opposition across all 114 ROIs of the cortex. This whole-cortex search confirmed significant phase opposition in the SMA after adjusting for 114 comparisons; in contrast, M1 did not survive multiple correction. We refrain from making interpretations of the null result for M1 in this whole-cortex search as this analysis is underpowered. Taking this into account, have rephrased the presentation of the results as follows (page 14):

“Based on the previous literature about the neural bases of the sense of agency (see e.g. ⁴⁴) and our results in the implanted patient (Experiment 1 and 2), we expected to observe phase opposition in motor and premotor areas. Consequently, we targeted our search for phase opposition on motor and premotor regions ($n = 12$, shown in Fig. S8). Two regions survived FDR correction for multiple comparisons across regions within this network. The stronger phase opposition was observed in the posterior part of the left supplementary motor area (SMA, $p = 0.0024$, Fig. 6c-e). Phase opposition in the dorsal part of the left primary motor cortex contralateral to the BMI movement was also significant ($p = 0.023$), consistent with our result in the implanted participant. No other ROI yielded significant phase opposition. For completeness, we repeated the analysis searching for significant phase opposition across all 114 ROIs of the source reconstruction. This whole-cortex search confirmed a significant phase opposition in SMA ($p = 0.03$), whereas phase opposition in M1 did not survive correction for multiple comparisons, likely due to the analysis being relatively underpowered. As shown in Fig. S9a-B, SMA effects were robust to the specific choice of the frequency range (Fig. S9a-b).”

It is also important to note that, while for further analyses in the main text we focused on SMA, the region showing the strongest effects, the key results on connectivity are also observed when using M1 as a seed (Fig. S10). Also note that we may have observed an effect only in M1 in the implanted participant simply because the only available recording site was in M1. We cannot exclude that, as in healthy participants, an effect, possibly even stronger, could have been observed in SMA if an electrode was also implanted there. An alternative explanation of the anatomical difference may be that, while the implanted participant was using genuine motor attempts to control the BMI, healthy participants had to use motor imagery, which is known to activate SMA more than M1. These considerations were added to the discussion on page 21:

“Interestingly, in healthy participants, the phase opposition effect was stronger in SMA than in M1. Since the invasive implant was limited to M1, we cannot exclude that a stronger phase opposition in SMA was also present in our tetraplegic participant but not directly observed.”

“Experiment 3 employed motor imagery rather than genuine motor attempts, as in Experiment 1. Given that motor imagery is known to induce comparably higher activations in SMA than in M1⁵⁶, a possibility, to be addressed in future work, is that the relative contributions of SMA and M1 vary depending on the modality of BMI control (e.g., imagery versus execution).”

We would also like to add that, in our opinion, the key common result cross-validating findings across Experiment 3 and Experiment 1-2 is not just that the phase opposition effect was found in the pre-movement period. Indeed, these results pertain the oscillatory phase, a very specific and peculiar feature of neural potentials, they were observed at similar frequencies (tied to the alpha peak), and effects in the contralateral M1 were significant in the implanted participant and in healthy controls. Further effects in SMA, a region that was simply not recorded in the implanted participant, were clearly observed only in healthy controls.

C13)

Theoretical issues.

A paper of this magnitude cannot be neutral towards the previous theories on the sense of agency. There are at least three theoretical models proposed to address the arising of the agency experience. The authors seem to consider only one of these nor they do make an effort to discuss whether and why their findings are in support of this specific theory. The "Comparator Model", the only model mentioned in the paper, postulates that the sense of agency arises from a series of comparator processes within the motor control system. However, in an alternative perspective, the "Apparent Mental Causation Theory" proposes that the emergence of the sense of agency is a post hoc and retrospective phenomenon. Accordingly, the sense of agency arises through an inferential sense-making process that occurs after the completion of a movement. Finally, in the "Cue Integration Theory", the sense of agency should arise from integrating low-level cues (sensorimotor proprioceptive and exteroceptive cues) and high-level cues (cognitive cues). The weight assigned to these different cues varies for the implicit and explicit components of agency. Hence, I feel that the authors should try to frame their data considering the previous theoretical and experimental efforts on the very same topic over the last 20 years.

R13)

Thank you for this suggestion. We are glad to discuss our results on pre-movement signals affecting the sense of agency in line with the current most relevant theories on this topic. The presence of a pre-movement signal correlating with the subsequent sense of agency (and neural connectivity) resonates with key elements of the “predictive” comparator model. Alpha band oscillations, gating cortical connectivity, are in our view a promising candidate to be the neural substrate carrying sensorimotor predictions from motor to sensory areas, and performing the subsequent comparisons. This mechanism is also compatible with the low-level components in the cue integration theory, while higher level neural processes on a slower temporal scale may contribute to postdictive inference. At the same time, our findings do not exclude that post-movement, retrospective inference of mental causation may also play a role in the phenomenology of agency. We have now integrated these aspects in the discussion on page 20:

“In our previous work¹³ (including data from Experiment 1), we showed how post-movement LFPs and multiunit activity in M1 encode congruency between motor commands and sensory feedback, a key aspect in the sensorimotor comparisons underlying the sense of agency. Our findings in Experiment 1 extend these previous results, suggesting that M1 and SMA also play pivotal roles in the sense of agency at an earlier stage, when sensorimotor predictions are computed during motor preparation and before execution. Our results do not rule out the contribution of post-movement signals to the sense of agency. These may be relevant for postdictive inference of causality (apparent mental causation theory⁷), or for sensorimotor comparisons, whereby sensorimotor oscillations may be integrated with higher level cognitive cues to

determine the final experience of agency (cue integration theory ⁶). This contribution may be reflected in a dissociation of post-movement LFP between high and low agency trials, which was here observed in Experiment 2 and extensively investigated in ¹³. Pre-movement sensorimotor oscillations may also serve as a trigger for apparent mental causation, and gauge the integration between low- and high-level cues in determining the sense of agency.”

In addition to the comparator model, we now also mention the alternative theories of the sense of agency in the introduction on page 3:

“Different explanations have been offered for the sense of agency ⁶⁻⁸. Among those, one influential view states that the sense of agency arises from the comparison between predicted and observed sensory outcomes of intended actions ^{8,9}.”

C14)

Style.

For sure, there is already a broad body of literature investigating the neural correlates of the sense of agency that makes an opening statement in the abstract (“..... the underlying neural mechanisms of the sense of agency are still unknown”) too strong for two reasons: (1) the previous evidence connecting brain physiology and the SOA (2) the correlational nature of the evidence provided; this lacks the identification of a clear "mechanism" that causally connects brain physiology and the complex phenomenology of the SOA.

R14)

We are sorry if we gave the impression of stating that no previous knowledge of the neural mechanisms of the sense of agency was available, which is not the case. We more specifically meant that the dynamical interaction between functional areas is still poorly understood, as most previous studies are either “static” fMRI activation maps, single-site lesion or perturbative studies. We have rephrased the abstract and introduction accordingly. Furthermore, we have thoroughly revised the text to clarify that we do not provide causal evidence about the role of pre-movement oscillations (pp. 4, 12, 19, 21). We also now acknowledge the correlational nature of our evidence in a new limitations paragraph which was added to the discussion on page 22:

“Nonetheless, without direct experimental intervention, the current evidence remains correlational. Further studies are needed to causally assess the contribution of pre-movement phase to sense of agency, both when measured explicitly and implicitly.”

Reviewer #2 (Remarks to the Author):

The manuscript by Bertoni et al reports very interesting data and experimental designs. The authors recorded single neurons and LFP from M1 of a tetraplegic patient who, with the help of a prosthetic device, is able to guide fake-hand reaching movements. The authors correlated the LFP with the reported simulated action of the subject and specifically with his sense of agency, demonstrating a phase dependence of alpha band endogenous oscillation with the sense of agency. To complete the study the authors measured the effect in typical subjects, and used high density EEG coupled with simulation of the moving hand to demonstrate again a phase dependence. Unfortunately and not surprising this data are less compelling with a small and just significant effect.

While I like the paper and the material presented, I have several unclear and not-discussed technical problems that make me uneasy to give a strong recommendation at this stage. The main one being the synchronization with the prosthetic device. For this reason I would like to postpone my acceptance/rejection recommendation after a major revision giving the possibility to the authors to explain better their technique and rationale and write a more clear and critical manuscript.

We thank the reviewer for their appreciation of our work, the many insightful comments, and the possibility of clarifying technical problems. A detailed response to the reviewer's main concerns, as well as other major points are provided in the following paragraphs. We highlighted our responses in bold, and indicated with quotation marks and *italics* passages reported from the revised text, highlighting changes in red. To allow referring to specific points, we have numbered reviewer comments and our responses (e.g., comment 1: C1; response 1: R1). The numbering continues across comments from the two reviewers.

C15)

Main problem

Phase depends on the time origin of the synchronisation trigger. Here the authors trust the prosthetic algorithm and set the synchronisation with the onset threshold for the fake-hand movement. However, this relies on a very strong assumption: the algorithm should operate with the same delay for the high and low agency perception. This may be wrong and would be mainly CONTRARY to their data. They show that the agency state signal can be derived from the phase of the LFP and supposedly also the decoding algorithm is using this signal, possibly by introducing a delay (50 ms would be sufficient) to give an apparent phase change. In other words, the argument to me is circular!

R15)

We understand the reviewer's concern, and apologize for an insufficiently clear presentation of technical aspects of the neuroprosthesis setup which may have led to a few critical misunderstandings.

First, the feature used by the BMI decoder in the implanted participant is the high frequency (>235 Hz) power (not phase) of the LFP signal, reflecting multiunit firing rate at each electrode. Due to the extreme gap in frequencies, and the fact that the power and not the phase of oscillations are considered, the BMI features are essentially independent of the 8 Hz phase. BMI input features are further smoothed over a 500 ms window, spanning and averaging several alpha cycles, and processed in externally determined 100 ms bins; thus, there is no reason to expect a dependency between BMI decoder threshold crossing and 8 Hz phase.

We report the passage detailing the input features of the decoder from page 24:

"The SVM used 96 input features consisting of the mean wavelet power (MWP) for each channel and 100 ms bin. To obtain the MWPs, neural activity was decomposed into 11 wavelet scales (Daubechies wavelet, MATLAB), and the coefficients of wavelets 3-6, corresponding to the multi-unit frequency band spanning from 235 to 3.75kHz, were averaged for each channel. Thus, the decoder's input features were closely

related to high frequency power at each channel, a robust and computationally not intensive proxy of multi-unit activity.”

We also have added further details in the methods to provide all the necessary information regarding this point, on page 25:

“At the end of each 100 ms acquisition bin, the decoder analysed neural signals and provided four numbers in the -1/1 range, indicating the decoded relative probability for each of the four movement. Again every 100 ms, the output of the decoder was further smoothed on a 500 ms time window to determine whether and which movement to implement.”

To support empirically that there is no dependence between decoding delay and 8 Hz phase/agency, we compared the decoding delay between high and low agency trials, and between trials starting in the optimal vs. non-optimal phase. No significant difference was found. These analyses are mentioned in the methods (page 32) and are shown in the Supplementary Materials (Fig. S13):

“To rule out that the relation between pre-movement phase and sense of agency could be due to differences in BMI decoding delays for high and low agency trials, we compared the delay between go cue and NMES movement onset between high and low agency trials, and between trials with optimal vs. non-optimal phase. We found no difference ($t(29) = 0.15$, $p = 0.84$ and $t(29) = 0.71$, $p = 0.48$ respectively, Fig. S13).” (p. 32)

We report Fig. S13 and its caption below:

“Figure S13: Control analyses on the decoder delays. The left panel shows the delay between go cue and NMES movement onset compared for high and low agency trials ($p = 0.84$). The right panel compares the same delay between trials starting in the optimal vs. non-optimal phase ($p = 0.48$). The grey bars represent standard errors.”

In addition to this, even if present, a “constant” delay of 50 ms (or any duration) may potentially affect agency judgements, but cannot lead to a systematic phase shift. This is because the “initial” phase (at the timepoint relative to which the delay is defined, e.g., the go cue) is randomly determined by the timeline of the experiment, which is independent from the phase of endogenous neural oscillations. Even if the phase at “time 0” was always the same, which is not the case, a constant delay would not introduce a phase shift, since the BMI decoder takes more than one second to trigger a movement (more than 3 with EEG), and temporal correlation between initial and current phase is lost after only a few oscillatory cycles, due to the different duration of each oscillatory cycle in noisy physiological processes. The average delay and its variability are now mentioned in the methods on page 25:

“Due to decoder output smoothing, plus neural and acquisition noise, the delay from the go cue to decoder threshold crossing was variable and significantly larger ($1.2\text{ s} \pm 0.48\text{ SD}$) than in natural movements. Being an expert user of the BMI system, the participant is accustomed to these long delays, allowing him to experience a strong sense of agency for BMI movements.”

Thus, the circularity mentioned by the reviewer could be present only if the decoder selectively time-locked its threshold crossing to a specific 8 Hz phase based on the agency judgement reported by the participant at the end of the trial. In other words, the decoder would need to be able to predict the participant’s agency judgement before the movement starts, and trigger the movement in the optimal phase when a positive judgement is predicted, and vice versa. This is clearly not possible. Finally, it is important to add that the phase effect is pre-movement and visible even at fixed sensory feedback (Fig. 2g), so it is independent of exogenous factors, such as sensory feedback, which may potentially interact with decoder output.

All the above reasonings also apply to the EEG experiment in healthy participants. In addition, in the EEG experiment the go-movement delay is regressed out from agency ratings to enhance the contribution of endogenous factors (i.e., pre-movement phase), and cannot thus contribute to our analysis.

For all these reasons, we believe there is no circularity between our phase-dependent effect on the sense of agency and how the BMI system decodes the participant’s motor intention.

C16)

However, the authors have access to single cell firing. A simple threshold to the ensemble recording firing could be used for synchronize the LFP and analyse the change in phase. It is true that they found a strong correlation between pre-movement of fake-hand LFP and neuronal firing. However, the amount of firing may be independent of onset phase. Maybe I misunderstood something, and the tetraplegic patient does not modulate at all M1 firing during Motor Imagery. But if so, this needs to be explained in detail, at least in the methods. In addition the variability of the decoding delay to generate the phantom hand movement needs to be reported!

R16)

We thank the reviewer for giving us the opportunity to discuss these important aspects of our analysis. First, let us clarify that differently from healthy participants, the tetraplegic participant did not perform motor imagery, but controlled his own hand by attempting actual hand movements, which was decoded by the BMI and triggered neuromuscular electrical stimulation of his forearm muscles (see Fig. 1 and caption). Thus, he did modulate M1 firing rates to generate hand movements, as shown by previous studies (Bouton et al., *Nature*, 2015; Serino et al., *Nature Human Behaviour*, 2022; Noel et al., *Biorxiv*, 2023) and the current data (see for example Fig. 5d, showing the buildup of M1 firing rate leading to the decoder crossing threshold). Movement-specific patterns of M1 firing were detected by the BMI system to recognize the intended movement. We have better clarified these aspects in the methods on page 24:

“The BMI system consisted in a 96 channels Utah array (Blackrock microsystems) acquiring M1 signals, a standard desktop computer decoding the intended movement from M1 activity, and an electrode patch stimulating right forearm muscles to translate decoded movements into functional hand movements. To account for natural changes in the signal from the Utah array, the decoder was re-trained before each experimental session. Training data was generated by asking the participant to attempt performing one of four hand movements (hand open, HO, hand close, HC, thumb extension, TE, thumb flexion, TF). Clearly, due to chronic paralysis, these motor attempts did not lead to actual hand movements as long as electrical stimulation was off, but modulated M1 firing rates in movement-specific patterns, which could be detected by the BMI algorithm. In each training session, the subject performed 7 blocks consisting of 3 repetitions per movement type each.”

Following the interesting suggestion of the reviewer, we now also analysed phase opposition after time-locking to a simple threshold on M1 firing rates. We found no significant phase opposition, as now stated in the methods (p. 32) and shown in the Supplementary Materials (Fig. S14):

“We also tested whether phase opposition results could be replicated by time-locking the LFPs to the crossing of a simple threshold on average M1 firing rates, rather than to the onset of the actual hand movement, and found no significant result (Fig. S14). We believe this was the case because the BMI decoder does not produce a movement by threshold crossing of population activity, but rather by decoding specific activity patterns across neurons³³. Thus, population rate threshold crossing is an internal event which, unlike BMI decoder crossing threshold, does not produce a movements or salient subjective percepts.” (p. 32)

Expanding the reasoning above, we believe the analysis did not yield significant results for one technical and one conceptual reason.

Technically, in each trial, the participant performed one of four (or two) randomized movements, each corresponding to a different pattern of M1 activation across the 96 channels of the Utah array. In addition, signals from the Utah array changed across experimental sessions due to natural movements of the electrode array in the cortical tissue (hence the necessity to re-train the decoder at each experimental session). Thus, the modulation of M1 firing rates varied across trials and experimental sessions, making a simple threshold on M1 firing not suitable to capture the onset of the participant’s motor commands. Furthermore, being used to the response delays of the BMI system and unable to generate natural movements, the participant does not generate strong bursts of M1 activity with a clear onset which could be used for time-locking, but slowly builds up M1 activity, gradually increasing his effort until a movement is generated (see for example firing rates in Fig. 5d). In other words, the best available proxy of the onset of the participant’s motor attempt is likely the neural decoder itself, which was optimized to detect complex patterns of activity corresponding to specific movements in each experimental session.

Besides these technical aspects, there is a more important and interesting conceptual aspect that in our view explains this null result. The analysis presented in the paper was not simply time-locked to the BMI decoder’s output, but to an actual sensory event with a clearly perceivable onset, i.e., the onset of the NMES stimulation inducing a hand movement, and the simultaneous onset of a virtual hand movement displayed on the screen. As now more clearly stated in the methods, NMES and visual stimulation invariably followed BMI threshold crossing by a fixed hardware delay. Due to the small variability of such delay (10 ms) compared to the duration of one 8 Hz oscillatory cycle, BMI threshold crossing and movement onset cannot be dissociated in our analyses (the average delay between go cue and BMI threshold crossing is already reported in the previous response). Our analyses thus reveal that it is the 8 Hz phase relative not only to BMI crossing threshold, but also (in our opinion, more importantly) to the onset of the hand movement and reafferent sensory feedback, that correlates with the sense of agency. In other words, the state of the brain in the precise moments preceding the onset of a movement determines the subsequent sense of agency. We have rephrased several paragraphs in the methods to better highlight the technical aspects underlying this reasoning (page 25 and 30):

“The participant was instructed to start attempting the cued movement when the go cue appeared, without anticipating. During the 4 seconds following the go cue, the BMI algorithm decoded changes in M1 multiunit activity generated by the participant as he attempted the cued movement, and translated them into visual and somatosensory feedback, according to the decoded movement and the feedback congruency assigned for that trial and sensory modality. Somatosensory feedback was delivered by eliciting the target movement through the NMES sleeve, and thus consisted in a functional hand movement. Visual feedback was constituted by an animation of a virtual hand performing the target movement, displayed on a screen placed horizontally to cover the participant’s right hand. Note that the participant has sufficient residual proprioception to recognize the hand movement performed even with his real hand being hidden by the screen (see^{13,33}).” (p. 25)

“Due to hardware delays, the onset of NMES stimulation followed the end of the 100 ms acquisition bin in which the neural decoder crossed threshold by a stereotyped 70 ± 10 ms (SD) delay.” (p. 25)

“For both LFP and multiunit activity, the data was epoched by time-locking to the onset of hand movements (sensory feedback). The exact timing of the onset of hand movements (70 ± 10 ms after the neural decoder

crossing threshold) was determined by detecting the 50 Hz stimulation artifact induced by the NMES system.” (p. 30)

In sum, we believe that, to observe phase-agency correlations, it is conceptually not appropriate to time-lock our analysis to the crossing of a firing rate threshold. This is because, unlike the crossing of the BMI threshold, the crossing of a firing threshold would be a purely “internal” event, as it would not necessarily be time-locked to an actual salient sensorimotor event, i.e. the onset of a hand movement. Indeed, due to the reasons stated above, there is a significant variability in the delay between M1 firing rate threshold crossing and hand movement/BMI threshold crossing. Put differently, we believe that our analysis worked when synchronized to the decoder crossing threshold not because this moment corresponds to a “special” internal state, but because this moment also corresponds to the onset of an actual movement (excluding an almost fixed hardware delay which was corrected as described above). If a random delay with larger variability (of e.g. 100 ms) had been added between decoder crossing threshold and movement onset, we believe the phase opposition would have been observed only when time-locking to the movement onset, and not to decoder crossing threshold.

C17)

Again, I am surprised that the Readiness Potentials, usually associated with SMA activity, are never present in recordings, either in the patient or in the typical subjects. Probably they have been filtered out given the high-pass with 0.2Hz. If so I advise to analyse and show them. These potentials are so strong and rich of information about the internal motor state and motor intention of the participants that it may contain already important information about agency.

R17)

Thank you for this interesting suggestion. We cannot exclude that, in our BMI setups, readiness potentials may be weaker than in natural able-bodied movements. Nevertheless, readiness potentials are visible in our data, and should not be affected by the high-pass filtering, since the cutoff frequency is lower than that of typical readiness potentials (notice that in the implanted participant the high-pass filter cannot be removed as it was performed at the hardware level). Readiness potentials were simply not evident in the main text figures due to the scale and aspect ratio of the plots presented in the manuscript.

We have now analysed readiness potentials hypothesizing that, in addition to the phase effect, a stronger (i.e., more negative) pre-movement deflection would lead to a higher sense of agency. In Experiment 1 the negative deflection constituting the readiness potential started at about -1.5 s, and the high and low agency signals overlapped until about -0.5s. After -0.5s, the two curves slightly dissociate, with a stronger negative deflection for high agency trials. However, this effect was statistically weaker than the phase effect. The minimum p-value for a T-test on the readiness potential between high and low agency trials was 0.002, and did not survive cluster-based correction, compared to 0.0001 for phase opposition (we use p-values to compare effect sizes, since there is no standard way to measure effect size for phase opposition). Also, this effect was not replicated in Experiment 2 or Experiment 3. We have now mentioned this analysis in the results (page 6), and show it in detail in the Supplementary Materials (Fig. S4):

“As a complementary analysis, we also investigated the relationship between agency and readiness potentials, the negative deflection of the LFP thought to be a correlate of pre-movement neural activity⁴⁰. We observed a trend of higher agency trials associated with a stronger negative deflection of the readiness potential. However, this effect did not survive correction for multiple comparisons (see Fig. S4).” (p. 6)

We report Fig. S4 and its caption below:

“Figure S4: analysis of readiness potentials. To decouple slow readiness potentials from phase opposition effects, the data was low pass filtered at 5 Hz, and LFPs/ERPs were compared between high and low agency trials. The panels show LFPs/ERPs for Experiment 1, Experiment 2, Experiment 3 – M1, and Experiment 3 – SMA (left to right). Shades indicate standard errors, the horizontal dashed line at 0 is shown to highlight the negative deflection corresponding to the readiness potential, the vertical dashed line indicates the time of the movement. No difference survived multiple comparison correction across timepoints.”

We are grateful for this suggestion, as these additional results help clarify the differences between the present work and other classic EEG literature on the sense of agency.

C18)

Figure 5. Could you split the M1-LFP correlation between high and low agency trials? Did you obtain a difference in the correlation? Low agency may reduce correlation and this would be a stronger and physiological more important effect that the phase difference of LFP.

R18)

Thank you for this insightful suggestion. We have compared the spike-LFP coupling between high and low agency trials in Experiment 1 (the small number of trials in Experiment 2 does not allow to perform the same analysis). We performed the analysis both in the pre- and post-movement 1 second time windows, as smaller time windows would not contain enough spikes to estimate PLV values. We did not observe any striking differences, but there was a slight increase of PLV values for low agency trials at 11 and 12 Hz. This result is contrary to the hypothesis proposed by the reviewer (and our own expectations), it occurs at significantly higher frequencies than 8 Hz, and did not survive multiple comparison correction across frequencies. We now mention the analysis in the results (page 12), and show it in detail in the Supplementary Materials (Fig. S6).

“Moreover, LFP-spike PLV values did not differ trials between trials of high vs high agency (Fig. S6), suggesting that only the oscillatory 8 Hz phase, but not the amount of coupling between such phase and spiking activity, covaries with agency ratings.” (p. 12)

C19)

Results of Fig 6 are somewhat disappointing. If anything, the phase opposition seems from Fig 6f stronger around 500 ms post movement. The scale of Fig 6d is not really readable to judge the robustness of the effect.

R19)

Please note that Fig. 6e showed results from an example participant, since averaging potentials from all

participants would not best show the phase opposition effect due to the differences in individual frequency of the effect (see new analyses in Fig. 6f). We have thus changed the visualization of our result, directly showing the output of the permutations used for our statistical analysis. Fig. 6e now shows how the observed phase opposition value lies far from the distribution of phase opposition values obtained by shuffling agency ratings within participants 10000 times.

C20)

Why has figure 7f directionally phase effects only on temporal cortex? Blue means that that ROI was not analysed? Please explain.

R20)

Thank you for pointing out the potential lack of clarity. Indeed, Fig. 7f (now 7g) shows results on our 114 ROI parcellation only within the temporal cortex, because only the temporal cortex showed significant results for the directionality analysis at the coarser parcellation level of 8 cerebral lobes. We have better clarified this in the main text (p. 18) and in the figure caption (p. 19):

“To better localise this effect, we performed the same directionality analysis at a finer spatial resolution only within the temporal lobe (the only region showing a significant effect at the coarser spatial resolution), at the scale of the original 114 ROIs used for source reconstruction.” (p. 18)

“(h) P-values for the directionality analysis, as in panel (g), performed only within the temporal lobe at the finer spatial scale of the original source reconstruction (114 ROIs). Dark blue areas indicate ROIs outside the temporal lobe, which were not analysed at this parcellation level.” (p. 19)

Also, to better characterize the spatial features of the connectivity changes, we now also present the results of the previous connectivity analysis at the level of the 114 ROIs in addition to the level of cerebral lobes (Fig. 7d), as reported on page 17:

“When performing the analysis at a finer spatial scale with 114 ROIs, six regions survived FDR correction: three regions in the middle prefrontal cortex and anterior cingulate, two in the superior part of the posterior parietal cortex, and one region in the anterior temporal cortex (Fig. 7d).”

We have rephrased passages in the discussion to incorporate these new results, as visible on page 21:

“Specifically, changes in functional connectivity occurred between SMA and part of the posterior parietal cortex, a region which is has been classically associated with visual guidance of movements^{57,58} (Fig. 7d). Further connectivity changes were observed between SMA and the medial prefrontal cortex and anterior cingulate, two regions implicated in action selection and initiation⁴⁴. These results point at SMA as a key hub in a network of regions classically associated with the sense of agency and are in line with the hypothesis⁴⁴ that sense of agency may emerge from neural connectivity within such network.”

C21)

Please make clear if 8Hz is theta or alpha for you.... To me looks like a typical subject alpha band LFP.

R21)

We have been careful in phrasing because of considerable debate in the community about the exact definition and limits of frequency bands. In this respect, note that Reviewer 1 instead raised doubts on whether the effect in the implanted participant actually is in the alpha band, and on the coherence of the results between implanted participant and healthy controls based on small frequency discrepancies (see C11). To address this comment, in the revised manuscript we show that the phase effect in healthy participants is tied to the individual alpha peak, following interindividual differences (see R11 and Fig. 6f). We report here the reviewed passage in the results, on page 14:

“The SMA effect peaked at 9 Hz, close to what was observed in our implanted participant, but was relatively spread across the whole alpha band (Fig. 6d). Since alpha-band peak frequencies vary across individuals⁴⁵ and correlations of alpha-band activity with behaviour are stronger at frequencies closer to the individual alpha peak⁴⁶, we predicted that individual variations in agency-related phase opposition might reflect individual variations in alpha peak frequency. Confirming this prediction, the frequency at which maximal SMA phase opposition was found for each subject correlated with their individual SMA alpha band frequency of maximal power ($R = 0.46$, $p = 0.011$, Fig. 6f). This suggests that individual variations in the frequency at which the phase better predicts agency depends on individuals’ idiosyncratic alpha band peak.” (p. 14)

Thus, the effect is consistent between the implanted participant and healthy controls when put in relation to the individual spectral peak. Indeed, we believe the position relative to the spectral peak to be a better indicator that these effects may be linked to the same neural phenomenon, rather than the specific frequency in Hertz.

Reviewer #1 (Remarks to the Author):

I thank the authors for their response and the clarifications they gave about their experiment. Despite these clarifications, I maintain reservations regarding the experimental paradigm used and the conclusions that can be drawn from it.

We thank the Reviewer for the deep review of our paper. We reply point by point below. Our responses are highlighted in bold and changes the revised version of the manuscript reported here are highlighted in red.

Experiment 2. The authors compared action timing judgments that were given after voluntary and involuntary actions, pooling together operant (Hand closing movement followed by sound) and non-operant (Hand opening movement alone) conditions (page 7 of the rebuttal letter).

1st issue: pooling together different actions and different conditions. The sense of agency is defined as “the feeling of making something happen” (Haggard, 2017). Here, the authors are mixing together conditions in which (different) actions caused a sound (i.e., the action makes happen a tone) with conditions in which the action did not produce any kind of external feedback (i.e., the action does not make happen a tone). In other words, they considered two very different conditions as equivalent, according to the literature on the sense of agency (see also the next comment).

We thank the Reviewer for raising these additional points. To demonstrate that our effect is not simply confounded by pooling operant and non-operant trials, we now show in Fig. S7 (reported below) that both the phase and the behavioural effect hold even when only considering the key condition of classical intentional binding studies, operant trials (see pages 11 and 48):

“

Figure S7. Main results of Experiment 2 for operant trials only. Panel (a) shows the anticipation of perceived movement timing for voluntary vs involuntary movements ($p = 0.011$, Wilcoxon). Panel (b) shows the phase

opposition product for early vs. late perception of the movement within voluntary operant trials. The red contour denotes the significant cluster of phase opposition ($p = 0.039$)."

In addition, the Reviewer's comment refers to the conditions used in the standard intentional binding paradigm. We would like to restate that our paradigm was inspired by the intentional binding paradigm in that it uses temporal judgements, but differs from it for several reasons, which makes comparison with previous results (and external validation) difficult, as we clearly acknowledged in the discussion (page 22). As far as sense of agency for the hand movement is concerned, as the topic here, we believe that both operant and non-operant conditions can provide meaningful information, and should be included. Importantly, the crucial signal we are studying precedes movement onset, and should not be related to further consequences of the action, which may or may not occur much later, but to the action itself.

2nd issue: Comparing voluntary vs involuntary actions. The authors compared action timing judgments after voluntary and involuntary (unpredictable) actions. The two conditions clearly differ based on the fact that the passive movement cannot be predicted: "In the involuntary session, the movement was randomly generated via the NMES system". This may explain the delay in action timing judgment observed with respect to voluntary movements.

The authors correctly cited Haggard et al., 2002 mentioning that this paper compared active and passive movements: indeed, Haggard et al. compared action timing judgements in voluntary vs involuntary actions yet this was done AFTER subtracting for the same measures collected in control conditions: timing judgment for voluntary actions recorded in the action+sound trials compared with timing judgment for voluntary actions recorded in the action-alone trials OR timing judgments for TMS-induced actions recorded in the action + sound trials compared with timing judgments for TMS-induced actions recorded in the action-alone trials, see Table 1 of Haggard et al., 2002).

In the present Ms, the authors adopted a different approach that cannot control for possible intervening factors not directly related to the agency dimension (e.g., surprise effect by unexpected externally generated muscle twitches).

Figure 2, shown in Haggard et al., 2017 may help in clarifying my comment (below called "The Figure"). <https://www.nature.com/articles/nrn.2017.14>

The potential confounding effect of surprise was indeed not properly controlled in our first revision. To search for a potential effect of surprise, we have compared tone perception following voluntary and involuntary movements. If the anticipation of the perceived stimulus timing was driven by its predictability in the voluntary condition, as compared to the surprise generated by involuntary movements, we would also observe the same effect also for the tone. We could not detect such an effect ($p = 0.25$), as now mentioned in the main text (page 8) and shown in Fig. S5 (reported below).

Control analysis on the perceived time of the sound in the operant condition. The p-value reported is from a Wilcoxon rank sum test, as for all other behavioural analyses in Experiment 2.

We would also like to point out that, as visible in Noel et al.'s complete report about this experiment, not only does action perception get anticipated in voluntary movements, but intention perception gets postponed when it is followed by an action. This further supports the intention-action binding interpretation, in a condition where surprise cannot play a role, since intention is internally generated and precedes the action.

Regarding the 1st issue, the authors here are pooling together conditions represented by the fourth (operant condition) and second (non-operant condition) rows of the upper part of the Figure (voluntary-action condition).

For what concerns the 2nd issue, the authors are comparing the conditions illustrated in the second/fourth row of the upper part of TheFigure with the second/third row of the lower part of The figure. To sum up, the authors of the present Ms are mixing together conditions that, in the intentional binding literature, refer to experimental and control conditions. Accordingly, there is no isolation of the crucial judgements of intentional acts and ensuing consequences.

We believe that our further analyses directly address these potential issues. Nevertheless, in the revised text we have acknowledged these potentially confounding factors. We have further toned down our interpretation of these findings, focusing strictly on objective facts: in this particular setup, action anticipation was associated with voluntary vs. involuntary movements with the very same 8 Hz phase associated with high agency movements (see pages 8, 9, 11). For example (page 11):

“Therefore, the same pre-movement 8 Hz oscillatory phase was associated with a higher explicit judgement of agency, and with anticipated action timing perception, which was also observed in voluntary vs. involuntary movements.”

Finally, we restate that, although partially inspired by the intentional binding paradigm, this experiment does not aim at fully reproducing its conditions and effects.

Experiment 3. In the first version of the paper, the authors acknowledged that participants were invited to “keep the mean (agency) rating around 5”. I criticised this approach since it seems that they were actively inviting their subjects to give specific rating values and thus manipulating the dependent variable of the experiment.

In the revised version of the paper, they now write that participants: “were also asked to focus on the differences between trials rather than on the absolute levels of agency, to provide variable ratings using all the available range, and consider 5 as an intermediate point to distinguish between higher and lower agency levels.”

This is something very different, and I am not sure that such changes from the original to the revised version are acceptable.

On a related note, they confirmed that no agency manipulation was applied in this experiment. This was aimed at measuring a sort of “intrinsic noise” in agency ratings. I am not sure that such noise can be considered related to the agency dimension since it does not vary according to a specific agency manipulation.

It is crucial to notice that our instructions, even in their original formulation could potentially only bias *average* ratings. As clearly stated in our previous response, a bias in participants’ *average* rating would have strictly no effect on our analyses, which are based on a median split between individual trials. The Reviewer then states that our paradigm measures “intrinsic noise” in agency ratings and that “I am not sure that such noise can be considered related to the agency dimension since it does not vary according to a specific agency manipulation”. The Reviewer seems to imply that only variations in subjective reports which are linked to external manipulations can be genuinely linked to agency. However, literature about perceptual awareness and pre-stimulus oscillations, which inspired our analyses, is grounded on studies in which variations in subjective reports *at fixed experimental conditions* are used as the key dependent variable, indicating that variability in subjective reports is not mere “intrinsic noise” but can be meaningfully studied (e.g., Busch et al., J Neurosci, 2009; Hanslmayr et al., Curr Bio, 2013; Ai et al., J Neurophysiol, 2014). This endogenous variability in the ratings can provide important insights into perceptual and cognitive phenomena, directly related to spontaneous fluctuations in the underlying neural activity, without potential confounds due to external manipulations. For example, previous studies used this approach to highlight the role of occipital alpha phase in visual awareness (Mathewson 2009). Here, it allowed us to uncover the role of M1-SMA oscillations for the sense of agency. Importantly, studying endogenous variability in subjective ratings at equal sensory stimulation is arguably the only appropriate approach to study pre-movement signals. Indeed, since a hypothetical agency manipulation on sensory feedback could only affect signals after movement onset, it would be impossible to observe our pre-

movement phase effect by contrasting experimental conditions. Indeed, exogenous manipulations act as a confounding factor for pre-movement signals, so all the effects we report were observed *despite* the effect of these manipulations, rather than thanks to them. This is another reason for which previous studies try to minimise stimulus variability (see papers cited above). These considerations have been integrated in our manuscript, to better explain and support our paradigm and its rationale on page 28:

“A similar approach, keeping experimental stimuli constant and contrasting trials based on fluctuations in subjective reports, is the one routinely applied in similar previous studies, investigating the link between perception and pre-stimulus oscillations (e.g.,^{29–31}).”

Finally, in my previous comment, I mentioned: “Contrary to what was seen in the patient, the normal control EEG data do not reveal activity in area M1 but in the SMA. The authors do not seem to be bothered by this. Yet this discrepancy deserves more than a comment. Indeed, as the two signals come from different areas, measured with different techniques, some differences should be expected. The key common thing here is that the crucial -small- EEG signal that correlates with the agency measures comes before action. This is the more solid cross-validation from the patient to the normal controls.”

The authors replied: “Based on the previous literature about the neural bases of the sense of agency (see e.g., Haggard, Nature Neuroscience Reviews, 2017) and our results in the implanted patient (Experiment 1 and 2), we expected to observe phase opposition in motor and premotor areas. Consequently, we targeted our search for phase opposition on motor and premotor regions ($n = 12$), applying FDR correction for multiple comparisons within this network. This analysis revealed the strongest phase opposition in SMA, and that the second strongest phase opposition in M1 ($p = 0.023$). If focusing purely on the left M1, with the purpose of replicating the effect observed in the implanted participant, and thus dropping multiple comparisons correction, the uncorrected M1 effect is highly significant with $p = 0.003$. The uncorrected p-value is not reported for in the text, for being maximally conservative. In sum, phase position was significant in both SMA and M1. For completeness, we also reported the results of a whole-cortex search for phase opposition across all 114 ROIs of the cortex. This whole-cortex search confirmed significant phase opposition in the SMA after adjusting for 114 comparisons; in contrast, M1 did not survive multiple correction. We refrain from making interpretations of the null result for M1 in this whole-cortex search as this analysis is underpowered”.

I have the following comments:

1. Haggard 2017 did not mention M1, but temporo-parietal and premotor regions.
2. The whole brain analysis cannot survive a formal multiple comparison correction, and the authors commented on this result as a consequence of low power: I am not sure that this kind of comment is acceptable; given the simplicity of running an EEG experiment in normal controls, there is no excuse for low-

powered experiments. Given that expanding the same sample of subjects at this stage is not acceptable, a replication of this experiment with a fresh sample of subjects could make the whole story more tenable.

We apologise if our previous submission was confusing about the whole brain analyses, which were presented just as examples and were not performed with the highest statistical power. Indeed, FDR correction was mainly used to support the SMA result, as in our view prior evidence from Experiment 1 and 2 was sufficient to lift the requirement for multiple comparison correction for M1. However, we believe that it is important not to draw conclusions from non-significant results obtained with analyses not offering the highest possible statistical power, and we thus removed FDR-corrected analyses from the text. We would like to clarify that we do indeed have the statistical power and cohort sizes to perform a whole brain analysis. We have now performed the most appropriate analysis for whole brain data, which is cluster-based correction (Maris & Oostenveld 2007). This analysis is the most established one for whole brain effects in EEG or MEG data, as it provides the highest statistical power exploiting regularities in the temporal, frequency or spatial domain. In the revised version of the manuscript, we thus applied cluster-based statistics to all whole brain comparisons (Figs 6 and 7), in order to support our results with “formal whole-brain analysis”. Specifically, M1 showed significant whole-brain corrected results, as the cluster formed by the two contiguous regions M1 and SMA was significant as reported on page 14:

“A cluster of two regions survived multiple comparison correction across all 114 ROIs ($p = 0.04$). These regions correspond to the posterior part of the left supplementary motor area (SMA), showing the strongest effect with uncorrected $p = 0.0002$, and the left M1 (uncorrected $p = 0.003$), consistent with our results in the implanted participant.”

On these grounds, we hope we have fully addressed the statistical doubts raised by the reviewer, and that the reviewer will agree that replicating the experiment in a fresh sample of participants is no longer necessary.

On the conceptual level, we do not see major discrepancies between Experiment 3 and previous results. Our key results in Experiment 3 are found in M1 *and* SMA, a key region for intention and sense of agency, immediately upstream of M1 in the intentional chain, and adjacent and strongly connected to it. As stated in the discussion, SMA was simply not observable in the implanted participant, where we could not decide where to implant the electrodes, but may have presented an even stronger result, if recorded. Actually, it is entirely possible that SMA may be the key origin of the observed effect, driving the effect in M1.

In any case, since the Reviewer was clearly not convinced by the design and results of Experiment 3, instead of merely replicating it, we decided to re-analyse previous unpublished data based on a classic agency

manipulation to demonstrate the same agency-related pre-movement phase effect from M1 and SMA. In the experiment, we used a classical (Yes/No) agency rating and a temporal manipulation of visuo-motor delays, whereby movements of a virtual hand could randomly occur before or after the participant's real hand movements to generate variable agency judgements. These data again show significant phase opposition for Yes vs. No agency judgements, localized in the left SMA and M1, further addressing statistical concern about our M1 result in Experiment 3. Due to length constraints in Nature Communications, the replication of our results is briefly mentioned in the main text (page 14, see below) and detailed in the methods (pages 29-32 and 38) and Supplementary Materials.

“To further confirm that pre-movement alpha oscillations discriminate high-agency and low-agency actions, we analysed data from an independent cohort of 10 participants, who performed a classic agency judgment paradigm. Briefly, participants were asked to freely lift their index finger, while receiving congruent visual feedback from a virtual hand, superimposed on their own. Visual feedback was delivered at various temporal delays from their actual movement. At the end of each trial, participants were asked to report whether they felt agency or not for the virtual hand. Comparing trials with “yes” vs. “no” agency reports we identified the same phase opposition in alpha in M1 and SMA, thus confirming and further generalizing our results to a different experimental paradigm (Experiment 4, see methods and Fig. S11).”

We provide a snapshot of the replication of our results (From Fig. S11) in M1 and SMA below:

Replication of our key results in M1 (top, $p = 0.027$) and SMA (bottom, $p = 0.0055$) in an independent cohort of 10 subjects using a different paradigm.

We believe these new results, replicating for the fourth time the same phase-opposition effect from M1 and/or SMA, strongly support our main conclusion.

On a final note, the authors mentioned Wegner as a support to their approach: “Finally, concerning the validation of our Sham BMI approach. As correctly pointed out by Wegner in his mental causation theory, it is entirely possible for healthy individuals to experience a genuine sense of agency even when not truly in

control of an external event, as we believe was the case in our sham BMI setup.” However, please note that this model was based on a “reconstructive” vision of the sense of agency, whereby the agency experience is formed AFTER the execution of the action. This is something different from what the authors are supporting in this paper.

This is an interesting point. First, we would like to state that we do not wish to take a strong stance in the prediction vs. postdiction debate, since our data does not necessarily support either of these views, and there is increasing support towards the idea that both predictive and postdictive aspects contribute to the sense of agency (e.g., Synofzik 2013). Here, we mainly report that pre-movement signals contribute to the (subsequent) sense of agency, but do not investigate *when* the subjective feeling of agency arises. We cited Wegner’s work as a prominent example that a sense of agency can be experienced regardless of the true causal relationship between motor commands and sensory feedback. However, this is also compatible with a “predictive” view of the sense of agency: if sensory feedback is congruent with expectations, a sense of agency is experienced, regardless of whether that sensory feedback was truly caused by motor commands. This stems from the simple fact that the brain has no direct access to causal relationships, but only to sensorimotor contingencies upon which it infers causalities, as pointed out in the paper aiming to reconcile predictive and postdictive theories (Synofzik 2013).

Reviewer #2 (Remarks to the Author):

The authors addressed my main concerns about the technical issue very thoroughly and clearly. I now appreciate well the design of the experiment and I found that the additional analysis that they performed dismissed all the possible criticisms that I raised in the previous version, and I believe that the data strongly support the fact that the phase of pre-movement low-alpha rhythms in M1 and SMA encodes the sense of agency. Given the importance and the novelty of the message, the originality of the technical design and the fact that the intracortical recording in M1 human are rare and precious, I strongly support publication of the manuscript in Nature Communication.

We thank the Prof. Morrone for her appreciation of our work and the constructive additional feedback. We provide our point-by-point response below, highlighting our responses in bold.

However, in the present form the manuscript still needs an additional revision to meet the standard of the journal.

Clarity of the writing:

1) The introduction is very general and not informative about the state of art of mechanisms and circuits of sense of agency; the same criticism applies to the literature on phase of endogenous oscillations encoding

motor information. Also about half of the introduction is about the presentation of the logic and result of the experiments, that is not useful for the reader given the complexity of the experiments. In addition, the same data from the same patients have already been published. It is important that the readers know about this in the introduction. The authors should clearly state and describe the results already obtained and published and, if possible, these should be used to motivate this new research. This means a fresh rewrite of most of the introduction.

Thank you for the suggestion which we hope will make the introduction more useful to readers. We have extensively modified the introduction removing the anticipation of our results, and expanding the presentation of the current state of research for sense of agency and neural oscillations. We have also addressed the link with our previous work in the introduction, rather than in the results as in the previous version of the manuscript.

2) The language used in the paper to explain analysis procedures is not appropriate, using many technical terms that are used currently in the EEG laboratories, but often mathematically incomplete. One author is an excellent mathematician, and he should check accurately the language. For example equation 1, line 891, they not report what is W_i and in any case as it not mathematically correct. If W_i is the complex number associated with the frequency it should be added the term Arctangent of the imagery/real part...if it is already phase, it cannot be divided by the norm of the phase! Similarly, no indication of the other term in the equation, like the number of trials. Another example in lines 944 and 1335. The Hilbert transform is not needed to calculate the phase at the various frequency. What exactly has been done to calculate it? Another example appearing many times, time-frequency point is a colloquial term: there are maxima energy points or other point that can be marked on other specifications in the time-frequency domain. Please correct. Another example in line 1325. Normalization is associated mathematically to a division, while inspecting the figure there is clearly also a subtraction. Line 243 very mathematically unclear: do you mean that phase is expressed with respect to a different origin? In other words, that the phase has been rotated to take into account the difference in delay respect to experiment 1? In most cases I understood what the authors have done, because it is what normally it is done in EEG or LPF analysis: but please express the procedure in mathematically correct language in these examples and in many other instances.

Thank you for pointing out these inaccuracies. We provide below a point-by-point summary of our edits to improve the mathematical rigour of methods presentation.

-Line 891. The formula we used for inter-trial-coherence is the one used in the cited paper (VanRullen 2016):

$$ITC_{ALL} = |\sum_{i=1:n} \omega_i / |\omega_i|| / n \quad (1)$$

where ω_i represents the analytic signal, the complex number whose absolute value is the instantaneous amplitude at a given frequency, and whose argument is the instantaneous phase at that frequency. We apologize that this crucial information was missing from the original manuscript. It has now been added, as well as the indication about the number of trials (see page 33). In words, complex vectors representing amplitude-phase are first normalized to have unit length (division by the norm) and then summed, and the length of the resulting vector is divided by the number of trials. This way, if all phase angles are perfectly aligned, ITC will be equal to 1. If phase angles are random, ITC will be equal to the length of the sum of random vectors, approaching 0 as the number of trials increases, with $ITC \sim 1/\sqrt{n}$, similarly to what observed for Brownian diffusion.

-Line 944-1335. As done in (Kayser 2009, ref. 41 in the main text), the Hilbert transform, coupled with a causal filter, was used as an alternative way to Morelet wavelets to compute analytic signals ω_i (then used to compute phase opposition) to demonstrate that the observed effect originates before sensory feedback onset. The signal was first band-passed in the frequency range of interest, then the analytic signal ω_i was computed from the signal as the complex number whose real part is the original signal, and the imaginary part is the Hilbert transform. We have provided these details in the methods (see page 35).

-Line 1325. As now stated in the revised figure legend, firing rates were divided by their average on the first 100 ms time bin. Please note that the normalized firing rate values are not visible in the figure, as they were only used to time-lock LFP values (displayed in the figure).

-Line 243. Sorry for the lack of clarity. We actually meant that we chose to directly plot phase angles at -256 ms (maximal phase opposition for Experiment 1) also for Experiment 2, rather than at the time of maximal phase opposition for Experiment 2 (-342 ms). This aimed at allowing comparison between the two experiments without rotating the phases, which could introduce distortions, and is justified by the fact that phase opposition at -256 ms is strongly significant also in Experiment 2 (see e.g., Fig. 3g., indicating $p < 0.0001$ for -256 ms). We have clarified this in the text.

Data analysis:

The authors assess differences in phases between the two conditions using inter-trial phase coherence and apply an equation that maximize anti-phase difference. However, in principle the system could work in quadrature phase that offers other advantages. It is essential to show that a difference in phase of the 8Hz component of the average LFP between the two conditions is statistically significant and report the value. This can be obtained simply by applying circular statistics across the vector (amplitude and phase) cluster for the two conditions. I think that it is important to consider simultaneously amplitude and phase of the 8 Hz

oscillation.

Thank you for raising this interesting point. To simultaneously study amplitude and phase of 8 Hz LFP oscillations, we compared the analytic signals (ω_i) for high and low agency trials at 8 Hz, -256 ms, the time and frequency of strongest phase opposition, without normalizing them to unit length to include amplitude information (figure below). To the best of our knowledge, circular statistics is used for unit length vectors not including amplitude information. We instead used a Hotelling T^2 to simultaneously compare phase and amplitude of the LFP. The test was significant ($T^2(208) = 21.4, p < 0.0001$), indicating that phase differences were significant also when taking into account 8 Hz LFP amplitude. A further T-test on the sole amplitude was not significant ($T(208) = 1.03, p = 0.31$), confirming that the observed differences emerge from phase rather than amplitude differences. These results are summarised below and reported on page 34.

Real (x axis) and imaginary (y axis) part of analytic signals at -256 ms, 8 Hz for individual trials in Experiment 1, extracted through Morelet wavelets.

Figure 3h reports LFP amplitude, but it is never defined what exactly it is measured. Is it the amplitude at 8 Hz, so it becomes negative due to the phase? Or it is the value in voltage of the LFP at the time point chosen? Please explain also in the text.

The figure shows the raw, unfiltered LFP value, as now clarified in the text and figure legend. The analysis aimed demonstrating that pre-movement 8 Hz phase differences are stronger than post-movement raw LFP value differences, to address a concern raised by the other Reviewer.

Concetta Morrone

Reviewer #2 (Remarks to the Author):

The authors have dealt adequately to all my criticisms, and I strongly support publication in the present form

We thank again Prof. Morrone for her support of our work and the constructive review process.